# Fine-Tuning Without Forgetting In-Context Learning:
# A Theoretical Analysis of Linear Attention Models

**Chungpa Lee**[†][1]   **Jy-yong Sohn**[1]   **Kangwook Lee**[2][3][4]

## Abstract

Transformer-based large language models exhibit in-context learning, enabling adaptation to downstream tasks via few-shot prompting with demonstrations. In practice, such models are often fine-tuned to improve zero-shot performance on downstream tasks, allowing them to solve tasks without examples and thereby reducing inference costs. However, fine-tuning can degrade in-context learning, limiting the performance of fine-tuned models on tasks not seen during fine-tuning. Using linear attention models, we provide a theoretical analysis that characterizes how fine-tuning objectives modify attention parameters and identifies conditions under which this leads to degraded few-shot performance. We show that fine-tuning all attention parameters can harm in-context learning, whereas restricting updates to the value matrix improves zero-shot performance while preserving in-context learning. We further show that incorporating an auxiliary few-shot loss enhances in-context learning primarily on the target task, at the expense of degraded in-context learning ability on tasks not seen during fine-tuning. We provide empirical evidence from synthetic and real-world datasets consistent with the qualitative predictions of our theory.

## 1. Introduction

Transformer-based large language models have demonstrated remarkable performance across a wide range of tasks (Vaswani et al., 2017a; Devlin et al., 2019; Radford et al., 2019; Touvron et al., 2023; Gemini Team, 2023;

Achiam et al., 2023). In practice, these pretrained models are often adapted to downstream tasks to further improve task performance (Han et al., 2021; Zhao et al., 2023). Two dominant paradigms have emerged for this adaptation: in-context learning and fine-tuning.

In-context learning is the ability of a pretrained model to adapt to a target task through *few-shot* prompting with demonstrations (Brown et al., 2020; Wei et al., 2022). However, this ability incurs substantial inference costs, as it relies on long prompts (Agarwal et al., 2024).

By contrast, fine-tuning is a process of adapting a pretrained model to a target task by updating its parameters through additional training (Hu et al., 2022). Despite the additional training cost, fine-tuning remains the standard approach in practice due to its strong and efficient *zero-shot* performance at test time, as task-specific behavior is learned in the model rather than provided through long prompts.

Although both in-context learning and fine-tuning have been independently studied, their interaction is not fully understood (Mosbach et al., 2023; Lampinen et al., 2025). Recent empirical studies report a trade-off between these two paradigms: improving zero-shot performance through fine-tuning on a target task can degrade in-context learning (Wang et al., 2024a;b). This degradation is particularly concerning, as it can limit the in-context learning ability of fine-tuned models on tasks that differ from those encountered during fine-tuning.

Motivated by these empirical observations, we analyze how fine-tuning on a target task alters attention parameters and how these changes affect in-context learning performance. Using linear attention models (Katharopoulos et al., 2020; Schlag et al., 2021; Ahn et al., 2024) as an analytically tractable setting, we derive closed-form solutions for the optimal parameters and the corresponding test errors under different fine-tuning regimes. These regimes are characterized by whether all attention parameters or only the value matrix is updated, and by whether auxiliary few-shot losses are incorporated. We also provide empirical evidence consistent with the qualitative predictions of our theory.

Our main findings are summarized below and in Table 1.

[†]Work done while visiting the University of Wisconsin–Madison. [1]Yonsei University, Korea [2]University of Wisconsin–Madison, USA [3]KRAFTON, Korea [4]Ludo Robotics, Korea. Correspondence to: Jy-yong Sohn <jysohn1108@yonsei.ac.kr>, Kangwook Lee <kangwooklee@krafton.com>.

*Proceedings of the 43rd International Conference on Machine Learning*, Seoul, South Korea. PMLR 306, 2026. Copyright 2026 by the author(s).

*Table 1.* Summary of our theoretical results in Section 4. We consider four training regimes and report the test error (Definition 3.1) evaluated at the optimal parameters that minimize the training objective for each regime. Fine-tuning is performed on data generated from the target task vector $\theta_0 \in \mathbb{R}^d$. Zero-shot errors are evaluated on the in-distribution task $\theta_0$ used for fine-tuning, whereas asymptotic few-shot errors ($m, n \to \infty$) are evaluated on both $\theta_0$ and the out-of-distribution task $-\theta_0$. The test error ranges from $\sigma^2$ (best) to $\sigma^2 + \theta_0^\top \Sigma \theta_0$ (worst). The tags (best), (better), (worse), and (worst) indicate the relative ordering of test errors across regimes, with comparisons made separately within the zero-shot and asymptotic few-shot settings. Although the asymptotic few-shot test errors are smaller than the zero-shot error in some cases, this advantage relies on the assumption of infinitely many shots. In practical settings with a limited number of shots, fine-tuned models can achieve better in-distribution zero-shot performance than few-shot methods. The last column references the corresponding theoretical results for each training regime, including the analysis of few-shot errors with finite shots.

| Section | Training Regime | Zero-shot error on $\theta_0$ (in-distribution) | Few-shot error on $\theta_0$ (in-distribution) | Few-shot error on $-\theta_0$ (out-of-distribution) | Results |
|---|---|---|---|---|---|
| §4.1 | Pretraining | $\sigma^2 + \theta_0^\top \Sigma \theta_0$ (worst) | $\sigma^2$ (best) | $\sigma^2$ (best) | Corollary 4.1 Corollary 4.2 |
| §4.2 | Fine-tuning **all parameters** with the **zero-shot** loss | $\sigma^2$ (best) | $\sigma^2 + \theta_0^\top \Sigma \theta_0$ (worst) | $\sigma^2 + \theta_0^\top \Sigma \theta_0$ (worst) | Theorem 4.3 Corollary 4.4 |
| §4.3 | Fine-tuning the **value matrix** with the **zero-shot** loss | $\sigma^2 + \frac{2}{d+4}\theta_0^\top \Sigma \theta_0$ (better) | $\sigma^2 + \frac{1}{(d+4)^2}\theta_0^\top \Sigma \theta_0$ (better) | $\sigma^2 + \frac{1}{(d+4)^2}\theta_0^\top \Sigma \theta_0$ (better) | Theorem 4.6 Proposition 4.7 Corollary 4.8 |
| §4.4 | Fine-tuning the **value matrix** with the **zero-shot** and **few-shot** losses | $\sigma^2 + \frac{2}{d+4}\theta_0^\top \Sigma \theta_0$ (better) | $\sigma^2$ (best) | $\sigma^2 + \frac{4}{(d+4)^2}\theta_0^\top \Sigma \theta_0$ (worse) | Theorem 4.9 Proposition 4.10 |

- Fine-tuning all attention parameters to minimize the zero-shot loss can substantially degrade few-shot performance.

- Restricting fine-tuning to the value matrix while freezing the remaining attention parameters mitigates this degradation and preserves few-shot performance.

- Fine-tuning with the zero-shot loss and an auxiliary few-shot loss further improves few-shot performance on the target task, but may reduce few-shot performance on tasks that differ from those used during fine-tuning.

## 2. Related Work

**Understanding In-context Learning of Transformers.** There are studies that explain why Transformer models exhibit in-context learning (Garg et al., 2022; Min et al., 2022; Akyürek et al., 2023; Dai et al., 2023; Bai et al., 2023; Li et al., 2023; Shen et al., 2024; Mahankali et al., 2024; Zhang et al., 2024; Lin & Lee, 2024). One prominent line of work interprets in-context learning as implicit Bayesian inference over latent task variables (Xie et al., 2022). Complementary work provides explanations by showing that simplified Transformer architectures can implement learning algorithms from the prompt: Von Oswald et al. (2023) prove that a linear self-attention layer is equivalent to a step of gradient descent, and Ahn et al. (2023) further show that linear Transformers trained on regression tasks learn to implement preconditioned gradient descent for in-context learning. Motivated by this line of work, we adopt a theoretical formulation of Transformer models that assumes the pretrained model exhibits in-context learning capability.

**Fine-Tuning Each Projection Matrix in Transformers.** Recent work has studied matrix-wise fine-tuning of attention

by updating the query, key, and value matrices separately, and finds that fine-tuning the value matrix often yields better performance than tuning the key or query matrices (Yao et al., 2025). For example, Table 5 of Hu et al. (2022) shows that low-rank adaptation on the value matrix achieves stronger performance on WikiSQL (Zhong et al., 2017) than adapting the query or key matrices, while attaining performance comparable to that of fine-tuning all matrices under the two-standard-error criterion. However, these studies primarily evaluate performance on the target-task objective and do not assess whether fine-tuning degrades a pretrained model's intrinsic capabilities, such as in-context learning. In our work, we show that value-matrix fine-tuning can achieve competitive target-task performance while preserving in-context learning capability, thereby maintaining general-purpose few-shot performance.

## 3. Problem Setup

We begin by clarifying notation. Let $\mathbf{0}_d := [0 \cdots 0]^\top$ and $\mathbf{1}_d := [1 \cdots 1]^\top$, where $d$ denotes the vector dimension. Let $\mathbf{I}_d$ denote the $d \times d$ identity matrix. For $n \in \mathbb{N}$, define $[n] := \{1, 2, \cdots, n\}$, and let $\mathrm{diag}(a_i)_{i \in [d]}$ denote the diagonal matrix with $i$-th diagonal entry $a_i$. We use block concatenation; for instance, $[\mathbf{Z}\ \mathbf{z}]$ denotes the matrix obtained by appending the column vector $\mathbf{z}$ to the right of $\mathbf{Z}$. Moreover, $[\mathbf{Z}]_{i,j}$ and $[\mathbf{z}]_i$ denote the $(i, j)$-th entry of $\mathbf{Z}$ and the $i$-th entry of $\mathbf{z}$, respectively.

In this paper, we compare various Transformers (Vaswani et al., 2017b) in terms of their test errors in zero-shot and few-shot settings. Following prior work (Von Oswald et al., 2023; Ahn et al., 2023; Zhang et al., 2024; Mahankali

et al., 2024; Li et al., 2024; Kim & Suzuki, 2024; Huang et al., 2024; Wu et al., 2024; Gozeten et al., 2025; Zhang et al., 2025; Lee, 2025), we focus on linear Transformers (Katharopoulos et al., 2020; Schlag et al., 2021) for linear regression tasks. We formalize the regression tasks, data distribution, Transformer architecture, evaluation metrics, and training objectives as follows.

**Regression Tasks and Data Distribution.** We consider linear regression tasks specified by a task vector $\theta \in \mathbb{R}^d$, where a $d$-dimensional input $\mathbf{x} \in \mathbb{R}^d$ is mapped to a response $y \in \mathbb{R}$. Given a task $\theta$, data points $(\mathbf{x}, y)$ are generated independently as $\mathbf{x} \sim \mathcal{N}(\mathbf{0}_d, \mathbf{\Sigma})$ and $y = \theta^\top \mathbf{x} + e$, where $e \sim \mathcal{N}(0, \sigma^2)$ with $\sigma^2 > 0$. The covariance matrix $\mathbf{\Sigma} := \mathbf{U}\mathbf{\Lambda}\mathbf{U}^\top \in \mathbb{S}_{++}^d$ is positive definite, where $\mathbf{U}$ is orthogonal and $\mathbf{\Lambda} = \mathrm{diag}(\lambda_i)_{i \in [d]}$ with $\lambda_1 \geq \cdots \geq \lambda_d > 0$.

For a given task $\theta$, we sample a dataset $\{(\mathbf{x}_i, y_i)\}_{i \in [n+1]}$. We define $\mathbf{z}_i := \begin{bmatrix} 1 & \mathbf{x}_i^\top & y_i \end{bmatrix}^\top \in \mathbb{R}^{d+2}$ for $i \in [n]$, and $\mathbf{z}_{n+1} := \begin{bmatrix} 1 & \mathbf{x}_{n+1}^\top & 0 \end{bmatrix}^\top \in \mathbb{R}^{d+2}$, whose response entry is masked to zero. We then define the prompt matrix as

$$\begin{bmatrix} \mathbf{Z}_{[n]} & \mathbf{z}_{n+1} \end{bmatrix} = \begin{bmatrix} \begin{bmatrix} 1 & \cdots & 1 \\ \mathbf{x}_1 & \cdots & \mathbf{x}_n \\ y_1 & \cdots & y_n \end{bmatrix} & \begin{bmatrix} 1 \\ \mathbf{x}_{n+1} \\ 0 \end{bmatrix} \end{bmatrix},$$

where $\mathbf{Z}_{[n]} := [\mathbf{z}_1 \cdots \mathbf{z}_n] \in \mathbb{R}^{(d+2) \times n}$ stacks the $n$ context examples. Hereafter, we omit the subscript $n+1$ on $\mathbf{x}_{n+1}$, $y_{n+1}$, and $\mathbf{z}_{n+1}$ for notational simplicity.

**Transformer Architecture.** To model the regression task, we consider a linear self-attention layer that replaces the softmax function in standard self-attention with a linear function. For a prompt $\mathbf{Z}$ with $\mathrm{ncol}(\mathbf{Z})$ columns, the linear attention model $f(\cdot)$ is defined as

$$f(\mathbf{Z}; \mathbf{V}, \mathbf{Q}) := \mathbf{Z} + \frac{1}{\mathrm{ncol}(\mathbf{Z})} \cdot \mathbf{V}\mathbf{Z}\mathbf{Z}^\top \mathbf{Q}\mathbf{Z}, \tag{1}$$

$$\mathbf{V} := \begin{bmatrix} 0 & \mathbf{0}_d^\top & 0 \\ \mathbf{0}_d & \mathbf{V}_{11} & \mathbf{v}_{12} \\ 0 & \mathbf{v}_{21}^\top & v_{22} \end{bmatrix}, \quad \mathbf{Q} := \begin{bmatrix} q & \mathbf{0}_d^\top & 0 \\ \mathbf{0}_d & \mathbf{Q}_{11} & \mathbf{q}_{12} \\ 0 & \mathbf{q}_{21}^\top & q_{22} \end{bmatrix},$$

where $\mathbf{V}, \mathbf{Q} \in \mathbb{R}^{(d+2) \times (d+2)}$ denote the value projection matrix and the merged query-key projection matrix, respectively. Here $\mathbf{V}_{11}, \mathbf{Q}_{11} \in \mathbb{R}^{d \times d}$ are matrices, $\mathbf{v}_{12}, \mathbf{v}_{21}, \mathbf{q}_{12}, \mathbf{q}_{21} \in \mathbb{R}^d$ are vectors, and $v_{22}, q, q_{22}$ are scalars. Our formulation generalizes existing linear attention models (Ahn et al., 2023; Zhang et al., 2024; 2025); in particular, setting $q = 0$ in (1) recovers their model. Moreover, Ahn et al. (2024) show that linear attention models trained on regression tasks serve as a proxy for studying the optimization of Transformers, exhibiting behavior similar to standard Transformers in practice.

Based on the model $f(\cdot)$ in (1), we take the last entry of its output as the prediction of $y_{n+1}$ from $\mathbf{x}_{n+1}$. Specifically,

$$\hat{y}_{\mathrm{FS}}(\mathbf{Z}_{[n]}, \mathbf{x}; \mathbf{V}, \mathbf{Q}) := \left[ f\left( \begin{bmatrix} \mathbf{Z}_{[n]} & \mathbf{z} \end{bmatrix}; \mathbf{V}, \mathbf{Q} \right) \right]_{d+2, n+1}, \tag{2}$$

$$\hat{y}_{\mathrm{ZS}}(\mathbf{x}; \mathbf{V}, \mathbf{Q}) := \left[ f(\mathbf{z}; \mathbf{V}, \mathbf{Q}) \right]_{d+2}, \tag{3}$$

where $\hat{y}_{\mathrm{FS}}$ in (2) denotes the few-shot prediction from a prompt containing $n$ examples, whereas $\hat{y}_{\mathrm{ZS}}$ in (3) denotes the zero-shot prediction from a prompt containing no examples, i.e., using only $\mathbf{x}_{n+1}$.

**Evaluation Metric and Training Objective.** We evaluate model performance under zero-shot and few-shot settings using the following test error.

**Definition 3.1** (Test Error). For $n \in \mathbb{N}$ and a task $\theta \in \mathbb{R}^d$, the $n$-shot test error on the task $\theta$ is defined as

$$\mathcal{E}(\mathbf{V}, \mathbf{Q}; n, \theta) := \mathbb{E}_{\mathbf{Z}_{[n]}, \mathbf{x}, y | \theta} \left[ \left( \hat{y}_{\mathrm{FS}}(\mathbf{Z}_{[n]}, \mathbf{x}; \mathbf{V}, \mathbf{Q}) - y \right)^2 \right],$$

and the zero-shot test error on the task $\theta$ is defined as

$$\mathcal{E}(\mathbf{V}, \mathbf{Q}; 0, \theta) := \mathbb{E}_{\mathbf{x}, y | \theta} \left[ \left( \hat{y}_{\mathrm{ZS}}(\mathbf{x}; \mathbf{V}, \mathbf{Q}) - y \right)^2 \right].$$

Motivated by prior work demonstrating that Transformers exhibit in-context learning on linear regression tasks (Zhang et al., 2024; Ahn et al., 2023; Li et al., 2024; Gozeten et al., 2025), we model pretraining by minimizing the expected $m$-shot test error averaged over tasks. This objective provides an analytical surrogate for studying how in-context learning can emerge from pretraining in a tractable setting.

Concretely, let $m \in \mathbb{N}$ denote the number of context examples used in the pretraining loss, which is defined as

$$\mathcal{L}(\mathbf{V}, \mathbf{Q}) := \mathbb{E}_\theta \left[ \mathcal{E}(\mathbf{V}, \mathbf{Q}; m, \theta) \right]. \tag{4}$$

$m$ may differ from the number of examples $n$ used at test time. We fix $m$ for tractability, while pretraining may involve multiple context lengths and more general objectives.

Starting from the model pretrained via (4), we consider fine-tuning to improve zero-shot performance on the target task vector $\theta_0 \in \mathbb{R}^d$, by minimizing the zero-shot prompt loss

$$\mathcal{L}_{\mathrm{ZS}}(\mathbf{V}, \mathbf{Q}; \theta_0) := \mathcal{E}(\mathbf{V}, \mathbf{Q}; 0, \theta_0). \tag{5}$$

We characterize the resulting models under these training objectives and analyze the zero-shot and few-shot test errors defined in Definition 3.1.

# 4. Optimal Parameters and Test Error of Learned Linear Attention Models

In this section, we theoretically analyze the linear attention model $f(\cdot)$ in (1) under different training regimes. Starting

from a pretrained model, we consider fine-tuning variants that improve zero-shot performance on a target task $\theta_0$ by minimizing the zero-shot prompt loss in (5). These variants differ in which attention parameters are updated and whether an auxiliary few-shot prompt loss is included.

### 4.1. Pretraining

First, we consider a linear attention model pretrained using the loss $\mathcal{L}(\mathbf{V}, \mathbf{Q})$ in (4). During pretraining, the training data are generated from task vectors $\theta$ sampled from $\mathcal{N}(\mathbf{0}_d, \mathbf{I}_d)$. In the zero-shot setting, where $y$ is predicted directly from $\mathbf{x}$ without observing any additional examples, a single input $\mathbf{x}$ does not provide information about the underlying task vector $\theta$. Consequently, the Bayesian optimal predictor under squared loss relies on the prior mean $\mathbb{E}[\theta] = \mathbf{0}_d$, yielding zero prediction for all $\mathbf{x} \in \mathbb{R}^d$. By contrast, when few-shot examples are provided, the model can implicitly infer the shared task vector and adapt its predictions accordingly.

A linear attention model pretrained by minimizing the loss $\mathcal{L}(\mathbf{V}, \mathbf{Q})$ in (4) exhibits this behavior, which is formalized by the following corollaries. Proofs of all statements in this section are provided in Appendix A.3 and A.4.

**Corollary 4.1** (Optimal Parameters of Pretrained Models; Corollary of Theorem 1 in Ahn et al. (2023)). *Suppose that $\mathbf{V}$ and $\mathbf{Q}$ in (1) satisfy $q \geq 0$ and that $\mathbf{Q}_{11}$ is positive definite. Consider the loss $\mathcal{L}(\mathbf{V}, \mathbf{Q})$ in (4) with context length $m \in \mathbb{N}$. Then, for sufficiently large $m$, the following parameters globally minimize the loss:*

$$\hat{\mathbf{V}} = \begin{bmatrix} \cdot & \cdot & \cdot \\ \cdot & \cdot & \cdot \\ \cdot & \cdot & \frac{m}{m+1+d} \end{bmatrix}, \hat{\mathbf{Q}} = \begin{bmatrix} \cdot & \cdot & \cdot \\ \cdot & \mathbf{U} \operatorname{diag}\left(\frac{m+1+d}{(m+1)\lambda_i + \operatorname{tr}(\mathbf{\Sigma})}\right)_{i \in [d]} \mathbf{U}^\top & \cdot \\ \cdot & \cdot & \cdot \end{bmatrix}, (6)$$

*All unspecified blocks (denoted by $\cdot$) are zero. Moreover, $\mathbf{U} \operatorname{diag}\left(\frac{m+1+d}{(m+1)\lambda_i + \operatorname{tr}(\mathbf{\Sigma})}\right)_{i \in [d]} \mathbf{U}^\top \to \mathbf{\Sigma}^{-1}$ as $m \to \infty$.*

**Corollary 4.2** (Test Error of Pretrained Models). *Let $\hat{\mathbf{V}}$ and $\hat{\mathbf{Q}}$ be the parameters in (6) which minimize the loss $\mathcal{L}(\mathbf{V}, \mathbf{Q})$ in (4). Then, for any $\theta \in \mathbb{R}^d$, the following holds:*

$$\sigma^2 + \theta^\top \mathbf{\Sigma}\theta = \mathcal{E}(\hat{\mathbf{V}}, \hat{\mathbf{Q}}; 0, \theta) \geq \mathcal{E}(\hat{\mathbf{V}}, \hat{\mathbf{Q}}; n, \theta),$$

*if $n \in \mathbb{N}$ is sufficiently large such that*

$$n \geq \sum_{i \in [d]} a_i^2 \cdot \frac{\lambda_1 \|\theta\|_2^2 + \sigma^2}{a_d(2 - a_d) \cdot \lambda_1 \|\theta\|_2^2} - 1 \qquad (7)$$

*where $a_i := \frac{m\lambda_i}{(m+1)\lambda_i + \operatorname{tr}(\mathbf{\Sigma})}$. Moreover, the n-shot error is monotone decreasing in $n$, and thus the minimum error is*

$$\lim_{n\to\infty} \mathcal{E}(\hat{\mathbf{V}}, \hat{\mathbf{Q}}; n, \theta) = \sigma^2 + \theta^\top \mathbf{U} \operatorname{diag}((a_i-1)^2\lambda_i)_{i \in [d]} \mathbf{U}^\top \theta,$$

*which converges to $\sigma^2$ as $m \to \infty$, since $a_i \to 1$ for all $i$.*

**Few-Shot Performance Outperforms Zero-Shot Only with Sufficient Demonstrations.** Corollary 4.2 shows that the few-shot performance improves monotonically as the number of shots $n$ increases, but can be worse than the zero-shot performance for small $n$.

To illustrate this phenomenon, consider an isotropic setting in which inputs are sampled as $\mathbf{x} \sim \mathcal{N}(\mathbf{0}_d, \lambda \mathbf{I}_d)$ for some $\lambda > 0$. In this case, the condition in (7) under which the $n$-shot performance outperforms the zero-shot reduces to

$$n \geq \frac{da}{2-a}\left(1 + \frac{\sigma^2}{\lambda \|\theta_0\|_2^2}\right) - 1, \quad a := \frac{m}{m+1+d}. (8)$$

This condition is both necessary and sufficient, see Corollary A.15. When $n$ is below this threshold, few-shot performance is strictly worse than the zero-shot baseline, in particular for $n \in \{1, \cdots, d-2\}$.

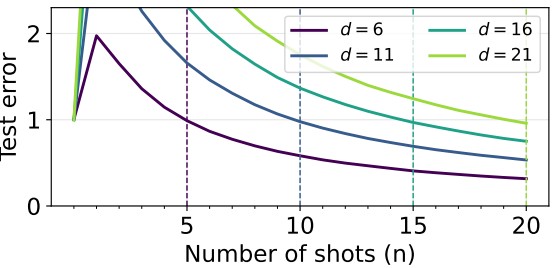

*Figure 1. $n$-shot test error on the target task $\theta_0$ of the pretrained model using the theory-derived parameters in Corollary 4.1. Each curve corresponds to a model pretrained with context length $m = 1000$ and a different input dimension $d$. We set $\sigma^2 = 0$, under which the condition in (8) predicts that the $n$-shot performance is worse than zero-shot performance for $n$ ranging from 1 to $d-2$.*

Figure 1 visualizes this effect for $m = 1000$ and $\sigma^2 = 0$, where the threshold in (8) is approximately $d-1$. This behavior arises from pretraining at a fixed and sufficiently large context length $m$, for which $a \to 1$. As a result, the same effect can appear even when multiple context lengths are used during pretraining, but may not persist if smaller context lengths are included. This phenomenon should not be confused with double descent or early ascent, as each curve corresponds to a final pretrained model.

The pretrained model specified by (6) exhibits two limitations. First, achieving improved performance requires a sufficiently large number of examples $n$. Second, in the zero-shot setting, the model predicts under the prior mean assumption $\mathbb{E}[\theta] = \mathbf{0}$, and therefore produces the same prediction regardless of the target task vector $\theta_0 \in \mathbb{R}^d$. This naturally motivates fine-tuning as a mechanism for specializing the model to a given target task.

For simplicity, unless stated otherwise, we assume the model is pretrained in the limit $m \to \infty$, so that the middle block $\hat{\mathbf{Q}}_{11}$ of $\hat{\mathbf{Q}}$ in (6) converges to $\mathbf{\Sigma}^{-1}$.

## 4.2. Fine-Tuning All Parameters with Zero-Shot Loss

We now study fine-tuned linear attention models, where all parameters $\mathbf{V}$ and $\mathbf{Q}$ are optimized to improve zero-shot performance on the target task $\theta_0 \in \mathbb{R}^d$. Fine-tuning starts from the pretrained model in (6) and minimizes the zero-shot loss $\mathcal{L}_{\mathrm{ZS}}(\mathbf{V}, \mathbf{Q}; \theta_0)$ in (5). The following results characterize the optimal parameters and the test error.

**Theorem 4.3** (Optimal Parameters of Fully Fine-Tuned Models). *Given a target task $\theta_0 \in \mathbb{R}^d$, consider the zero-shot prompt loss $\mathcal{L}_{\mathrm{ZS}}(\mathbf{V}, \mathbf{Q}; \theta_0)$ in (5). Then, for any $w > 0$, the following parameters globally minimize the loss:*

$$\hat{\mathbf{V}}(w) = \begin{bmatrix} \cdot & \cdot & \cdot \\ \cdot & \cdot & \cdot \\ \cdot & w\theta_0^\top & 1 \end{bmatrix}, \quad \hat{\mathbf{Q}}(w) = \begin{bmatrix} 1/w & \cdot & \cdot \\ \cdot & \cdot & \cdot \\ \cdot & \cdot & \cdot \end{bmatrix}. \quad (9)$$

**Corollary 4.4** (Test Error of Fully Fine-Tuned Models). *Given a target task $\theta_0 \in \mathbb{R}^d$, consider the family of parameters $\big(\hat{\mathbf{V}}(w), \hat{\mathbf{Q}}(w)\big)$ in (9). Then, for any $n \in \mathbb{N}$ and any $w > 0$, the following inequality holds:*

$$\sigma^2 = \mathcal{E}(\hat{\mathbf{V}}(w), \hat{\mathbf{Q}}(w); 0, \theta_0) < \mathcal{E}(\hat{\mathbf{V}}(w), \hat{\mathbf{Q}}(w); n, \theta_0).$$

*Moreover, for any $\theta \in \mathbb{R}^d$, the test errors are given by*

$$\mathcal{E}(\hat{\mathbf{V}}(w), \hat{\mathbf{Q}}(w); 0, \theta) = \sigma^2 + (\theta - \theta_0)^\top \mathbf{\Sigma}(\theta - \theta_0),$$
$$\lim_{n \to \infty} \mathcal{E}(\hat{\mathbf{V}}(w), \hat{\mathbf{Q}}(w); n, \theta) = \sigma^2 + \theta^\top \mathbf{\Sigma}\theta.$$

**Fine-Tuning All Parameters Degrades Few-Shot Performance Even on the Target Task.** Corollary 4.4 reveals a trade-off when all parameters are fine-tuned to optimize zero-shot performance. While the resulting model attains the minimum test error $\sigma^2$ in the zero-shot setting, this gain comes at the cost of degraded few-shot performance.

As the number of shots $n$ increases, the few-shot test error on a task $\theta$ converges to $\sigma^2 + \theta^\top \mathbf{\Sigma}\theta$, which can be substantially larger than the zero-shot error. Notably, this degradation persists even on the target task $\theta_0$, although the model is optimal for zero-shot performance on $\theta_0$.

Figure 2 illustrates this phenomenon. Although minimizing the zero-shot loss admits a family of optimal solutions parameterized by a free parameter $w > 0$ in Theorem 4.3, all such choices yield worse few-shot performance than zero-shot performance. In particular, for every $w > 0$, the few-shot error exceeds the zero-shot error and converges to $\theta_0^\top \mathbf{\Sigma}\theta_0$ as $n \to \infty$, which equals 1 in the setting of Figure 2.

These results reveal a limitation of full fine-tuning. While it optimizes zero-shot performance on the target task $\theta_0$, it degrades few-shot performance, making few-shot prediction strictly worse than zero-shot prediction even on the target task $\theta_0$. Thus, this degradation cannot be explained only by a mismatch between the fine-tuning task

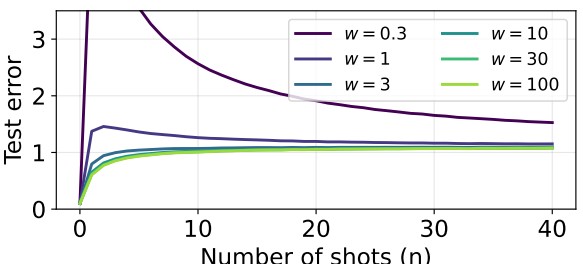

*Figure 2. $n$-shot test error on the target task $\theta_0$ of the fully fine-tuned model using the theory-derived parameters in Theorem 4.3. Each curve corresponds to a model with a different choice of the parameter $w$, evaluated under $\sigma^2 = 0.1$ and $\theta_0^\top \mathbf{\Sigma}\theta_0 = 1$. For all values of $w$, the zero-shot error is 0.1, whereas the few-shot error is larger than the zero-shot error and converges to 1 as $n \to \infty$.*

and the evaluation task. This observation motivates alternative fine-tuning strategies that preserve effective few-shot performance, which we study next.

## 4.3. Fine-Tuning Value Matrix with Zero-Shot Loss

To improve zero-shot performance on the target task $\theta_0$ while preserving few-shot performance, we revisit the structure of the optimal parameters derived in the previous subsections. A key insight from Theorem 4.3 is that, under full fine-tuning, the target task vector $\theta_0$ may be encoded in the value matrix through the parameter $\mathbf{v}_{21}$. In this sense, the value component of the attention mechanism provides a pathway to improved zero-shot performance.

We formalize this by characterizing when zero-shot performance outperforms few-shot performance, and show that this comparison is governed by the value parameter $\mathbf{v}_{21}$.

**Theorem 4.5.** *Suppose $\mathbf{V}$ and $\mathbf{Q}$ in (1) satisfy $v_{22} = 1$, $\mathbf{q}_{21} = \mathbf{0}_d$, and $\mathbf{Q}_{11}$ is positive definite. For any $\theta \in \mathbb{R}^d$, the following inequality holds:*

$$\mathcal{E}(\mathbf{V}, \mathbf{Q}; 0, \theta) \leq \mathcal{E}(\mathbf{V}, \mathbf{Q}; n, \theta),$$

*if and only if*

$$\big(\mathbf{v}_{21} - \mathbf{A}^{-1}\mathbf{B}\theta\big)^\top \mathbf{A} \big(\mathbf{v}_{21} - \mathbf{A}^{-1}\mathbf{B}\theta\big) \leq c, \quad (10)$$

*where $c := \theta^\top (\mathbf{B}\mathbf{A}^{-1}\mathbf{B} + \mathbf{C})\theta + (\mathrm{tr}(\mathbf{Q}_{11}\mathbf{\Sigma}\mathbf{Q}_{11}\mathbf{\Sigma}) + q^2)\sigma^2$, and $\mathbf{A}, \mathbf{B}, \mathbf{C}$ are matrices determined by $\mathbf{Q}, \mathbf{\Sigma}$, and $n$.*

Theorem 4.5 makes the role of the value parameter explicit. While $\mathbf{A}$, $\mathbf{B}$, and $c$ depend on $\mathbf{Q}$, $\mathbf{\Sigma}$, $\sigma^2$, $n$, and the task vector $\theta$ at which the test error is evaluated, they are independent of $\mathbf{v}_{21}$. Consequently, the inequality in (10) can be satisfied only through updates to $\mathbf{v}_{21}$. In particular, moving $\mathbf{v}_{21}$ toward $\mathbf{A}^{-1}\mathbf{B}\theta$ makes the condition hold, which corresponds to a regime where zero-shot performance outperforms few-shot performance.

At the same time, recall that few-shot performance in the pretrained model relies on the query-key matrix $\mathbf{Q}$, and in particular on the condition $\mathbf{Q}_{11} = \mathbf{\Sigma}^{-1}$ in Corollary 4.1. This observation suggests a fine-tuning strategy: rather than updating both $\mathbf{V}$ and $\mathbf{Q}$, we freeze $\mathbf{Q}$ to preserve the mechanism enabling few-shot performance and fine-tune only the value matrix $\mathbf{V}$ to improve zero-shot performance. We refer to this strategy as *value-matrix fine-tuning*.

The following theorem characterizes the models obtained by value-matrix fine-tuning under the zero-shot prompt loss.

**Theorem 4.6** (Optimal Parameters and Test Error of Value–Matrix Fine-Tuned Models). *Given a target task $\theta_0 \in \mathbb{R}^d$, consider the zero-shot prompt loss $\mathcal{L}_{\mathrm{ZS}}(\mathbf{V}, \mathbf{Q}; \theta_0)$ in (5). For any $w > 0$, define*

$$
\hat{\mathbf{V}}(w) = \begin{bmatrix} \cdot & \cdot & \cdot \\ \cdot & \cdot & \cdot \\ \cdot & \frac{1}{d+4}\theta_0^\top & w \end{bmatrix}, \quad \hat{\mathbf{Q}} = \begin{bmatrix} \cdot & \cdot & \cdot \\ \cdot & \mathbf{\Sigma}^{-1} & \cdot \\ \cdot & \cdot & \cdot \end{bmatrix}. \quad (11)
$$

*Then, $\hat{\mathbf{V}}(w)$ minimizes the zero-shot loss for fixed $\hat{\mathbf{Q}}$, i.e.,*

$$
\hat{\mathbf{V}}(w) \in \arg\min_{\mathbf{V}} \mathcal{L}_{\mathrm{ZS}}(\mathbf{V}, \hat{\mathbf{Q}}; \theta_0),
$$

*and the minimum value is $\sigma^2 + \frac{2}{d+4}\theta_0^\top \mathbf{\Sigma}\theta_0$. Under this choice of parameters, for any $\theta \in \mathbb{R}^d$,*

$$
\lim_{n\to\infty} \mathcal{E}\big(\hat{\mathbf{V}}(w), \hat{\mathbf{Q}}; n, \theta\big)
$$
$$
= \sigma^2 + \big(\tfrac{1}{d+4}\theta_0 + (w-1)\theta\big)^\top \mathbf{\Sigma}\big(\tfrac{1}{d+4}\theta_0 + (w-1)\theta\big).
$$

Under the constraint that $\mathbf{Q}$ is frozen at the matrix $\hat{\mathbf{Q}}$ learned during pretraining, the model cannot generally attain the irreducible minimum error $\sigma^2$ in the zero-shot setting. Nevertheless, the resulting zero-shot error, $\sigma^2 + \frac{2}{d+4}\theta_0^\top \mathbf{\Sigma}\theta_0$, is close to $\sigma^2$ for sufficiently large $d$. More importantly, by preserving the pretrained query-key matrix, value-matrix fine-tuning maintains strong few-shot performance and can outperform full fine-tuning in the few-shot regime.

The parameter $w$ in (11) does not affect zero-shot performance, so the zero-shot objective does not uniquely determine $w$. Because fine-tuning starts from the pretrained parameters in (6), it is natural to choose $w$ to make the update as small as possible. Among all zero-shot optimal solutions $\hat{\mathbf{V}}(w)$ in Theorem 4.6, we select the one closest to the pretrained parameters. The following results formalize this choice of $w$ and characterize the resulting model.

**Proposition 4.7.** *Assume $\mathbf{\Sigma} = \mathbf{I}_d$, and let $\hat{\mathbf{V}}$ and $\hat{\mathbf{Q}}$ be the parameters in (6) which minimize the loss $\mathcal{L}(\mathbf{V}, \mathbf{Q})$ in (4). Given a target task $\theta_0 \in \mathbb{R}^d$, consider $\hat{\mathbf{V}}(w)$ in (11). Then*

$$
\hat{\mathbf{V}}\left(\tfrac{m}{m+1+d}\right)
$$
$$
= \arg\min \left\{ \left\|\mathbf{V}-\hat{\mathbf{V}}\right\|_{\mathrm{F}}^2 \,\middle|\, \mathbf{V} \in \arg\min_{\mathbf{V}} \mathcal{L}_{\mathrm{ZS}}(\mathbf{V}, \hat{\mathbf{Q}}; \theta_0)\right\},
$$

*where $\|\cdot\|_{\mathrm{F}}$ denotes the Frobenius norm.*

*Moreover, if $m = n$, then for sufficiently large $n + d$, the choice $w = \frac{m}{m+1+d}$ closely approximates the task-averaged optimum $w^\star(n)$, defined in Corollary 4.8 below.*

**Corollary 4.8** (Task-Averaged Optimal $w$). *Given a target task $\theta_0 \in \mathbb{R}^d$, consider the family of parameters $\big(\hat{\mathbf{V}}(w), \hat{\mathbf{Q}}\big)$ in (11). For any $n \in \mathbb{N}$, the few-shot test error averaged over tasks $\theta \sim \mathcal{N}(\mathbf{0}_d, \mathbf{I}_d)$ is minimized at*

$$
w^\star(n) := \arg\min_{w} \mathbb{E}_\theta\Big[\mathcal{E}\big(\hat{\mathbf{V}}(w), \hat{\mathbf{Q}}; n, \theta\big)\Big] \quad (12)
$$
$$
= \frac{(n+1)\,\mathrm{tr}(\mathbf{\Sigma})}{(n+1+d)\,\mathrm{tr}(\mathbf{\Sigma}) + d\sigma^2}.
$$

*Then $w^\star(n) \to 1$ as $n \to \infty$. Hence, for any $\theta \in \mathbb{R}^d$,*

$$
\lim_{n\to\infty} \mathcal{E}\big(\hat{\mathbf{V}}(1), \hat{\mathbf{Q}}; n, \theta\big) = \sigma^2 + \frac{1}{(d+4)^2}\theta_0^\top \mathbf{\Sigma}\theta_0.
$$

**Value-Matrix Fine-Tuning Improves Zero-Shot Performance While Preserving Few-Shot Performance.** Proposition 4.7 fixes the free parameter $w$ using a minimal-update rule: among all value-matrix fine-tuning solutions $\hat{\mathbf{V}}(w)$ that minimize the zero-shot loss, it chooses the one closest to the pretrained matrix $\hat{\mathbf{V}}$.

Corollary 4.8 provides a complementary characterization of this choice from the few-shot perspective. It shows that, when tasks are drawn from $\theta \sim \mathcal{N}(\mathbf{0}_d, \mathbf{I}_d)$, the task-averaged few-shot test error admits a unique minimizer $w^\star(n)$ in (12). In the limit as $n \to \infty$, we have $w^\star(n) \to 1$, implying that the asymptotic few-shot error becomes identical for all tasks $\theta \in \mathbb{R}^d$. Moreover, when $m = n$ and $n + d$ is sufficiently large, the choice $w = \frac{m}{m+1+d}$ from Proposition 4.7 closely approximates $w^\star(n)$, which implies that the resulting value-matrix fine-tuned model preserves few-shot performance uniformly over tasks.

### 4.4. Fine-Tuning Value Matrix with Zero-Shot and Auxiliary Few-Shot Losses

Value-matrix fine-tuning with the zero-shot loss improves zero-shot performance while preserving few-shot performance. However, the zero-shot loss alone does not uniquely determine the value-matrix solution. As shown in Theorem 4.6, all solutions in the family $\hat{\mathbf{V}}(w)$ defined in (11) achieve the same zero-shot optimum for fixed $\hat{\mathbf{Q}}$, but different choices of $w$ lead to different few-shot behavior. Thus, the zero-shot objective leaves a degree of freedom that can be used to further adapt the model in the few-shot regime.

We first characterize how the few-shot test error depends on this remaining degree of freedom. In particular, the optimal value of $w$ depends on both the evaluation task $\theta$ and the number of demonstrations $n$ at test time. Therefore,

choosing $w$ to improve few-shot performance on the target task $\theta_0$ may not be optimal for other tasks.

Figure 3 illustrates this dependence by plotting the $n$-shot test error of models with parameters $(\hat{\mathbf{V}}(w), \hat{\mathbf{Q}})$ in (11) for different values of $w$. For each fixed $n$, there exists an optimal choice of $w$ that minimizes the $n$-shot test error on the target task. The following theorem formalizes this observation by characterizing the task-wise optimal choice of $w$ as a function of the evaluation task $\theta$ and the number of demonstrations $n$.

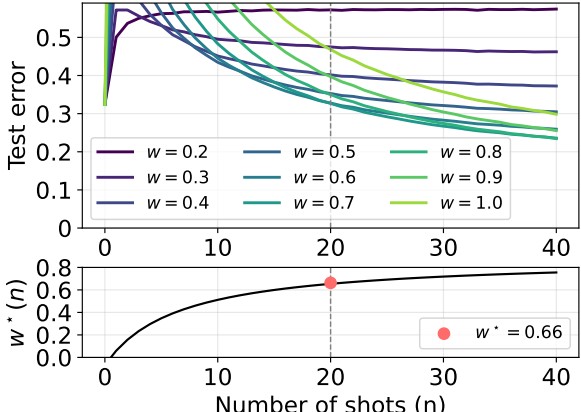

*Figure 3.* *(Top)* $n$-shot test error on the target task $\theta_0$ for the value-matrix fine-tuned model with the theory-derived parameters $(\hat{\mathbf{V}}(w), \hat{\mathbf{Q}})$ in Theorem 4.6. Each curve corresponds to a different choice of $w$. We use $d = 5$, $\theta_0^\top \mathbf{\Sigma} \theta_0 = 1$, and $\sigma^2 = 0.1$, for which the zero-shot test error equals $3.1 (= \sigma^2 + \frac{2}{d+4} \theta_0^\top \mathbf{\Sigma} \theta_0)$ for all models. *(Bottom)* Optimal $w^\star(n; \theta_0)$ that minimizes the $n$-shot test error on the target task $\theta_0$. The optimal value increases monotonically with $n$ and converges to $8/9 (= \frac{d+3}{d+4})$. For example, the 20-shot test error is minimized at $w \approx 0.66$.

**Theorem 4.9** (Task-Wise Optimal $w$). *Given a target task $\theta_0 \in \mathbb{R}^d$, consider the family of parameters $(\hat{\mathbf{V}}(w), \hat{\mathbf{Q}})$ in (11). For any $n \in \mathbb{N}$ and any $\theta \in \mathbb{R}^d$, the few-shot test error on the task $\theta$ is minimized at*

$$w^\star(n; \theta) := \arg\min_w \mathcal{E}(\hat{\mathbf{V}}(w), \hat{\mathbf{Q}}; n, \theta) \qquad (13)$$

$$= \frac{(n+1)\,\theta^\top \mathbf{\Sigma} \theta - \frac{n+3+2d}{d+4} \theta_0^\top \mathbf{\Sigma} \theta}{(n+1+d)\,\theta^\top \mathbf{\Sigma} \theta + d\sigma^2}.$$

*Then $w^\star(n; \theta) \to \frac{d+3}{d+4}$ as $n \to \infty$. Hence, for any $\theta \in \mathbb{R}^d$,*

$$\lim_{n\to\infty} \mathcal{E}(\hat{\mathbf{V}}(\tfrac{d+3}{d+4}), \hat{\mathbf{Q}}; n, \theta) = \sigma^2 + \tfrac{1}{(d+4)^2}(\theta - \theta_0)^\top \mathbf{\Sigma}(\theta - \theta_0).$$

For the target task $\theta_0$, Theorem 4.9 implies that the optimal value $w^\star(n; \theta_0)$ increases with $n$ and converges to $w = \frac{d+3}{d+4}$ as $n \to \infty$. Thus, the remaining degree of freedom in $w$ can be used to adapt the model to a desired few-shot regime.

To guide this adaptation during fine-tuning, we introduce an auxiliary few-shot prompt loss with $n$ demonstrations. For the target task vector $\theta_0 \in \mathbb{R}^d$, we define the few-shot prompt loss as

$$\mathcal{L}_{\mathrm{FS}}(\mathbf{V}, \mathbf{Q}; \theta_0) := \mathcal{E}(\mathbf{V}, \mathbf{Q}; n, \theta_0) \qquad (14)$$

and fine-tune using a combination of the zero-shot loss (5) and the auxiliary few-shot loss. Since the auxiliary loss depends on $n$, it guides the learned parameter toward values of $w$ that improve $n$-shot performance.

**Fine-Tuning with an Auxiliary Few-Shot Loss Improves Few-Shot Performance at the Expense of Generalization.** The task-wise optimal value $w^\star(n; \theta)$ in (13) depends on the evaluation task $\theta$. As a result, choosing $w$ to optimize few-shot performance on the target task $\theta_0$ need not be optimal for other tasks $\theta(\neq \theta_0)$. The following proposition quantifies this trade-off by characterizing the excess error incurred when the model uses $w^\star(n; \theta_0)$ instead of the task-wise optimal choice $w^\star(n; \theta)$.

**Proposition 4.10.** *Given a target task $\theta_0 \in \mathbb{R}^d$, consider the family of parameters $(\hat{\mathbf{V}}(w), \hat{\mathbf{Q}})$ in (11). Fix $n \in \mathbb{N}$. For simplicity, define $\mathcal{E}(w; \theta) := \mathcal{E}(\hat{\mathbf{V}}(w), \hat{\mathbf{Q}}; n, \theta)$ and $w^\star(\theta) := w^\star(n; \theta)$. For any $\theta \in \mathbb{R}^d$, the excess error is*

$$\mathcal{E}(w^\star(\theta_0); \theta) - \mathcal{E}(w^\star(\theta); \theta) \qquad (15)$$

$$= \frac{n\big((n+1+d)\theta^\top \mathbf{\Sigma} \theta + d\sigma^2\big)}{(n+1)^2} \cdot \big(w^\star(\theta_0) - w^\star(\theta)\big)^2 \geq 0,$$

*with equality if and only if $w^\star(\theta) = w^\star(\theta_0)$. Moreover, if $\theta$ satisfies $c := \theta^\top \mathbf{\Sigma} \theta = \theta_0^\top \mathbf{\Sigma} \theta_0$, then (15) reduces to*

$$\mathcal{E}(w^\star(\theta_0); \theta) - \mathcal{E}(w^\star(\theta); \theta)$$

$$= \frac{n(n+3+2d)^2 c^2}{(n+1)^2(d+4)^2\big((n+1+d)c + d\sigma^2\big)} \cdot \big(1 - \rho(\theta, \theta_0)\big)^2,$$

*where $\rho(\theta, \theta_0) = \theta^\top \mathbf{\Sigma} \theta_0 / c \in [-1, 1]$ is the cosine similarity induced by the $\mathbf{\Sigma}$-inner product.*

Proposition 4.10 shows that tuning $w$ using the auxiliary few-shot loss improves $n$-shot performance on the target task $\theta_0$, but can degrade performance on other tasks $\theta(\neq \theta_0)$. Moreover, the degradation increases with the discrepancy between tasks, quantified by the cosine similarity $\rho(\theta, \theta_0)$ under the $\mathbf{\Sigma}$-inner product. This formalizes a fundamental limitation of auxiliary few-shot fine-tuning: while it enhances adaptation to the target task in the few-shot regime, it may reduce generalization across tasks.

In Appendix A.5, we further analyze the test error when the zero-shot loss in (5) is combined with the auxiliary few-shot

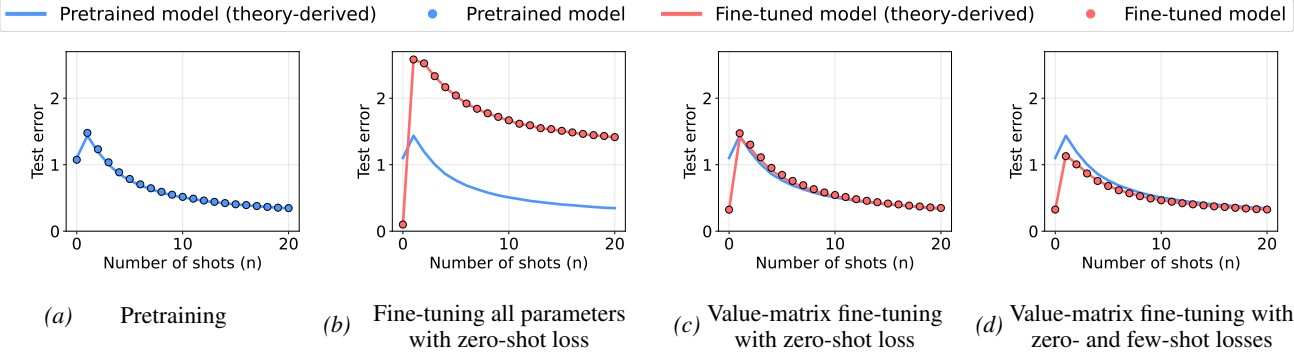

*Figure 4.* $n$-shot test errors on the target task $\theta_0$ for linear attention models under different training regimes. Curves show theoretical predictions from Section 4, while points correspond to models trained empirically in Section 5.1. Empirical results match the theoretical values. The parameters used for the theoretical predictions are given by: *(a)* (6) from Corollary 4.1. *(b)* (9) from Theorem 4.3 with $w = 0.52$. *(c)* (11) from Theorem 4.6 with $w = 0.77$ ($= \frac{m}{m+1+d}$) (Proposition 4.7). *(d)* (11) with $w = 0.66$ (Theorem 4.9; see Figure 3).

loss in (14). In particular, Theorem A.20 shows that an additional gradient descent step on the few-shot loss increases the zero-shot test error once the parameters approach the optimal zero-shot solution. Motivated by this observation, we adopt an annealing strategy for the auxiliary few-shot loss in the following experimental section, gradually reducing its weight to zero during fine-tuning.

## 5. Experiments

In this section, we provide empirical evidence for the theoretical results in Section 4 through experiments on linear regression tasks. We further complement these findings with experiments on fine-tuning a large language model on the MMLU benchmark (Hendrycks et al., 2021).

### 5.1. Experiments on Linear Regression Tasks

We train the linear attention model in (1) on linear regression tasks, where datasets are generated according to the setup in Section 3. Following prior work (Ahn et al., 2023), we set $d = 5$, $\sigma^2 = 0.1$, $\mathbf{\Sigma} = \mathbf{I}_5$, $\|\theta_0\|_2^2 = 1$, and $m = n = 20$. We first pretrain the model and then fine-tune it using three strategies described in Section 4. Further details are provided in Appendix B.1.

Figure 4 reports the test error on the target task $\theta_0$, where points denote empirical results and curves show the theoretical predictions from Section 4. Across all settings, the empirical trends are consistent with the theoretical predictions. Figure 5 further reports test errors on the out-of-distribution task $-\theta_0$ under different fine-tuning strategies. Fine-tuning all parameters with both the zero-shot and auxiliary few-shot losses attains low error on $\theta_0$, but exhibits higher few-shot error on $-\theta_0$. In contrast, value-matrix fine-tuning yields lower few-shot error on $-\theta_0$, indicating better retention of few-shot performance across tasks, an effect that is more pronounced without the auxiliary few-shot loss.

In Appendix B.1, we further evaluate architectural variants

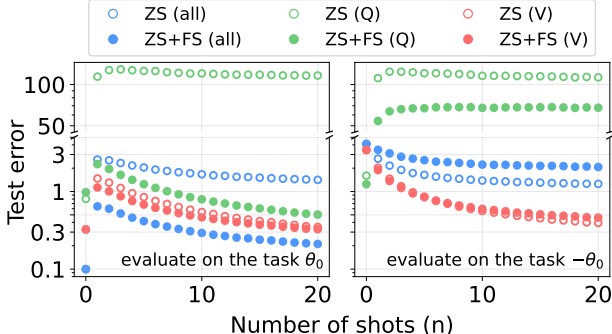

*Figure 5.* $n$-shot test errors evaluated on the in-distribution task $\theta_0$ *(Left)* and the out-of-distribution task $-\theta_0$ *(Right)*. Empty circles denote models fine-tuned with the zero-shot (ZS) loss only, whereas filled circles denote models fine-tuned with both the zero-shot and few-shot (FS) losses. Parentheses indicate which parameters are updated: `all` updates all parameters, `Q` updates the query-key matrix, and `V` updates the value matrix. Value-matrix fine-tuning without the auxiliary few-shot loss gives the lowest error on the out-of-distribution task $-\theta_0$.

with separate query and key matrices, as well as multi-head linear attention models. Under our framework, extending the analysis to multi-head linear attention is straightforward when the query-key parameters are shared across heads. In this case, the outputs from different heads form a linear combination of the corresponding value matrices, which is equivalent to a single-head model with an aggregated value matrix. Although our two-head experiments do not impose this shared query-key structure, they still exhibit qualitatively similar trends, suggesting that the observed behavior is not specific to the merged query-key formulation.

### 5.2. Experiments on the MMLU Benchmark

We complement our theoretical analysis with experiments on the MMLU benchmark using `Qwen2.5-3B-Instruct` (Yang et al., 2024). We fine-tune models to improve zero-shot accuracy on the *Humanities* category and evaluate both zero-shot and

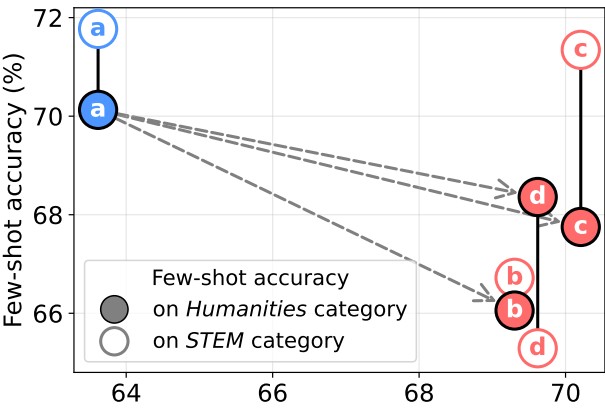

*Figure 6.* Zero-shot (x-axis) and few-shot (y-axis) accuracy of `Qwen2.5-3B-Instruct` on the *Humanities* and *STEM* categories of the MMLU benchmark. We compare the pretrained model with three models fine-tuned to improve zero-shot performance on *Humanities*. *(a)* **The pretrained model** exhibits higher few-shot than zero-shot accuracy on *Humanities*. *(b)* **Fine-tuning with the zero-shot loss** improves zero-shot accuracy relative to the pretrained model but degrades few-shot accuracy. *(c)* **Value-matrix fine-tuning with the zero-shot loss** preserves few-shot accuracy while achieving zero-shot accuracy comparable to *(b)* . *(d)* **Value-matrix fine-tuning with both zero-shot and few-shot losses** improves few-shot accuracy on *Humanities*, at the cost of reduced few-shot accuracy on other categories such as *STEM*.

7-shot accuracy on *Humanities* and *STEM*. We compare the pretrained model with three fine-tuning strategies.

Figure 6 illustrates trends consistent with our theoretical predictions. Full fine-tuning improves zero-shot accuracy on *Humanities* but degrades few-shot accuracy. In contrast, value-matrix fine-tuning preserves few-shot performance while achieving comparable zero-shot accuracy. Incorporating an auxiliary few-shot loss further improves few-shot accuracy on *Humanities*, but reduces few-shot accuracy on *STEM*. Additional results are provided in Appendix B.2.

Since `Qwen2.5-3B-Instruct` may have undergone instruction tuning, these experiments should be viewed as complementary evidence for the qualitative trends predicted by our theory. Appendix B.2 further reports an auxiliary MMLU word-count task, in which the base model performs close to chance in the zero-shot setting and value-matrix fine-tuning slightly outperforms full fine-tuning in the 7-shot setting.

## 6. Conclusion

We investigate the effect of task-specific fine-tuning on the in-context learning capability of Transformers with linear attention. Specifically, we derive closed-form expressions for the parameters that minimize the pretraining and fine-tuning objectives, which reveal how different fine-tuning regimes induce a distinct trade-off between zero-shot and few-shot

performance. Our analysis shows that fine-tuning all parameters to optimize the zero-shot performance can degrade the few-shot performance. In contrast, restricting updates to the value matrix preserves the few-shot performance while improving the zero-shot performance. Furthermore, although incorporating an auxiliary few-shot loss enhances the performance on the target task, it may lead to degradation on out-of-distribution tasks.

**Limitations and Future Work.** Our theoretical analysis relies on linear attention models, which offer analytical tractability and enable closed-form characterization of optimal parameters and test errors, but do not capture all architectural and optimization features of Transformer models. Our pretraining objective is also an analytical surrogate for studying in-context learning, rather than a literal description of real-world language model pretraining. While experiments on the MMLU benchmark indicate that qualitatively similar phenomena may arise in large-scale models, these results should be interpreted as illustrative rather than conclusive. Extending the theory to more realistic Transformer architectures, including softmax attention, multiple layers, and MLP blocks, remains an important direction for future work. Finally, our notion of task similarity is defined through task vectors in the linear regression setting, and extending this notion to real-world tasks is necessary for understanding how these insights generalize to broader parameter-efficient fine-tuning settings.

## Acknowledgements

This work was supported by the National Science Foundation (NSF) Award DMS-2023239, the NSF CAREER Award CCF-2339978, an Amazon Research Award, and a grant from FuriosaAI. In addition, it was supported by the National Research Foundation of Korea (NRF) grant funded by the Korean Ministry of Science and ICT (MSIT) (RS-2024-00345351, RS-2024-00408003), by the Institute of Information & Communications Technology Planning & Evaluation (IITP) grant funded by MSIT (RS-2023-00259934, RS-2025-02283048, RS-2026-25544647), and by the Brain Korea 21 program funded by the Korean Ministry of Education and the NRF (BK21 FOUR, No. 5199990913980).

We thank Jungtaek Kim, Jaden Park, Jaewoo Shin, Thomas Zeng, Yuchen Zeng, and especially Jongwon Jeong for helpful discussions.

## Impact Statement

This paper presents work whose goal is to advance the field of Machine Learning. There are many potential societal consequences of our work, none of which we feel must be specifically highlighted here.

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

# A. Proofs

## A.1. Proofs of Preliminary Lemma

**Lemma A.1** (Theorem 1 in Hagedorn (2022); specialized version of Theorem 2.2.9 (ii) in Kollo & Von Rosen (2005))**.** *Let* $\mathbf{Q} \sim \mathcal{W}(\mathbf{\Sigma}, n)$ *be a* $d \times d$ *random matrix following the Wishart distribution, where* $\mathbf{\Sigma} \in \mathbb{S}_{++}^d$ *is positive definite and* $n \in \mathbb{N}$. *Consider a symmetric matrix* $\mathbf{A} \in \mathbb{R}^{d \times d}$, *then*

$$\mathbb{E}[\mathbf{Q}\mathbf{A}\mathbf{Q}] = n \cdot \mathrm{tr}(\mathbf{A}\mathbf{\Sigma}) \cdot \mathbf{\Sigma} + n(n+1)\mathbf{\Sigma}\mathbf{A}\mathbf{\Sigma}.$$

**Lemma A.2** (Theorem 4.2 in Magnus & Neudecker (1979); see also Tracy & Sultan (1993))**.** *Let* $\mathbf{x} \sim \mathcal{N}(\mathbf{0}_d, \mathbf{\Sigma})$ *be a* $d$-*dimensional random vector following the multivariate normal distribution, where* $\mathbf{\Sigma} \in \mathbb{S}_{++}^d$ *is positive definite. Consider symmetric matrices* $\mathbf{A}, \mathbf{B}, \mathbf{C} \in \mathbb{R}^{d \times d}$, *then*

*(i)* $\mathbb{E}[\mathbf{x}^\top \mathbf{A}\mathbf{x}\mathbf{x}^\top \mathbf{B}\mathbf{x}] = \mathrm{tr}(\mathbf{A}\mathbf{\Sigma})\,\mathrm{tr}(\mathbf{B}\mathbf{\Sigma}) + 2\,\mathrm{tr}(\mathbf{A}\mathbf{\Sigma}\mathbf{B}\mathbf{\Sigma}).$

*(ii)* $\mathbb{E}[\mathbf{x}^\top \mathbf{A}\mathbf{x}\mathbf{x}^\top \mathbf{B}\mathbf{x}\mathbf{x}^\top \mathbf{C}\mathbf{x}] = \mathrm{tr}(\mathbf{A}\mathbf{\Sigma})\,\mathrm{tr}(\mathbf{B}\mathbf{\Sigma})\,\mathrm{tr}(\mathbf{C}\mathbf{\Sigma}) + 2\,\mathrm{tr}(\mathbf{A}\mathbf{\Sigma})\,\mathrm{tr}(\mathbf{B}\mathbf{\Sigma}\mathbf{C}\mathbf{\Sigma})$

$$+2\,\mathrm{tr}(\mathbf{B}\mathbf{\Sigma})\,\mathrm{tr}(\mathbf{A}\mathbf{\Sigma}\mathbf{C}\mathbf{\Sigma}) + 2\,\mathrm{tr}(\mathbf{C}\mathbf{\Sigma})\,\mathrm{tr}(\mathbf{A}\mathbf{\Sigma}\mathbf{B}\mathbf{\Sigma}) + 8\,\mathrm{tr}(\mathbf{A}\mathbf{\Sigma}\mathbf{B}\mathbf{\Sigma}\mathbf{C}\mathbf{\Sigma}).$$

**Lemma A.3.** *Let* $\mathbf{x} \sim \mathcal{N}(\mathbf{0}_d, \mathbf{\Sigma})$ *be a* $d$-*dimensional random vector where* $\mathbf{\Sigma} \in \mathbb{S}_{++}^d$ *is positive definite. Consider matrices* $\mathbf{A}, \mathbf{B} \in \mathbb{R}^{d \times d}$, *then*

$$\mathbb{E}[\mathbf{x}\mathbf{x}^\top \mathbf{A}\mathbf{x}\mathbf{x}^\top \mathbf{B}\mathbf{x}\mathbf{x}^\top] = \mathrm{tr}(\mathbf{A}\mathbf{\Sigma})\,\mathrm{tr}(\mathbf{B}\mathbf{\Sigma}) \cdot \mathbf{\Sigma} + \mathrm{tr}(\mathbf{A}\mathbf{\Sigma}\mathbf{B}\mathbf{\Sigma}) \cdot \mathbf{\Sigma} + \mathrm{tr}(\mathbf{A}\mathbf{\Sigma}\mathbf{B}^\top\mathbf{\Sigma}) \cdot \mathbf{\Sigma} + \mathrm{tr}(\mathbf{A}\mathbf{\Sigma}) \cdot \mathbf{\Sigma}(\mathbf{B}+\mathbf{B}^\top)\mathbf{\Sigma}$$

$$+ \mathrm{tr}(\mathbf{B}\mathbf{\Sigma}) \cdot \mathbf{\Sigma}(\mathbf{A}+\mathbf{A}^\top)\mathbf{\Sigma} + \mathbf{\Sigma}(\mathbf{A}+\mathbf{A}^\top)\mathbf{\Sigma}(\mathbf{B}+\mathbf{B}^\top)\mathbf{\Sigma} + \mathbf{\Sigma}(\mathbf{B}+\mathbf{B}^\top)\mathbf{\Sigma}(\mathbf{A}+\mathbf{A}^\top)\mathbf{\Sigma}.$$

*If* $\mathbf{A} = \mathbf{B}^\top$, *it reduces to*

$$\mathbb{E}[\mathbf{x}\mathbf{x}^\top \mathbf{A}\mathbf{x}\mathbf{x}^\top \mathbf{B}\mathbf{x}\mathbf{x}^\top] = \mathrm{tr}(\mathbf{A}\mathbf{\Sigma})^2 \cdot \mathbf{\Sigma} + \mathrm{tr}(\mathbf{A}\mathbf{\Sigma}\mathbf{A}^\top\mathbf{\Sigma}) \cdot \mathbf{\Sigma} + \mathrm{tr}(\mathbf{A}\mathbf{\Sigma}\mathbf{A}\mathbf{\Sigma}) \cdot \mathbf{\Sigma}$$

$$+ 2\,\mathrm{tr}(\mathbf{A}\mathbf{\Sigma}) \cdot \mathbf{\Sigma}(\mathbf{A}+\mathbf{A}^\top)\mathbf{\Sigma} + 2\mathbf{\Sigma}(\mathbf{A}+\mathbf{A}^\top)\mathbf{\Sigma}(\mathbf{A}+\mathbf{A}^\top)\mathbf{\Sigma}.$$

*Proof.* Denote $x_i := [\mathbf{x}]_i$, $\sigma_{i,j} := [\mathbf{\Sigma}]_{i,j}$, $a_{i,j} := [\mathbf{A}]_{i,j}$ and $b_{i,j} := [\mathbf{B}]_{i,j}$. Then

$$[\mathbf{x}\mathbf{x}^\top \mathbf{A}\mathbf{x}\mathbf{x}^\top \mathbf{B}\mathbf{x}\mathbf{x}^\top]_{ij} = \sum_{k\in[d]}\sum_{l\in[d]}\sum_{m\in[d]}\sum_{n\in[d]} a_{kl}b_{mn} \cdot x_i x_k x_l x_m x_n x_j.$$

By Isserlis (1918) theorem, for $i, j, k, l, m \in [d]$, we have

$$\mathbb{E}[x_i x_j x_k x_l x_m x_n] = \sigma_{ij}\sigma_{kl}\sigma_{mn} + \sigma_{ij}\sigma_{km}\sigma_{nl} + \sigma_{ij}\sigma_{kn}\sigma_{ml} + \sigma_{ik}\sigma_{lj}\sigma_{mn} + \sigma_{ik}\sigma_{lm}\sigma_{nj} + \sigma_{ik}\sigma_{ln}\sigma_{mj}$$

$$+ \sigma_{il}\sigma_{kj}\sigma_{mn} + \sigma_{il}\sigma_{km}\sigma_{nj} + \sigma_{il}\sigma_{kn}\sigma_{mj} + \sigma_{im}\sigma_{nj}\sigma_{kl} + \sigma_{im}\sigma_{nk}\sigma_{lj} + \sigma_{im}\sigma_{nl}\sigma_{kj}$$

$$+ \sigma_{in}\sigma_{mj}\sigma_{kl} + \sigma_{in}\sigma_{mk}\sigma_{lj} + \sigma_{in}\sigma_{ml}\sigma_{kj}.$$

We further denote $a_{i,j}^\top := [\mathbf{A}^\top]_{i,j}$ and $b_{i,j}^\top := [\mathbf{B}^\top]_{i,j}$, which implies $a_{i,j} = a_{j,i}^\top$ and $b_{i,j} = b_{j,i}^\top$, then

$$a_{kl}b_{mn} \cdot \mathbb{E}[x_i x_j x_k x_l x_m x_n] = \sigma_{ij} \cdot \sigma_{kl}a_{lk}^\top \cdot \sigma_{mn}b_{nm}^\top + \sigma_{ij}\left(\sigma_{km}b_{mn}\sigma_{nl}a_{lk}^\top + \sigma_{kn}b_{nm}^\top\sigma_{ml}a_{lk}^\top\right)$$

$$+ \sigma_{ik}a_{kl}\sigma_{lj} \cdot \sigma_{mn}b_{nm}^\top + \sigma_{ik}a_{kl}\sigma_{lm}b_{mn}\sigma_{nj} + \sigma_{ik}a_{kl}\sigma_{ln}b_{nm}^\top\sigma_{mj}$$

$$+ \sigma_{il}a_{lk}^\top\sigma_{kj} \cdot \sigma_{mn}b_{nm}^\top + \sigma_{il}a_{lk}^\top\sigma_{km}b_{mn}\sigma_{nj} + \sigma_{il}a_{lk}^\top\sigma_{kn}b_{nm}^\top\sigma_{mj}$$

$$+ \sigma_{im}b_{mn}\sigma_{nj} \cdot \sigma_{kl}a_{lk}^\top + \sigma_{im}b_{mn}\sigma_{nk}a_{kl}\sigma_{lj} + \sigma_{im}b_{mn}\sigma_{nl}a_{lk}^\top\sigma_{kj}$$

$$+ \sigma_{in}b_{nm}^\top\sigma_{mj} \cdot \sigma_{kl}a_{lk}^\top + \sigma_{in}b_{nm}^\top\sigma_{mk}a_{kl}\sigma_{lj} + \sigma_{in}b_{nm}^\top\sigma_{ml}a_{lk}^\top\sigma_{kj}.$$

Therefore,

$$
\begin{aligned}
&\mathbb{E}[\mathbf{x}\mathbf{x}^\top\mathbf{A}\mathbf{x}\mathbf{x}^\top\mathbf{B}\mathbf{x}\mathbf{x}^\top]_{ij}\\
&= \sum_{k\in[d]}\sum_{l\in[d]}\sum_{m\in[d]}\sum_{n\in[d]} a_{kl}b_{mn}\cdot\mathbb{E}[x_ix_kx_lx_mx_nx_j]\\
&= \sigma_{ij}\cdot\mathrm{tr}(\mathbf{\Sigma}\mathbf{A}^\top)\,\mathrm{tr}(\mathbf{\Sigma}\mathbf{B}^\top) + \sigma_{ij}\cdot\big(\mathrm{tr}(\mathbf{\Sigma}\mathbf{B}\mathbf{\Sigma}\mathbf{A}^\top) + \mathrm{tr}(\mathbf{\Sigma}\mathbf{B}^\top\mathbf{\Sigma}\mathbf{A}^\top)\big) + \sum_{k\in[d]}\sum_{l\in[d]} \sigma_{ik}a_{kl}\sigma_{lj}\cdot\mathrm{tr}(\mathbf{\Sigma}\mathbf{B}^\top)\\
&\quad + \sum_{k\in[d]}\sum_{l\in[d]}\sum_{m\in[d]}\sum_{n\in[d]} \big(\sigma_{ik}a_{kl}\sigma_{lm}b_{mn}\sigma_{nj} + \sigma_{ik}a_{kl}\sigma_{ln}b_{nm}^\top\sigma_{mj}\big) + \sum_{k\in[d]}\sum_{l\in[d]} \sigma_{il}a_{lk}^\top\sigma_{kj}\cdot\mathrm{tr}(\mathbf{\Sigma}\mathbf{B}^\top)\\
&\quad + \sum_{k\in[d]}\sum_{l\in[d]}\sum_{m\in[d]}\sum_{n\in[d]} \big(\sigma_{il}a_{lk}^\top\sigma_{km}b_{mn}\sigma_{nj} + \sigma_{il}a_{lk}^\top\sigma_{kn}b_{nm}^\top\sigma_{mj}\big) + \sum_{m\in[d]}\sum_{n\in[d]} \sigma_{im}b_{mn}\sigma_{nj}\cdot\mathrm{tr}(\mathbf{\Sigma}\mathbf{A}^\top)\\
&\quad + \sum_{k\in[d]}\sum_{l\in[d]}\sum_{m\in[d]}\sum_{n\in[d]} \big(\sigma_{im}b_{mn}\sigma_{nk}a_{kl}\sigma_{lj} + \sigma_{im}b_{mn}\sigma_{nl}a_{lk}^\top\sigma_{kj}\big) + \sum_{m\in[d]}\sum_{n\in[d]} \sigma_{in}b_{nm}^\top\sigma_{mj}\cdot\mathrm{tr}(\mathbf{\Sigma}\mathbf{A}^\top)\\
&\quad + \sum_{k\in[d]}\sum_{l\in[d]}\sum_{m\in[d]}\sum_{n\in[d]} \big(\sigma_{in}b_{nm}^\top\sigma_{mk}a_{kl}\sigma_{lj} + \sigma_{in}b_{nm}^\top\sigma_{ml}a_{lk}^\top\sigma_{kj}\big),
\end{aligned}
$$

which implies

$$
\begin{aligned}
&\mathbb{E}[\mathbf{x}\mathbf{x}^\top\mathbf{A}\mathbf{x}\mathbf{x}^\top\mathbf{B}\mathbf{x}\mathbf{x}^\top]\\
&= \mathbf{\Sigma}\cdot\mathrm{tr}(\mathbf{\Sigma}\mathbf{A}^\top)\,\mathrm{tr}(\mathbf{\Sigma}\mathbf{B}^\top) + \mathbf{\Sigma}\cdot\mathrm{tr}(\mathbf{\Sigma}\mathbf{B}\mathbf{\Sigma}\mathbf{A}^\top) + \mathbf{\Sigma}\cdot\mathrm{tr}(\mathbf{\Sigma}\mathbf{B}^\top\mathbf{\Sigma}\mathbf{A}^\top)\\
&\quad + \mathbf{\Sigma}\mathbf{A}\mathbf{\Sigma}\cdot\mathrm{tr}(\mathbf{\Sigma}\mathbf{B}^\top) + \mathbf{\Sigma}\mathbf{A}\mathbf{\Sigma}\mathbf{B}\mathbf{\Sigma} + \mathbf{\Sigma}\mathbf{A}\mathbf{\Sigma}\mathbf{B}^\top\mathbf{\Sigma} + \mathbf{\Sigma}\mathbf{A}^\top\mathbf{\Sigma}\cdot\mathrm{tr}(\mathbf{\Sigma}\mathbf{B}^\top) + \mathbf{\Sigma}\mathbf{A}^\top\mathbf{\Sigma}\mathbf{B}\mathbf{\Sigma} + \mathbf{\Sigma}\mathbf{A}^\top\mathbf{\Sigma}\mathbf{B}^\top\mathbf{\Sigma}\\
&\quad + \mathbf{\Sigma}\mathbf{B}\mathbf{\Sigma}\cdot\mathrm{tr}(\mathbf{\Sigma}\mathbf{A}^\top) + \mathbf{\Sigma}\mathbf{B}\mathbf{\Sigma}\mathbf{A}\mathbf{\Sigma} + \mathbf{\Sigma}\mathbf{B}\mathbf{\Sigma}\mathbf{A}^\top\mathbf{\Sigma} + \mathbf{\Sigma}\mathbf{B}^\top\mathbf{\Sigma}\cdot\mathrm{tr}(\mathbf{\Sigma}\mathbf{A}^\top) + \mathbf{\Sigma}\mathbf{B}^\top\mathbf{\Sigma}\mathbf{A}\mathbf{\Sigma} + \mathbf{\Sigma}\mathbf{B}^\top\mathbf{\Sigma}\mathbf{A}^\top\mathbf{\Sigma}\\
&= \mathrm{tr}(\mathbf{A}\mathbf{\Sigma})\,\mathrm{tr}(\mathbf{B}\mathbf{\Sigma})\cdot\mathbf{\Sigma} + \mathrm{tr}(\mathbf{A}\mathbf{\Sigma}\mathbf{B}\mathbf{\Sigma})\cdot\mathbf{\Sigma} + \mathrm{tr}(\mathbf{A}\mathbf{\Sigma}\mathbf{B}^\top\mathbf{\Sigma})\cdot\mathbf{\Sigma} + \mathrm{tr}(\mathbf{A}\mathbf{\Sigma})\cdot\mathbf{\Sigma}(\mathbf{B}+\mathbf{B}^\top)\mathbf{\Sigma}\\
&\quad + \mathrm{tr}(\mathbf{B}\mathbf{\Sigma})\cdot\mathbf{\Sigma}(\mathbf{A}+\mathbf{A}^\top)\mathbf{\Sigma} + \mathbf{\Sigma}(\mathbf{A}+\mathbf{A}^\top)\mathbf{\Sigma}(\mathbf{B}+\mathbf{B}^\top)\mathbf{\Sigma} + \mathbf{\Sigma}(\mathbf{B}+\mathbf{B}^\top)\mathbf{\Sigma}(\mathbf{A}+\mathbf{A}^\top)\mathbf{\Sigma}.
\end{aligned}
$$

Moreover, if $\mathbf{A} = \mathbf{B}^\top$, we have

$$
\begin{aligned}
&\mathbb{E}[\mathbf{x}\mathbf{x}^\top\mathbf{A}\mathbf{x}\mathbf{x}^\top\mathbf{B}\mathbf{x}\mathbf{x}^\top]\\
&= \mathrm{tr}(\mathbf{A}\mathbf{\Sigma})\,\mathrm{tr}(\mathbf{A}^\top\mathbf{\Sigma})\cdot\mathbf{\Sigma} + \mathrm{tr}(\mathbf{A}\mathbf{\Sigma}\mathbf{A}^\top\mathbf{\Sigma})\cdot\mathbf{\Sigma} + \mathrm{tr}(\mathbf{A}\mathbf{\Sigma}\mathbf{A}\mathbf{\Sigma})\cdot\mathbf{\Sigma} + \mathrm{tr}(\mathbf{A}\mathbf{\Sigma})\cdot\mathbf{\Sigma}(\mathbf{A}+\mathbf{A}^\top)\mathbf{\Sigma}\\
&\quad + \mathrm{tr}(\mathbf{A}^\top\mathbf{\Sigma})\cdot\mathbf{\Sigma}(\mathbf{A}+\mathbf{A}^\top)\mathbf{\Sigma} + 2\mathbf{\Sigma}(\mathbf{A}+\mathbf{A}^\top)\mathbf{\Sigma}(\mathbf{A}+\mathbf{A}^\top)\mathbf{\Sigma}\\
&= \mathrm{tr}(\mathbf{A}\mathbf{\Sigma})^2\cdot\mathbf{\Sigma} + \mathrm{tr}(\mathbf{A}\mathbf{\Sigma}\mathbf{A}^\top\mathbf{\Sigma})\cdot\mathbf{\Sigma} + \mathrm{tr}(\mathbf{A}\mathbf{\Sigma}\mathbf{A}\mathbf{\Sigma})\cdot\mathbf{\Sigma} + 2\,\mathrm{tr}(\mathbf{A}\mathbf{\Sigma})\cdot\mathbf{\Sigma}(\mathbf{A}+\mathbf{A}^\top)\mathbf{\Sigma} + 2\mathbf{\Sigma}(\mathbf{A}+\mathbf{A}^\top)\mathbf{\Sigma}(\mathbf{A}+\mathbf{A}^\top)\mathbf{\Sigma},
\end{aligned}
$$

where the last equality follows from the fact that $\mathrm{tr}(\mathbf{A}\mathbf{\Sigma}) = \mathrm{tr}(\mathbf{\Sigma}\mathbf{A}^\top) = \mathrm{tr}(\mathbf{A}^\top\mathbf{\Sigma})$. One can confirm that this result for $\mathbf{A} = \mathbf{B} = \mathbf{I}$ is consistent with Lemma A.2 in the special case where $\mathbf{A} = \mathbf{B} = \mathbf{C} = \mathbf{I}$. $\qquad\square$

## A.2. Proofs of Statistical Properties of Model Outputs

Given the task vector $\theta_0 \in \mathbb{R}^d$, the test data point $(\mathbf{x}_i, y_i)$ for $i \in [n+1]$ is independently generated as

$$\mathbf{x}_i \overset{\text{iid}}{\sim} \mathcal{N}(\mathbf{0}_d, \mathbf{\Sigma}), \qquad e_i \overset{\text{iid}}{\sim} \mathcal{N}(0, \sigma^2), \qquad y_i = \theta_0^\top \mathbf{x}_i + e_i.$$

It follows that for each $i \in [n+1]$, we have $y_i \overset{\text{iid}}{\sim} \mathcal{N}(0, \theta_0^\top \mathbf{\Sigma} \theta_0 + \sigma^2)$ and thus

$$\begin{bmatrix} \mathbf{x}_i \\ y_i \end{bmatrix} \overset{\text{iid}}{\sim} \mathcal{N}\left(\mathbf{0}_{d+1}, \mathbf{\Sigma}_{\text{joint}}\right), \qquad \mathbf{\Sigma}_{\text{joint}} := \begin{bmatrix} \mathbf{\Sigma} & \mathbf{\Sigma}\theta_0 \\ \theta_0^\top \mathbf{\Sigma} & \theta_0^\top \mathbf{\Sigma}\theta_0 + \sigma^2 \end{bmatrix}.$$

Based on these distributions, we have to determine the statistics of the predictions, which are defined as

$$\hat{y}_{\text{FS}}(\mathbf{Z}_{[n]}, \mathbf{x}_{n+1}; \mathbf{V}, \mathbf{Q}) := \frac{n}{n+1} \cdot \hat{y}_{\text{Net-FS}}(\mathbf{Z}_{[n]}, \mathbf{x}_{n+1}; \mathbf{V}, \mathbf{Q}) + \frac{1}{n+1} \cdot \hat{y}_{\text{ZS}}(\mathbf{x}_{n+1}; \mathbf{V}, \mathbf{Q}),$$

where

$$\hat{y}_{\text{Net-FS}}(\mathbf{Z}_{[n]}, \mathbf{x}_{n+1}; \mathbf{V}, \mathbf{Q}) := \begin{bmatrix} \mathbf{v}_{21} \\ v_{22} \end{bmatrix}^\top \left( \frac{1}{n} \sum_{i \in [n]} \begin{bmatrix} \mathbf{x}_i \\ y_i \end{bmatrix} \begin{bmatrix} \mathbf{x}_i \\ y_i \end{bmatrix}^\top \right) \begin{bmatrix} \mathbf{Q}_{11} \\ \mathbf{q}_{21}^\top \end{bmatrix} \mathbf{x}_{n+1} + \begin{bmatrix} \mathbf{v}_{21} \\ v_{22} \end{bmatrix}^\top \left( \frac{1}{n} \sum_{i \in [n]} \begin{bmatrix} \mathbf{x}_i \\ y_i \end{bmatrix} \right) \cdot q, \quad (16)$$

$$\hat{y}_{\text{ZS}}(\mathbf{x}_{n+1}; \mathbf{V}, \mathbf{Q}) := \mathbf{v}_{21}^\top \mathbf{x}_{n+1} \mathbf{x}_{n+1}^\top \mathbf{Q}_{11} \mathbf{x}_{n+1} + \mathbf{v}_{21}^\top \mathbf{x}_{n+1} \cdot q. \quad (17)$$

**Lemma A.4.** *Given $\mathbf{V}$, $\mathbf{Q}$, and $\theta_0 \in \mathbb{R}^d$, (co)variances of $\hat{y}_{\text{Net-FS}}(\mathbf{Z}_{[n]}, \mathbf{x}_{n+1}; \mathbf{V}, \mathbf{Q})$ in (16), $\hat{y}_{\text{ZS}}(\mathbf{x}_{n+1}; \mathbf{V}, \mathbf{Q})$ in (17), and $y_{n+1}$ are as follows:*

$$\text{Var}\left[\hat{y}_{\text{Net-FS}}(\mathbf{Z}_{[n]}, \mathbf{x}_{n+1}; \mathbf{V}, \mathbf{Q})\right] = \frac{1}{n} \cdot \text{tr}\left( \begin{bmatrix} \mathbf{Q}_{11} \\ \mathbf{q}_{21}^\top \end{bmatrix} \mathbf{\Sigma} \begin{bmatrix} \mathbf{Q}_{11} \\ \mathbf{q}_{21}^\top \end{bmatrix}^\top \mathbf{\Sigma}_{\text{joint}} \right) \cdot \begin{bmatrix} \mathbf{v}_{21} \\ v_{22} \end{bmatrix}^\top \mathbf{\Sigma}_{\text{joint}} \begin{bmatrix} \mathbf{v}_{21} \\ v_{22} \end{bmatrix}$$

$$+ \frac{n+1}{n} \cdot \begin{bmatrix} \mathbf{v}_{21} \\ v_{22} \end{bmatrix}^\top \mathbf{\Sigma}_{\text{joint}} \begin{bmatrix} \mathbf{Q}_{11} \\ \mathbf{q}_{21}^\top \end{bmatrix} \mathbf{\Sigma} \begin{bmatrix} \mathbf{Q}_{11} \\ \mathbf{q}_{21}^\top \end{bmatrix}^\top \mathbf{\Sigma}_{\text{joint}} \begin{bmatrix} \mathbf{v}_{21} \\ v_{22} \end{bmatrix} + \frac{q^2}{n} \cdot \begin{bmatrix} \mathbf{v}_{21} \\ v_{22} \end{bmatrix}^\top \mathbf{\Sigma}_{\text{joint}} \begin{bmatrix} \mathbf{v}_{21} \\ v_{22} \end{bmatrix},$$

$$\text{Var}\left[\hat{y}_{\text{ZS}}(\mathbf{x}_{n+1}; \mathbf{V}, \mathbf{Q})\right] = \left( \text{tr}(\mathbf{Q}_{11}\mathbf{\Sigma})^2 + \text{tr}(\mathbf{Q}_{11}\mathbf{\Sigma}\mathbf{Q}_{11}^\top\mathbf{\Sigma}) + \text{tr}(\mathbf{Q}_{11}\mathbf{\Sigma}\mathbf{Q}_{11}\mathbf{\Sigma}) \right) \cdot \mathbf{v}_{21}^\top \mathbf{\Sigma} \mathbf{v}_{21}$$

$$+ 4\,\text{tr}(\mathbf{Q}_{11}\mathbf{\Sigma}) \cdot \mathbf{v}_{21}^\top \mathbf{\Sigma}\mathbf{Q}_{11}\mathbf{\Sigma}\mathbf{v}_{21} + 2\mathbf{v}_{21}^\top \mathbf{\Sigma}(\mathbf{Q}_{11} + \mathbf{Q}_{11}^\top)\mathbf{\Sigma}(\mathbf{Q}_{11} + \mathbf{Q}_{11}^\top)\mathbf{\Sigma}\mathbf{v}_{21}$$

$$+ 2q \cdot \text{tr}(\mathbf{Q}_{11}\mathbf{\Sigma}) \cdot \mathbf{v}_{21}^\top \mathbf{\Sigma}\mathbf{v}_{21} + 4q \cdot \mathbf{v}_{21}^\top \mathbf{\Sigma}\mathbf{Q}_{11}\mathbf{\Sigma}\mathbf{v}_{21} + q^2 \cdot \mathbf{v}_{21}^\top \mathbf{\Sigma}\mathbf{v}_{21},$$

$$\text{Var}\left[y_{n+1}\right] = \theta_0^\top \mathbf{\Sigma}\theta_0 + \sigma^2,$$

$$\text{Cov}\left[\hat{y}_{\text{Net-FS}}(\mathbf{Z}_{[n]}, \mathbf{x}_{n+1}; \mathbf{V}, \mathbf{Q}), \hat{y}_{\text{ZS}}(\mathbf{x}_{n+1}; \mathbf{V}, \mathbf{Q})\right] = \begin{bmatrix} \mathbf{v}_{21} \\ v_{22} \end{bmatrix}^\top \mathbf{\Sigma}_{\text{joint}} \begin{bmatrix} \mathbf{Q}_{11} \\ \mathbf{q}_{21}^\top \end{bmatrix} \mathbf{\Sigma} \left( (\mathbf{Q}_{11} + \mathbf{Q}_{11}^\top)\mathbf{\Sigma} + \text{tr}(\mathbf{Q}_{11}\mathbf{\Sigma})\mathbf{I}_d + q \cdot \mathbf{I}_d \right) \mathbf{v}_{21},$$

$$\text{Cov}\left[\hat{y}_{\text{Net-FS}}(\mathbf{Z}_{[n]}, \mathbf{x}_{n+1}; \mathbf{V}, \mathbf{Q}), y_{n+1}\right] = \begin{bmatrix} \mathbf{v}_{21} \\ v_{22} \end{bmatrix}^\top \mathbf{\Sigma}_{\text{joint}} \begin{bmatrix} \mathbf{Q}_{11} \\ \mathbf{q}_{21}^\top \end{bmatrix} \mathbf{\Sigma}\theta_0,$$

$$\text{Cov}\left[\hat{y}_{\text{ZS}}(\mathbf{x}_{n+1}; \mathbf{V}, \mathbf{Q}), y_{n+1}\right] = \mathbf{v}_{21}^\top \left( \mathbf{\Sigma}(\mathbf{Q}_{11} + \mathbf{Q}_{11}^\top)\mathbf{\Sigma} + \text{tr}(\mathbf{Q}_{11}\mathbf{\Sigma}) \cdot \mathbf{\Sigma} + q \cdot \mathbf{\Sigma} \right) \theta_0.$$

*Proof.* First, note that $\hat{y}_{\text{Net-FS}}(\mathbf{Z}_{[n]}, \mathbf{x}_{n+1}; \mathbf{V}, \mathbf{Q})$, $\hat{y}_{\text{ZS}}(\mathbf{x}_{n+1}; \mathbf{V}, \mathbf{Q})$, and $y_{n+1}$ are linear or cubic function of the zero-mean Gaussian variable $\mathbf{x}_{n+1}$. By Isserlis (1918) theorem, we have

$$\mathbb{E}[\hat{y}_{\text{Net-FS}}(\mathbf{Z}_{[n]}, \mathbf{x}_{n+1}; \mathbf{V}, \mathbf{Q})] = \mathbb{E}[\hat{y}_{\text{ZS}}(\mathbf{x}_{n+1}; \mathbf{V}, \mathbf{Q})] = \mathbb{E}[y_{n+1}] = 0.$$

Therefore, it suffices to determine their second moments, as they coincide with the (co)variances.

From the definition of $\hat{y}_{\text{Net-FS}}(\mathbf{Z}_{[n]}, \mathbf{x}_{n+1}; \mathbf{V}, \mathbf{Q})$ in (16), we have

$$
\begin{aligned}
\text{Var}\left[\hat{y}_{\text{Net-FS}}(\mathbf{Z}_{[n]}, \mathbf{x}_{n+1}; \mathbf{V}, \mathbf{Q})\right] &= \mathbb{E}\left[\hat{y}_{\text{Net-FS}}(\mathbf{Z}_{[n]}, \mathbf{x}_{n+1}; \mathbf{V}, \mathbf{Q})^2\right] \\
&= \mathbb{E}\left[\begin{bmatrix}\mathbf{v}_{21}\\v_{22}\end{bmatrix}^\top \left(\frac{1}{n}\sum_{i\in[n]}\begin{bmatrix}\mathbf{x}_i\\y_i\end{bmatrix}\begin{bmatrix}\mathbf{x}_i\\y_i\end{bmatrix}^\top\right)\begin{bmatrix}\mathbf{Q}_{11}\\\mathbf{q}_{21}^\top\end{bmatrix}\boldsymbol{\Sigma}\begin{bmatrix}\mathbf{Q}_{11}\\\mathbf{q}_{21}^\top\end{bmatrix}^\top\left(\frac{1}{n}\sum_{i\in[n]}\begin{bmatrix}\mathbf{x}_i\\y_i\end{bmatrix}\begin{bmatrix}\mathbf{x}_i\\y_i\end{bmatrix}^\top\right)\begin{bmatrix}\mathbf{v}_{21}\\v_{22}\end{bmatrix}\right] \\
&\quad + \mathbb{E}\left[\begin{bmatrix}\mathbf{v}_{21}\\v_{22}\end{bmatrix}^\top\left(\frac{1}{n}\sum_{i\in[n]}\begin{bmatrix}\mathbf{x}_i\\y_i\end{bmatrix}\right)\left(\frac{1}{n}\sum_{i\in[n]}\begin{bmatrix}\mathbf{x}_i\\y_i\end{bmatrix}^\top\right)\begin{bmatrix}\mathbf{v}_{21}\\v_{22}\end{bmatrix}\cdot q^2\right] \\
&= \frac{1}{n^2}\begin{bmatrix}\mathbf{v}_{21}\\v_{22}\end{bmatrix}^\top\mathbb{E}\left[\sum_{i\in[n]}\begin{bmatrix}\mathbf{x}_i\\y_i\end{bmatrix}\begin{bmatrix}\mathbf{x}_i\\y_i\end{bmatrix}^\top\begin{bmatrix}\mathbf{Q}_{11}\\\mathbf{q}_{21}^\top\end{bmatrix}\boldsymbol{\Sigma}\begin{bmatrix}\mathbf{Q}_{11}\\\mathbf{q}_{21}^\top\end{bmatrix}^\top\sum_{i\in[n]}\begin{bmatrix}\mathbf{x}_i\\y_i\end{bmatrix}\begin{bmatrix}\mathbf{x}_i\\y_i\end{bmatrix}^\top\right]\begin{bmatrix}\mathbf{v}_{21}\\v_{22}\end{bmatrix} \\
&\quad + \frac{q^2}{n^2}\begin{bmatrix}\mathbf{v}_{21}\\v_{22}\end{bmatrix}^\top\mathbb{E}\left[\left(\sum_{i\in[n]}\begin{bmatrix}\mathbf{x}_i\\y_i\end{bmatrix}\right)\left(\sum_{i\in[n]}\begin{bmatrix}\mathbf{x}_i\\y_i\end{bmatrix}^\top\right)\right]\begin{bmatrix}\mathbf{v}_{21}\\v_{22}\end{bmatrix},
\end{aligned}
\tag{18}
$$

where (18) holds from Isserlis (1918) theorem.

Note that $\sum_{i\in[n]}\begin{bmatrix}\mathbf{x}_i\\y_i\end{bmatrix}\begin{bmatrix}\mathbf{x}_i\\y_i\end{bmatrix}^\top$ follows the Wishart distribution $\mathcal{W}(\boldsymbol{\Sigma}_{\text{joint}}, n)$ because $\begin{bmatrix}\mathbf{x}_i\\y_i\end{bmatrix} \overset{\text{iid}}{\sim} \mathcal{N}\left(\mathbf{0}_{d+1}, \boldsymbol{\Sigma}_{\text{joint}}\right)$ for all $i \in [n]$. By using Lemma A.1, we have

$$
\begin{aligned}
&\mathbb{E}\left[\sum_{i\in[n]}\begin{bmatrix}\mathbf{x}_i\\y_i\end{bmatrix}\begin{bmatrix}\mathbf{x}_i\\y_i\end{bmatrix}^\top\begin{bmatrix}\mathbf{Q}_{11}\\\mathbf{q}_{21}^\top\end{bmatrix}\boldsymbol{\Sigma}\begin{bmatrix}\mathbf{Q}_{11}\\\mathbf{q}_{21}^\top\end{bmatrix}^\top\sum_{i\in[n]}\begin{bmatrix}\mathbf{x}_i\\y_i\end{bmatrix}\begin{bmatrix}\mathbf{x}_i\\y_i\end{bmatrix}^\top\right] \\
&= n \cdot \text{tr}\left(\begin{bmatrix}\mathbf{Q}_{11}\\\mathbf{q}_{21}^\top\end{bmatrix}\boldsymbol{\Sigma}\begin{bmatrix}\mathbf{Q}_{11}\\\mathbf{q}_{21}^\top\end{bmatrix}^\top\boldsymbol{\Sigma}_{\text{joint}}\right)\cdot\boldsymbol{\Sigma}_{\text{joint}} + n(n+1)\boldsymbol{\Sigma}_{\text{joint}}\begin{bmatrix}\mathbf{Q}_{11}\\\mathbf{q}_{21}^\top\end{bmatrix}\boldsymbol{\Sigma}\begin{bmatrix}\mathbf{Q}_{11}\\\mathbf{q}_{21}^\top\end{bmatrix}^\top\boldsymbol{\Sigma}_{\text{joint}}.
\end{aligned}
$$

Therefore, we have

$$
\begin{aligned}
&\text{Var}\left[\hat{y}_{\text{Net-FS}}(\mathbf{Z}_{[n]}, \mathbf{x}_{n+1}; \mathbf{V}, \mathbf{Q})\right] \\
&= \frac{1}{n}\cdot\text{tr}\left(\begin{bmatrix}\mathbf{Q}_{11}\\\mathbf{q}_{21}^\top\end{bmatrix}\boldsymbol{\Sigma}\begin{bmatrix}\mathbf{Q}_{11}\\\mathbf{q}_{21}^\top\end{bmatrix}^\top\boldsymbol{\Sigma}_{\text{joint}}\right)\cdot\begin{bmatrix}\mathbf{v}_{21}\\v_{22}\end{bmatrix}^\top\boldsymbol{\Sigma}_{\text{joint}}\begin{bmatrix}\mathbf{v}_{21}\\v_{22}\end{bmatrix} + \frac{n+1}{n}\cdot\begin{bmatrix}\mathbf{v}_{21}\\v_{22}\end{bmatrix}^\top\boldsymbol{\Sigma}_{\text{joint}}\begin{bmatrix}\mathbf{Q}_{11}\\\mathbf{q}_{21}^\top\end{bmatrix}\boldsymbol{\Sigma}\begin{bmatrix}\mathbf{Q}_{11}\\\mathbf{q}_{21}^\top\end{bmatrix}^\top\boldsymbol{\Sigma}_{\text{joint}}\begin{bmatrix}\mathbf{v}_{21}\\v_{22}\end{bmatrix} \\
&\quad + \frac{q^2}{n}\cdot\begin{bmatrix}\mathbf{v}_{21}\\v_{22}\end{bmatrix}^\top\boldsymbol{\Sigma}_{\text{joint}}\begin{bmatrix}\mathbf{v}_{21}\\v_{22}\end{bmatrix},
\end{aligned}
$$

where the last term comes from $\sum_{i\in[n]}\begin{bmatrix}\mathbf{x}_i\\y_i\end{bmatrix} \sim \mathcal{N}\left(\mathbf{0}_{d+1}, n\boldsymbol{\Sigma}_{\text{joint}}\right)$.

On the other hand, from the definition of $\hat{y}_{\text{ZS}}(\mathbf{x}_{n+1}; \mathbf{V}, \mathbf{Q})$ in (17), we obtain

$$
\begin{aligned}
&\text{Var}\left[\hat{y}_{\text{ZS}}(\mathbf{x}_{n+1}; \mathbf{V}, \mathbf{Q})\right] \\
&= \mathbb{E}\left[\hat{y}_{\text{ZS}}(\mathbf{x}_{n+1}; \mathbf{V}, \mathbf{Q})^2\right] \\
&= \mathbb{E}\left[\mathbf{v}_{21}^\top\mathbf{x}_{n+1}\mathbf{x}_{n+1}^\top\mathbf{Q}_{11}\mathbf{x}_{n+1}\mathbf{x}_{n+1}^\top\mathbf{Q}_{11}^\top\mathbf{x}_{n+1}\mathbf{x}_{n+1}^\top\mathbf{v}_{21}\right] + 2q\cdot\mathbb{E}\left[\mathbf{x}_{n+1}^\top\mathbf{v}_{21}\mathbf{v}_{21}^\top\mathbf{x}_{n+1}\mathbf{x}_{n+1}^\top\mathbf{Q}_{11}\mathbf{x}_{n+1}\right] + q^2\cdot\mathbb{E}\left[\mathbf{v}_{21}^\top\mathbf{x}_{n+1}\mathbf{x}_{n+1}^\top\mathbf{v}_{21}\right] \\
&= \mathbf{v}_{21}^\top\mathbb{E}\left[\mathbf{x}_{n+1}\mathbf{x}_{n+1}^\top\mathbf{Q}_{11}\mathbf{x}_{n+1}\mathbf{x}_{n+1}^\top\mathbf{Q}_{11}^\top\mathbf{x}_{n+1}\mathbf{x}_{n+1}^\top\right]\mathbf{v}_{21} + q\cdot\mathbb{E}\left[\mathbf{x}_{n+1}^\top\mathbf{v}_{21}\mathbf{v}_{21}^\top\mathbf{x}_{n+1}\mathbf{x}_{n+1}^\top(\mathbf{Q}_{11} + \mathbf{Q}_{11}^\top)\mathbf{x}_{n+1}\right] + q^2\cdot\mathbf{v}_{21}^\top\boldsymbol{\Sigma}\mathbf{v}_{21}.
\end{aligned}
$$

By using Lemma A.3,

$$\mathbb{E}\left[\mathbf{x}_{n+1}\mathbf{x}_{n+1}^\top\mathbf{Q}_{11}\mathbf{x}_{n+1}\mathbf{x}_{n+1}^\top\mathbf{Q}_{11}^\top\mathbf{x}_{n+1}\mathbf{x}_{n+1}^\top\right] = \operatorname{tr}(\mathbf{Q}_{11}\boldsymbol{\Sigma})^2 \cdot \boldsymbol{\Sigma} + \operatorname{tr}(\mathbf{Q}_{11}\boldsymbol{\Sigma}\mathbf{Q}_{11}^\top\boldsymbol{\Sigma}) \cdot \boldsymbol{\Sigma} + \operatorname{tr}(\mathbf{Q}_{11}\boldsymbol{\Sigma}\mathbf{Q}_{11}\boldsymbol{\Sigma}) \cdot \boldsymbol{\Sigma}$$
$$+ 2\operatorname{tr}(\mathbf{Q}_{11}\boldsymbol{\Sigma}) \cdot \boldsymbol{\Sigma}(\mathbf{Q}_{11} + \mathbf{Q}_{11}^\top)\boldsymbol{\Sigma} + 2\boldsymbol{\Sigma}(\mathbf{Q}_{11} + \mathbf{Q}_{11}^\top)\boldsymbol{\Sigma}(\mathbf{Q}_{11} + \mathbf{Q}_{11}^\top)\boldsymbol{\Sigma},$$

and by using Lemma A.2,

$$\mathbb{E}\left[\mathbf{x}_{n+1}^\top\mathbf{v}_{21}\mathbf{v}_{21}^\top\mathbf{x}_{n+1}\mathbf{x}_{n+1}^\top(\mathbf{Q}_{11} + \mathbf{Q}_{11}^\top)\mathbf{x}_{n+1}\right] = \operatorname{tr}(\mathbf{v}_{21}\mathbf{v}_{21}^\top\boldsymbol{\Sigma})\operatorname{tr}((\mathbf{Q}_{11} + \mathbf{Q}_{11}^\top)\boldsymbol{\Sigma}) + 2\operatorname{tr}(\mathbf{v}_{21}\mathbf{v}_{21}^\top\boldsymbol{\Sigma}(\mathbf{Q}_{11} + \mathbf{Q}_{11}^\top)\boldsymbol{\Sigma})$$
$$= \operatorname{tr}((\mathbf{Q}_{11} + \mathbf{Q}_{11}^\top)\boldsymbol{\Sigma}) \cdot \mathbf{v}_{21}^\top\boldsymbol{\Sigma}\mathbf{v}_{21} + 2\mathbf{v}_{21}^\top\boldsymbol{\Sigma}(\mathbf{Q}_{11} + \mathbf{Q}_{11}^\top)\boldsymbol{\Sigma}\mathbf{v}_{21}$$
$$= 2\operatorname{tr}(\mathbf{Q}_{11}\boldsymbol{\Sigma}) \cdot \mathbf{v}_{21}^\top\boldsymbol{\Sigma}\mathbf{v}_{21} + 4\mathbf{v}_{21}^\top\boldsymbol{\Sigma}\mathbf{Q}_{11}\boldsymbol{\Sigma}\mathbf{v}_{21},$$

which implies

$$\operatorname{Var}\left[\hat{y}_{\text{ZS}}(\mathbf{x}_{n+1}; \mathbf{V}, \mathbf{Q})\right] = \left(\operatorname{tr}(\mathbf{Q}_{11}\boldsymbol{\Sigma})^2 + \operatorname{tr}(\mathbf{Q}_{11}\boldsymbol{\Sigma}\mathbf{Q}_{11}^\top\boldsymbol{\Sigma}) + \operatorname{tr}(\mathbf{Q}_{11}\boldsymbol{\Sigma}\mathbf{Q}_{11}\boldsymbol{\Sigma})\right) \cdot \mathbf{v}_{21}^\top\boldsymbol{\Sigma}\mathbf{v}_{21}$$
$$+ 4\operatorname{tr}(\mathbf{Q}_{11}\boldsymbol{\Sigma}) \cdot \mathbf{v}_{21}^\top\boldsymbol{\Sigma}\mathbf{Q}_{11}\boldsymbol{\Sigma}\mathbf{v}_{21} + 2\mathbf{v}_{21}^\top\boldsymbol{\Sigma}(\mathbf{Q}_{11} + \mathbf{Q}_{11}^\top)\boldsymbol{\Sigma}(\mathbf{Q}_{11} + \mathbf{Q}_{11}^\top)\boldsymbol{\Sigma}\mathbf{v}_{21}$$
$$+ 2q \cdot \operatorname{tr}(\mathbf{Q}_{11}\boldsymbol{\Sigma}) \cdot \mathbf{v}_{21}^\top\boldsymbol{\Sigma}\mathbf{v}_{21} + 4q \cdot \mathbf{v}_{21}^\top\boldsymbol{\Sigma}\mathbf{Q}_{11}\boldsymbol{\Sigma}\mathbf{v}_{21} + q^2 \cdot \mathbf{v}_{21}^\top\boldsymbol{\Sigma}\mathbf{v}_{21}.$$

In a similar manner, we can determine the covariances as follows.

$$\operatorname{Cov}\left[\hat{y}_{\text{Net-FS}}(\mathbf{Z}_{[n]}, \mathbf{x}_{n+1}; \mathbf{V}, \mathbf{Q}), \hat{y}_{\text{ZS}}(\mathbf{x}_{n+1}; \mathbf{V}, \mathbf{Q})\right]$$

$$= \mathbb{E}\left[\begin{bmatrix}\mathbf{v}_{21}\\v_{22}\end{bmatrix}^\top \left(\frac{1}{n}\sum_{i\in[n]}\begin{bmatrix}\mathbf{x}_i\\y_i\end{bmatrix}\begin{bmatrix}\mathbf{x}_i\\y_i\end{bmatrix}^\top\right)\begin{bmatrix}\mathbf{Q}_{11}\\\mathbf{q}_{21}^\top\end{bmatrix}\mathbf{x}_{n+1}\mathbf{x}_{n+1}^\top\mathbf{Q}_{11}^\top\mathbf{x}_{n+1}\mathbf{x}_{n+1}^\top\mathbf{v}_{21}\right]$$

$$+ \mathbb{E}\left[\begin{bmatrix}\mathbf{v}_{21}\\v_{22}\end{bmatrix}^\top \left(\frac{1}{n}\sum_{i\in[n]}\begin{bmatrix}\mathbf{x}_i\\y_i\end{bmatrix}\begin{bmatrix}\mathbf{x}_i\\y_i\end{bmatrix}^\top\right)\begin{bmatrix}\mathbf{Q}_{11}\\\mathbf{q}_{21}^\top\end{bmatrix}\mathbf{x}_{n+1}\mathbf{x}_{n+1}^\top\mathbf{v}_{21} \cdot q\right]$$

$$= \begin{bmatrix}\mathbf{v}_{21}\\v_{22}\end{bmatrix}^\top \boldsymbol{\Sigma}_{\text{joint}}\begin{bmatrix}\mathbf{Q}_{11}\\\mathbf{q}_{21}^\top\end{bmatrix}\mathbb{E}\left[\mathbf{x}_{n+1}\mathbf{x}_{n+1}^\top\mathbf{Q}_{11}^\top\mathbf{x}_{n+1}\mathbf{x}_{n+1}^\top\right]\mathbf{v}_{21} + \begin{bmatrix}\mathbf{v}_{21}\\v_{22}\end{bmatrix}^\top \boldsymbol{\Sigma}_{\text{joint}}\begin{bmatrix}\mathbf{Q}_{11}\\\mathbf{q}_{21}^\top\end{bmatrix}\boldsymbol{\Sigma}\mathbf{v}_{21} \cdot q$$

$$= \begin{bmatrix}\mathbf{v}_{21}\\v_{22}\end{bmatrix}^\top \boldsymbol{\Sigma}_{\text{joint}}\begin{bmatrix}\mathbf{Q}_{11}\\\mathbf{q}_{21}^\top\end{bmatrix}\boldsymbol{\Sigma}\left((\mathbf{Q}_{11} + \mathbf{Q}_{11}^\top)\boldsymbol{\Sigma} + \operatorname{tr}(\mathbf{Q}_{11}\boldsymbol{\Sigma})\mathbf{I}_d + q \cdot \mathbf{I}_d\right)\mathbf{v}_{21},$$

where the mean of quartic forms comes from $\mathbb{E}[\mathbf{x}\mathbf{x}^\top\mathbf{A}\mathbf{x}\mathbf{x}^\top] = \boldsymbol{\Sigma}(\mathbf{A} + \mathbf{A}^\top)\boldsymbol{\Sigma} + \operatorname{tr}(\mathbf{A}\boldsymbol{\Sigma}) \cdot \boldsymbol{\Sigma}$ for an arbitrary matrix $\mathbf{A}$, see Sec.8.2.4. in Petersen & Pedersen (2012). Moreover, we have

$$\operatorname{Cov}\left[\hat{y}_{\text{Net-FS}}(\mathbf{Z}_{[n]}, \mathbf{x}_{n+1}; \mathbf{V}, \mathbf{Q}), y_{n+1}\right] = \mathbb{E}\left[\begin{bmatrix}\mathbf{v}_{21}\\v_{22}\end{bmatrix}^\top \left(\frac{1}{n}\sum_{i\in[n]}\begin{bmatrix}\mathbf{x}_i\\y_i\end{bmatrix}\begin{bmatrix}\mathbf{x}_i\\y_i\end{bmatrix}^\top\right)\begin{bmatrix}\mathbf{Q}_{11}\\\mathbf{q}_{21}^\top\end{bmatrix}\mathbf{x}_{n+1}(\theta_0^\top\mathbf{x}_{n+1} + e_{n+1})\right]$$

$$= \begin{bmatrix}\mathbf{v}_{21}\\v_{22}\end{bmatrix}^\top \boldsymbol{\Sigma}_{\text{joint}}\begin{bmatrix}\mathbf{Q}_{11}\\\mathbf{q}_{21}^\top\end{bmatrix}\mathbb{E}\left[\mathbf{x}_{n+1}(\theta_0^\top\mathbf{x}_{n+1} + e_{n+1})\right]$$

$$= \begin{bmatrix}\mathbf{v}_{21}\\v_{22}\end{bmatrix}^\top \boldsymbol{\Sigma}_{\text{joint}}\begin{bmatrix}\mathbf{Q}_{11}\\\mathbf{q}_{21}^\top\end{bmatrix}\boldsymbol{\Sigma}\theta_0,$$

$$\operatorname{Cov}\left[\hat{y}_{\text{ZS}}(\mathbf{x}_{n+1}; \mathbf{V}, \mathbf{Q}), y_{n+1}\right] = \mathbb{E}\left[\mathbf{v}_{21}^\top\mathbf{x}_{n+1}\mathbf{x}_{n+1}^\top\mathbf{Q}_{11}\mathbf{x}_{n+1}(\theta_0^\top\mathbf{x}_{n+1} + e_{n+1})\right] + \mathbb{E}\left[\mathbf{v}_{21}^\top\mathbf{x}_{n+1} \cdot q \cdot (\theta_0^\top\mathbf{x}_{n+1} + e_{n+1})\right]$$

$$= \mathbf{v}_{21}^\top\mathbb{E}\left[\mathbf{x}_{n+1}\mathbf{x}_{n+1}^\top\mathbf{Q}_{11}\mathbf{x}_{n+1}\mathbf{x}_{n+1}^\top\right]\theta_0 + q \cdot \mathbf{v}_{21}^\top\mathbb{E}\left[\mathbf{x}_{n+1}\mathbf{x}_{n+1}^\top\right]\theta_0$$
$$= \mathbf{v}_{21}^\top\left(\boldsymbol{\Sigma}(\mathbf{Q}_{11} + \mathbf{Q}_{11}^\top)\boldsymbol{\Sigma} + \operatorname{tr}(\mathbf{Q}_{11}^\top\boldsymbol{\Sigma})\boldsymbol{\Sigma} + q \cdot \boldsymbol{\Sigma}\right)\theta_0.$$

$\square$

**Lemma A.5.** *Given $\mathbf{V}$, $\mathbf{Q}$, and $\theta_0 \in \mathbb{R}^d$, the expected squared errors of $\hat{y}_{\text{Net-FS}}(\mathbf{Z}_{[n]}, \mathbf{x}; \mathbf{V}, \mathbf{Q})$ in (16) and $\hat{y}_{\text{ZS}}(\mathbf{x}; \mathbf{V}, \mathbf{Q})$ in (17), as well as the expected value of their cross term, are as follows:*

$$
\mathbb{E}_{\mathbf{Z}_{[n]}, \mathbf{x}, y | \theta_0} \left[ \left( \hat{y}_{\text{Net-FS}}(\mathbf{Z}_{[n]}, \mathbf{x}; \mathbf{V}, \mathbf{Q}) - y \right)^2 \right]
$$

$$
= \frac{1}{n} \cdot \text{tr} \left( \begin{bmatrix} \mathbf{Q}_{11} \\ \mathbf{q}_{21}^\top \end{bmatrix} \mathbf{\Sigma} \begin{bmatrix} \mathbf{Q}_{11} \\ \mathbf{q}_{21}^\top \end{bmatrix}^\top \mathbf{\Sigma}_{\text{joint}} \right) \cdot \begin{bmatrix} \mathbf{v}_{21} \\ v_{22} \end{bmatrix}^\top \mathbf{\Sigma}_{\text{joint}} \begin{bmatrix} \mathbf{v}_{21} \\ v_{22} \end{bmatrix} + \frac{n+1}{n} \cdot \begin{bmatrix} \mathbf{v}_{21} \\ v_{22} \end{bmatrix}^\top \mathbf{\Sigma}_{\text{joint}} \begin{bmatrix} \mathbf{Q}_{11} \\ \mathbf{q}_{21}^\top \end{bmatrix} \mathbf{\Sigma} \begin{bmatrix} \mathbf{Q}_{11} \\ \mathbf{q}_{21}^\top \end{bmatrix}^\top \mathbf{\Sigma}_{\text{joint}} \begin{bmatrix} \mathbf{v}_{21} \\ v_{22} \end{bmatrix}
$$

$$
+ \frac{q^2}{n} \cdot \begin{bmatrix} \mathbf{v}_{21} \\ v_{22} \end{bmatrix}^\top \mathbf{\Sigma}_{\text{joint}} \begin{bmatrix} \mathbf{v}_{21} \\ v_{22} \end{bmatrix} - 2 \begin{bmatrix} \mathbf{v}_{21} \\ v_{22} \end{bmatrix}^\top \mathbf{\Sigma}_{\text{joint}} \begin{bmatrix} \mathbf{Q}_{11} \\ \mathbf{q}_{21}^\top \end{bmatrix} \mathbf{\Sigma} \theta_0 + \theta_0^\top \mathbf{\Sigma} \theta_0 + \sigma^2,
$$

$$
\mathbb{E}_{\mathbf{x}, y | \theta_0} \left[ \left( \hat{y}_{\text{ZS}}(\mathbf{x}; \mathbf{V}, \mathbf{Q}) - y \right)^2 \right]
$$

$$
= \left( \text{tr}(\mathbf{Q}_{11}\mathbf{\Sigma})^2 + \text{tr}(\mathbf{Q}_{11}\mathbf{\Sigma}\mathbf{Q}_{11}^\top\mathbf{\Sigma}) + \text{tr}(\mathbf{Q}_{11}\mathbf{\Sigma}\mathbf{Q}_{11}\mathbf{\Sigma}) \right) \cdot \mathbf{v}_{21}^\top \mathbf{\Sigma} \mathbf{v}_{21} + 4 \, \text{tr}(\mathbf{Q}_{11}\mathbf{\Sigma}) \cdot \mathbf{v}_{21}^\top \mathbf{\Sigma} \mathbf{Q}_{11} \mathbf{\Sigma} \mathbf{v}_{21}
$$

$$
+ 2\mathbf{v}_{21}^\top \mathbf{\Sigma}(\mathbf{Q}_{11} + \mathbf{Q}_{11}^\top)\mathbf{\Sigma}(\mathbf{Q}_{11} + \mathbf{Q}_{11}^\top)\mathbf{\Sigma} \mathbf{v}_{21} + 2q \cdot \text{tr}(\mathbf{Q}_{11}\mathbf{\Sigma}) \cdot \mathbf{v}_{21}^\top \mathbf{\Sigma} \mathbf{v}_{21} + 4q \cdot \mathbf{v}_{21}^\top \mathbf{\Sigma} \mathbf{Q}_{11} \mathbf{\Sigma} \mathbf{v}_{21} + q^2 \cdot \mathbf{v}_{21}^\top \mathbf{\Sigma} \mathbf{v}_{21}
$$

$$
- 2\mathbf{v}_{21}^\top \left( \mathbf{\Sigma}(\mathbf{Q}_{11} + \mathbf{Q}_{11}^\top)\mathbf{\Sigma} + \text{tr}(\mathbf{Q}_{11}\mathbf{\Sigma}) \cdot \mathbf{\Sigma} + q \cdot \mathbf{\Sigma} \right) \theta_0 + \theta_0^\top \mathbf{\Sigma} \theta_0 + \sigma^2,
$$

$$
\mathbb{E}_{\mathbf{Z}_{[n]}, \mathbf{x}, y | \theta_0} \left[ \left( \hat{y}_{\text{Net-FS}}(\mathbf{Z}_{[n]}, \mathbf{x}; \mathbf{V}, \mathbf{Q}) - y \right) \left( \hat{y}_{\text{ZS}}(\mathbf{x}; \mathbf{V}, \mathbf{Q}) - y \right) \right]
$$

$$
= \left( \mathbf{v}_{21}^\top \mathbf{\Sigma} \mathbf{Q}_{11} + v_{22}\theta_0^\top \mathbf{\Sigma} \mathbf{Q}_{11} + \mathbf{v}_{21}^\top \mathbf{\Sigma} \theta_0 \mathbf{q}_{21}^\top + v_{22}(\theta_0^\top \mathbf{\Sigma} \theta_0 + \sigma^2)\mathbf{q}_{21}^\top \right) \mathbf{\Sigma} \left( (\mathbf{Q}_{11} + \mathbf{Q}_{11}^\top)\mathbf{\Sigma} \mathbf{v}_{21} + \text{tr}(\mathbf{Q}_{11}\mathbf{\Sigma})\mathbf{v}_{21} + q\mathbf{v}_{21} - \theta_0 \right)
$$

$$
- \mathbf{v}_{21}^\top \left( \mathbf{\Sigma}(\mathbf{Q}_{11} + \mathbf{Q}_{11}^\top)\mathbf{\Sigma} + \text{tr}(\mathbf{Q}_{11}\mathbf{\Sigma}) \cdot \mathbf{\Sigma} + q \cdot \mathbf{\Sigma} \right) \theta_0 + \theta_0^\top \mathbf{\Sigma} \theta_0 + \sigma^2.
$$

*Moreover, if $v_{22} = 1$, $\mathbf{q}_{21} = \mathbf{0}_d$, and $\mathbf{Q}_{11} = \mathbf{Q}_{11}^\top$, we have*

$$
\mathbb{E}_{\mathbf{Z}_{[n]}, \mathbf{x}, y | \theta_0} \left[ \left( \hat{y}_{\text{FS}}(\mathbf{Z}_{[n]}, \mathbf{x}; \mathbf{V}, \mathbf{Q}) - y \right)^2 \right] = \frac{1}{(n+1)^2} \cdot \mathbb{E}_{\mathbf{x}, y | \theta_0} \left[ \left( \hat{y}_{\text{ZS}}(\mathbf{x}; \mathbf{V}, \mathbf{Q}) - y \right)^2 \right] + \frac{n}{(n+1)^2} \cdot h(\mathbf{V}, \mathbf{Q}; \theta_0) \tag{19}
$$

*where*

$$
\mathbb{E}_{\mathbf{x}, y | \theta_0} \left[ \left( \hat{y}_{\text{ZS}}(\mathbf{x}; \mathbf{V}, \mathbf{Q}) - y \right)^2 \right]
$$

$$
= \mathbf{v}_{21}^\top \left( \text{tr}(\mathbf{Q}_{11}\mathbf{\Sigma})^2 \cdot \mathbf{\Sigma} + 2\,\text{tr}(\mathbf{Q}_{11}\mathbf{\Sigma}\mathbf{Q}_{11}\mathbf{\Sigma}) \cdot \mathbf{\Sigma} + 4\,\text{tr}(\mathbf{Q}_{11}\mathbf{\Sigma}) \cdot \mathbf{\Sigma} \mathbf{Q}_{11} \mathbf{\Sigma} + 8\mathbf{\Sigma} \mathbf{Q}_{11} \mathbf{\Sigma} \mathbf{Q}_{11} \mathbf{\Sigma} \right) \mathbf{v}_{21}
$$

$$
- 2\mathbf{v}_{21}^\top \left( 2\mathbf{\Sigma} \mathbf{Q}_{11} \mathbf{\Sigma} + \text{tr}(\mathbf{Q}_{11}\mathbf{\Sigma}) \cdot \mathbf{\Sigma} \right) \theta_0 + \theta_0^\top \mathbf{\Sigma} \theta_0 + \sigma^2
$$

$$
+ q^2 \cdot \mathbf{v}_{21}^\top \mathbf{\Sigma} \mathbf{v}_{21} + 2q \cdot \text{tr}(\mathbf{Q}_{11}\mathbf{\Sigma}) \cdot \mathbf{v}_{21}^\top \mathbf{\Sigma} \mathbf{v}_{21} + 4q \cdot \mathbf{v}_{21}^\top \mathbf{\Sigma} \mathbf{Q}_{11} \mathbf{\Sigma} \mathbf{v}_{21} - 2q \cdot \mathbf{v}_{21}^\top \mathbf{\Sigma} \theta_0,
$$

$$
h(\mathbf{V}, \mathbf{Q}; \theta_0) = \mathbf{v}_{21}^\top \left( \text{tr}(\mathbf{Q}_{11}\mathbf{\Sigma}\mathbf{Q}_{11}\mathbf{\Sigma}) \cdot \mathbf{\Sigma} + (n+5) \cdot \mathbf{\Sigma} \mathbf{Q}_{11} \mathbf{\Sigma} \mathbf{Q}_{11} \mathbf{\Sigma} + 2\,\text{tr}(\mathbf{Q}_{11}\mathbf{\Sigma}) \cdot \mathbf{\Sigma} \mathbf{Q}_{11} \mathbf{\Sigma} \right) \mathbf{v}_{21}
$$

$$
+ 2\mathbf{v}_{21}^\top \left( \text{tr}(\mathbf{Q}_{11}\mathbf{\Sigma}\mathbf{Q}_{11}\mathbf{\Sigma}) \cdot \mathbf{\Sigma} + (n+3) \cdot \mathbf{\Sigma} \mathbf{Q}_{11} \mathbf{\Sigma} \mathbf{Q}_{11} \mathbf{\Sigma} + \text{tr}(\mathbf{Q}_{11}\mathbf{\Sigma}) \cdot \mathbf{\Sigma} \mathbf{Q}_{11} \mathbf{\Sigma} \right) \theta_0
$$

$$
- 2\mathbf{v}_{21}^\top \left( (n+3) \cdot \mathbf{\Sigma} \mathbf{Q}_{11} \mathbf{\Sigma} + \text{tr}(\mathbf{Q}_{11}\mathbf{\Sigma}) \cdot \mathbf{\Sigma} \right) \theta_0 + (n+1) \cdot \theta_0^\top \mathbf{\Sigma} \mathbf{Q}_{11} \mathbf{\Sigma} \mathbf{Q}_{11} \mathbf{\Sigma} \theta_0
$$

$$
- 2(n+1) \cdot \theta_0^\top \mathbf{\Sigma} \mathbf{Q}_{11} \mathbf{\Sigma} \theta_0 + (\text{tr}(\mathbf{Q}_{11}\mathbf{\Sigma}\mathbf{Q}_{11}\mathbf{\Sigma}) + n + 2)(\theta_0^\top \mathbf{\Sigma} \theta_0 + \sigma^2)
$$

$$
+ q^2 \cdot \left( (\mathbf{v}_{21}^\top + \theta_0^\top)\mathbf{\Sigma}(\mathbf{v}_{21} + \theta_0) + \sigma^2 \right) + 2q \cdot \left( \mathbf{v}_{21}^\top \mathbf{\Sigma} \mathbf{Q}_{11} + \theta_0^\top \mathbf{\Sigma} \mathbf{Q}_{11} - \theta_0^\top \right) \mathbf{\Sigma} \mathbf{v}_{21}.
$$

*Proof.* By using Lemma A.4, we obtain

$$
\mathbb{E}_{\mathbf{Z}_{[n]}, \mathbf{x}, y | \theta_0} \left[ \left( \hat{y}_{\text{Net-FS}}(\mathbf{Z}_{[n]}, \mathbf{x}; \mathbf{V}, \mathbf{Q}) - y \right)^2 \right]
$$

$$
= \text{Var} \left[ \hat{y}_{\text{Net-FS}}(\mathbf{Z}_{[n]}, \mathbf{x}; \mathbf{V}, \mathbf{Q}) \right] - 2\text{Cov} \left[ \hat{y}_{\text{Net-FS}}(\mathbf{Z}_{[n]}, \mathbf{x}; \mathbf{V}, \mathbf{Q}), y \right] + \text{Var}\left[ y \right]
$$

$$
= \frac{1}{n} \cdot \text{tr} \left( \begin{bmatrix} \mathbf{Q}_{11} \\ \mathbf{q}_{21}^\top \end{bmatrix} \mathbf{\Sigma} \begin{bmatrix} \mathbf{Q}_{11} \\ \mathbf{q}_{21}^\top \end{bmatrix}^\top \mathbf{\Sigma}_{\text{joint}} \right) \cdot \begin{bmatrix} \mathbf{v}_{21} \\ v_{22} \end{bmatrix}^\top \mathbf{\Sigma}_{\text{joint}} \begin{bmatrix} \mathbf{v}_{21} \\ v_{22} \end{bmatrix} + \frac{n+1}{n} \cdot \begin{bmatrix} \mathbf{v}_{21} \\ v_{22} \end{bmatrix}^\top \mathbf{\Sigma}_{\text{joint}} \begin{bmatrix} \mathbf{Q}_{11} \\ \mathbf{q}_{21}^\top \end{bmatrix} \mathbf{\Sigma} \begin{bmatrix} \mathbf{Q}_{11} \\ \mathbf{q}_{21}^\top \end{bmatrix}^\top \mathbf{\Sigma}_{\text{joint}} \begin{bmatrix} \mathbf{v}_{21} \\ v_{22} \end{bmatrix}
$$

$$
+ \frac{q^2}{n} \cdot \begin{bmatrix} \mathbf{v}_{21} \\ v_{22} \end{bmatrix}^\top \mathbf{\Sigma}_{\text{joint}} \begin{bmatrix} \mathbf{v}_{21} \\ v_{22} \end{bmatrix} - 2 \begin{bmatrix} \mathbf{v}_{21} \\ v_{22} \end{bmatrix}^\top \mathbf{\Sigma}_{\text{joint}} \begin{bmatrix} \mathbf{Q}_{11} \\ \mathbf{q}_{21}^\top \end{bmatrix} \mathbf{\Sigma} \theta_0 + \theta_0^\top \mathbf{\Sigma} \theta_0 + \sigma^2, \tag{20}
$$

$$
\begin{aligned}
\mathbb{E}_{\mathbf{x},y|\theta_0}\left[\left(\hat{y}_{\mathrm{ZS}}(\mathbf{x};\mathbf{V},\mathbf{Q})-y\right)^2\right] &= \mathrm{Var}\left[\hat{y}_{\mathrm{ZS}}(\mathbf{x};\mathbf{V},\mathbf{Q})\right] - 2\mathrm{Cov}\left[\hat{y}_{\mathrm{ZS}}(\mathbf{x};\mathbf{V},\mathbf{Q}),y\right] + \mathrm{Var}\left[y\right] \\
&= \left(\mathrm{tr}(\mathbf{Q}_{11}\boldsymbol{\Sigma})^2 + \mathrm{tr}(\mathbf{Q}_{11}\boldsymbol{\Sigma}\mathbf{Q}_{11}^\top\boldsymbol{\Sigma}) + \mathrm{tr}(\mathbf{Q}_{11}\boldsymbol{\Sigma}\mathbf{Q}_{11}\boldsymbol{\Sigma})\right)\cdot\mathbf{v}_{21}^\top\boldsymbol{\Sigma}\mathbf{v}_{21} \\
&\quad + 4\,\mathrm{tr}(\mathbf{Q}_{11}\boldsymbol{\Sigma})\cdot\mathbf{v}_{21}^\top\boldsymbol{\Sigma}\mathbf{Q}_{11}\boldsymbol{\Sigma}\mathbf{v}_{21} + 2\mathbf{v}_{21}^\top\boldsymbol{\Sigma}(\mathbf{Q}_{11}+\mathbf{Q}_{11}^\top)\boldsymbol{\Sigma}(\mathbf{Q}_{11}+\mathbf{Q}_{11}^\top)\boldsymbol{\Sigma}\mathbf{v}_{21} \\
&\quad + 2q\cdot\mathrm{tr}(\mathbf{Q}_{11}\boldsymbol{\Sigma})\cdot\mathbf{v}_{21}^\top\boldsymbol{\Sigma}\mathbf{v}_{21} + 4q\cdot\mathbf{v}_{21}^\top\boldsymbol{\Sigma}\mathbf{Q}_{11}\boldsymbol{\Sigma}\mathbf{v}_{21} + q^2\cdot\mathbf{v}_{21}^\top\boldsymbol{\Sigma}\mathbf{v}_{21} \\
&\quad - 2\mathbf{v}_{21}^\top\left(\boldsymbol{\Sigma}(\mathbf{Q}_{11}+\mathbf{Q}_{11}^\top)\boldsymbol{\Sigma} + \mathrm{tr}(\mathbf{Q}_{11}\boldsymbol{\Sigma})\cdot\boldsymbol{\Sigma} + q\cdot\boldsymbol{\Sigma}\right)\theta_0 + \theta_0^\top\boldsymbol{\Sigma}\theta_0 + \sigma^2,
\end{aligned}
$$

and

$$
\begin{aligned}
&\mathbb{E}_{\mathbf{Z}_{[n]},\mathbf{x},y|\theta_0}\left[\left(\hat{y}_{\mathrm{Net\text{-}FS}}(\mathbf{Z}_{[n]},\mathbf{x};\mathbf{V},\mathbf{Q})-y\right)\left(\hat{y}_{\mathrm{ZS}}(\mathbf{x};\mathbf{V},\mathbf{Q})-y\right)\right] \\
&= \begin{bmatrix}\mathbf{v}_{21}\\v_{22}\end{bmatrix}^\top\boldsymbol{\Sigma}_{\mathrm{joint}}\begin{bmatrix}\mathbf{Q}_{11}\\\mathbf{q}_{21}^\top\end{bmatrix}\boldsymbol{\Sigma}\left((\mathbf{Q}_{11}+\mathbf{Q}_{11}^\top)\boldsymbol{\Sigma}+\mathrm{tr}(\mathbf{Q}_{11}\boldsymbol{\Sigma})+q\cdot\mathbf{I}_d\right)\mathbf{v}_{21} + \left(\theta_0^\top\boldsymbol{\Sigma}\theta_0+\sigma^2\right) \\
&\quad - \begin{bmatrix}\mathbf{v}_{21}\\v_{22}\end{bmatrix}^\top\boldsymbol{\Sigma}_{\mathrm{joint}}\begin{bmatrix}\mathbf{Q}_{11}\\\mathbf{q}_{21}^\top\end{bmatrix}\boldsymbol{\Sigma}\theta_0 - \mathbf{v}_{21}^\top\left(\boldsymbol{\Sigma}(\mathbf{Q}_{11}+\mathbf{Q}_{11}^\top)\boldsymbol{\Sigma}+\mathrm{tr}(\mathbf{Q}_{11}\boldsymbol{\Sigma})\cdot\boldsymbol{\Sigma}+q\cdot\boldsymbol{\Sigma}\right)\theta_0 \\
&= \begin{bmatrix}\mathbf{v}_{21}\\v_{22}\end{bmatrix}^\top\boldsymbol{\Sigma}_{\mathrm{joint}}\begin{bmatrix}\mathbf{Q}_{11}\\\mathbf{q}_{21}^\top\end{bmatrix}\boldsymbol{\Sigma}\left((\mathbf{Q}_{11}+\mathbf{Q}_{11}^\top)\boldsymbol{\Sigma}\mathbf{v}_{21}+\mathrm{tr}(\mathbf{Q}_{11}\boldsymbol{\Sigma})\cdot\mathbf{v}_{21}+q\cdot\mathbf{v}_{21}-\theta_0\right) \\
&\quad - \mathbf{v}_{21}^\top\left(\boldsymbol{\Sigma}(\mathbf{Q}_{11}+\mathbf{Q}_{11}^\top)\boldsymbol{\Sigma}+\mathrm{tr}(\mathbf{Q}_{11}\boldsymbol{\Sigma})\cdot\boldsymbol{\Sigma}+q\cdot\boldsymbol{\Sigma}\right)\theta_0+\theta_0^\top\boldsymbol{\Sigma}\theta_0+\sigma^2 \\
&= \left(\mathbf{v}_{21}^\top\boldsymbol{\Sigma}\mathbf{Q}_{11}+v_{22}\theta_0^\top\boldsymbol{\Sigma}\mathbf{Q}_{11}+\mathbf{v}_{21}^\top\boldsymbol{\Sigma}\theta_0\mathbf{q}_{21}^\top+v_{22}(\theta_0^\top\boldsymbol{\Sigma}\theta_0+\sigma^2)\mathbf{q}_{21}^\top\right) \\
&\quad \cdot\boldsymbol{\Sigma}\left((\mathbf{Q}_{11}+\mathbf{Q}_{11}^\top)\boldsymbol{\Sigma}\mathbf{v}_{21}+\mathrm{tr}(\mathbf{Q}_{11}\boldsymbol{\Sigma})\cdot\mathbf{v}_{21}+q\cdot\mathbf{v}_{21}-\theta_0\right) \\
&\quad - \mathbf{v}_{21}^\top\left(\boldsymbol{\Sigma}(\mathbf{Q}_{11}+\mathbf{Q}_{11}^\top)\boldsymbol{\Sigma}+\mathrm{tr}(\mathbf{Q}_{11}\boldsymbol{\Sigma})\cdot\boldsymbol{\Sigma}+q\cdot\boldsymbol{\Sigma}\right)\theta_0+\theta_0^\top\boldsymbol{\Sigma}\theta_0+\sigma^2,
\end{aligned} \tag{21}
$$

where the last equality comes from

$$
\begin{bmatrix}\mathbf{v}_{21}\\v_{22}\end{bmatrix}^\top\boldsymbol{\Sigma}_{\mathrm{joint}}\begin{bmatrix}\mathbf{Q}_{11}\\\mathbf{q}_{21}^\top\end{bmatrix} = \mathbf{v}_{21}^\top\boldsymbol{\Sigma}\mathbf{Q}_{11}+v_{22}\theta_0^\top\boldsymbol{\Sigma}\mathbf{Q}_{11}+\mathbf{v}_{21}^\top\boldsymbol{\Sigma}\theta_0\mathbf{q}_{21}^\top+v_{22}(\theta_0^\top\boldsymbol{\Sigma}\theta_0+\sigma^2)\mathbf{q}_{21}^\top.
$$

Now consider the case where $v_{22}=1$, $\mathbf{q}_{21}=\mathbf{0}_d$, and $\mathbf{Q}_{11}=\mathbf{Q}_{11}^\top$. It follows that

$$
\mathrm{tr}\left(\begin{bmatrix}\mathbf{Q}_{11}\\\mathbf{q}_{21}^\top\end{bmatrix}\boldsymbol{\Sigma}\begin{bmatrix}\mathbf{Q}_{11}\\\mathbf{q}_{21}^\top\end{bmatrix}^\top\boldsymbol{\Sigma}_{\mathrm{joint}}\right) = \mathrm{tr}\left(\boldsymbol{\Sigma}\begin{bmatrix}\mathbf{Q}_{11}&\mathbf{0}_d\end{bmatrix}\begin{bmatrix}\boldsymbol{\Sigma}&\boldsymbol{\Sigma}\theta_0\\\theta_0^\top\boldsymbol{\Sigma}&\theta_0^\top\boldsymbol{\Sigma}\theta_0+\sigma^2\end{bmatrix}\begin{bmatrix}\mathbf{Q}_{11}\\\mathbf{0}_d^\top\end{bmatrix}\right) = \mathrm{tr}\left(\mathbf{Q}_{11}\boldsymbol{\Sigma}\mathbf{Q}_{11}\boldsymbol{\Sigma}\right),
$$

$$
\boldsymbol{\Sigma}_{\mathrm{joint}}\begin{bmatrix}\mathbf{v}_{21}\\v_{22}\end{bmatrix} = \begin{bmatrix}\boldsymbol{\Sigma}&\boldsymbol{\Sigma}\theta_0\\\theta_0^\top\boldsymbol{\Sigma}&\theta_0^\top\boldsymbol{\Sigma}\theta_0+\sigma^2\end{bmatrix}\begin{bmatrix}\mathbf{v}_{21}\\1\end{bmatrix} = \begin{bmatrix}\boldsymbol{\Sigma}(\mathbf{v}_{21}+\theta_0)\\\theta_0^\top\boldsymbol{\Sigma}(\mathbf{v}_{21}+\theta_0)+\sigma^2\end{bmatrix},
$$

$$
\begin{bmatrix}\mathbf{Q}_{11}\\\mathbf{q}_{21}^\top\end{bmatrix}^\top\boldsymbol{\Sigma}_{\mathrm{joint}}\begin{bmatrix}\mathbf{v}_{21}\\v_{22}\end{bmatrix} = \begin{bmatrix}\mathbf{Q}_{11}&\mathbf{0}_d\end{bmatrix}\begin{bmatrix}\boldsymbol{\Sigma}(\mathbf{v}_{21}+\theta_0)\\\theta_0^\top\boldsymbol{\Sigma}(\mathbf{v}_{21}+\theta_0)+\sigma^2\end{bmatrix} = \mathbf{Q}_{11}\boldsymbol{\Sigma}(\mathbf{v}_{21}+\theta_0),
$$

which implies that the expressions in (20) and (21) simplify to

$$
\begin{aligned}
&\mathbb{E}_{\mathbf{Z}_{[n]},\mathbf{x},y|\theta_0}\left[\left(\hat{y}_{\mathrm{Net\text{-}FS}}(\mathbf{Z}_{[n]},\mathbf{x};\mathbf{V},\mathbf{Q})-y\right)^2\right] \\
&= \frac{1}{n}\cdot\mathrm{tr}\left(\mathbf{Q}_{11}\boldsymbol{\Sigma}\mathbf{Q}_{11}\boldsymbol{\Sigma}\right)\cdot\begin{bmatrix}\mathbf{v}_{21}\\1\end{bmatrix}^\top\begin{bmatrix}\boldsymbol{\Sigma}(\mathbf{v}_{21}+\theta_0)\\\theta_0^\top\boldsymbol{\Sigma}(\mathbf{v}_{21}+\theta_0)+\sigma^2\end{bmatrix} + \frac{n+1}{n}\cdot(\mathbf{v}_{21}^\top+\theta_0^\top)\boldsymbol{\Sigma}\mathbf{Q}_{11}\cdot\boldsymbol{\Sigma}\cdot\mathbf{Q}_{11}\boldsymbol{\Sigma}(\mathbf{v}_{21}+\theta_0) \\
&\quad + \frac{q^2}{n}\cdot\begin{bmatrix}\mathbf{v}_{21}\\1\end{bmatrix}^\top\begin{bmatrix}\boldsymbol{\Sigma}(\mathbf{v}_{21}+\theta_0)\\\theta_0^\top\boldsymbol{\Sigma}(\mathbf{v}_{21}+\theta_0)+\sigma^2\end{bmatrix} - 2(\mathbf{v}_{21}^\top+\theta_0^\top)\boldsymbol{\Sigma}\mathbf{Q}_{11}\cdot\boldsymbol{\Sigma}\theta_0 + \theta_0^\top\boldsymbol{\Sigma}\theta_0+\sigma^2 \\
&= \frac{1}{n}\cdot\left(\mathrm{tr}\left(\mathbf{Q}_{11}\boldsymbol{\Sigma}\mathbf{Q}_{11}\boldsymbol{\Sigma}\right)+q^2\right)\cdot\left((\mathbf{v}_{21}^\top+\theta_0^\top)\boldsymbol{\Sigma}(\mathbf{v}_{21}+\theta_0)+\sigma^2\right) \\
&\quad + \frac{n+1}{n}\cdot(\mathbf{v}_{21}^\top+\theta_0^\top)\boldsymbol{\Sigma}\mathbf{Q}_{11}\boldsymbol{\Sigma}\mathbf{Q}_{11}\boldsymbol{\Sigma}(\mathbf{v}_{21}+\theta_0)-2(\mathbf{v}_{21}^\top+\theta_0^\top)\boldsymbol{\Sigma}\mathbf{Q}_{11}\boldsymbol{\Sigma}\theta_0+\theta_0^\top\boldsymbol{\Sigma}\theta_0+\sigma^2,
\end{aligned}
$$

$$\mathbb{E}_{\mathbf{Z}_{[n]},\mathbf{x},y|\theta_0}\left[\left(\hat{y}_{\text{Net-FS}}(\mathbf{Z}_{[n]},\mathbf{x};\mathbf{V},\mathbf{Q})-y\right)(\hat{y}_{\text{ZS}}(\mathbf{x};\mathbf{V},\mathbf{Q})-y)\right]$$

$$= \left(\mathbf{v}_{21}^\top\boldsymbol{\Sigma}\mathbf{Q}_{11}+\theta_0^\top\boldsymbol{\Sigma}\mathbf{Q}_{11}\right)\boldsymbol{\Sigma}\left(2\mathbf{Q}_{11}\boldsymbol{\Sigma}\mathbf{v}_{21}+\text{tr}(\mathbf{Q}_{11}\boldsymbol{\Sigma})\cdot\mathbf{v}_{21}+q\cdot\mathbf{v}_{21}-\theta_0\right)$$

$$- \mathbf{v}_{21}^\top\left(2\boldsymbol{\Sigma}\mathbf{Q}_{11}\boldsymbol{\Sigma}+\text{tr}(\mathbf{Q}_{11}\boldsymbol{\Sigma})\cdot\boldsymbol{\Sigma}+q\cdot\boldsymbol{\Sigma}\right)\theta_0+\theta_0^\top\boldsymbol{\Sigma}\theta_0+\sigma^2.$$

These give

$$n\cdot\mathbb{E}_{\mathbf{Z}_{[n]},\mathbf{x},y|\theta_0}\left[\left(\hat{y}_{\text{Net-FS}}(\mathbf{Z}_{[n]},\mathbf{x};\mathbf{V},\mathbf{Q})-y\right)^2\right]+2\cdot\mathbb{E}_{\mathbf{Z}_{[n]},\mathbf{x},y|\theta_0}\left[\left(\hat{y}_{\text{Net-FS}}(\mathbf{Z}_{[n]},\mathbf{x};\mathbf{V},\mathbf{Q})-y\right)(\hat{y}_{\text{ZS}}(\mathbf{x};\mathbf{V},\mathbf{Q})-y)\right]$$

$$= \left(\text{tr}(\mathbf{Q}_{11}\boldsymbol{\Sigma}\mathbf{Q}_{11}\boldsymbol{\Sigma})+q^2\right)\cdot\left((\mathbf{v}_{21}^\top+\theta_0^\top)\boldsymbol{\Sigma}(\mathbf{v}_{21}+\theta_0)+\sigma^2\right)+(n+1)\cdot(\mathbf{v}_{21}^\top+\theta_0^\top)\boldsymbol{\Sigma}\mathbf{Q}_{11}\boldsymbol{\Sigma}\mathbf{Q}_{11}\boldsymbol{\Sigma}(\mathbf{v}_{21}+\theta_0)$$

$$- 2n\cdot(\mathbf{v}_{21}^\top+\theta_0^\top)\boldsymbol{\Sigma}\mathbf{Q}_{11}\boldsymbol{\Sigma}\theta_0+2\cdot\left(\mathbf{v}_{21}^\top\boldsymbol{\Sigma}\mathbf{Q}_{11}+\theta_0^\top\boldsymbol{\Sigma}\mathbf{Q}_{11}\right)\boldsymbol{\Sigma}\left(2\mathbf{Q}_{11}\boldsymbol{\Sigma}\mathbf{v}_{21}+\text{tr}(\mathbf{Q}_{11}\boldsymbol{\Sigma})\cdot\mathbf{v}_{21}+q\cdot\mathbf{v}_{21}-\theta_0\right)$$

$$- 2\cdot\mathbf{v}_{21}^\top\left(2\boldsymbol{\Sigma}\mathbf{Q}_{11}\boldsymbol{\Sigma}+\text{tr}(\mathbf{Q}_{11}\boldsymbol{\Sigma})\cdot\boldsymbol{\Sigma}+q\cdot\boldsymbol{\Sigma}\right)\theta_0+(n+2)\cdot(\theta_0^\top\boldsymbol{\Sigma}\theta_0+\sigma^2)$$

$$= \mathbf{v}_{21}^\top\left(\text{tr}(\mathbf{Q}_{11}\boldsymbol{\Sigma}\mathbf{Q}_{11}\boldsymbol{\Sigma})\cdot\boldsymbol{\Sigma}+(n+5)\cdot\boldsymbol{\Sigma}\mathbf{Q}_{11}\boldsymbol{\Sigma}\mathbf{Q}_{11}\boldsymbol{\Sigma}+2\,\text{tr}(\mathbf{Q}_{11}\boldsymbol{\Sigma})\cdot\boldsymbol{\Sigma}\mathbf{Q}_{11}\boldsymbol{\Sigma}\right)\mathbf{v}_{21}$$

$$+ 2\mathbf{v}_{21}^\top\left(\text{tr}(\mathbf{Q}_{11}\boldsymbol{\Sigma}\mathbf{Q}_{11}\boldsymbol{\Sigma})\cdot\boldsymbol{\Sigma}+(n+3)\cdot\boldsymbol{\Sigma}\mathbf{Q}_{11}\boldsymbol{\Sigma}\mathbf{Q}_{11}\boldsymbol{\Sigma}+\text{tr}(\mathbf{Q}_{11}\boldsymbol{\Sigma})\cdot\boldsymbol{\Sigma}\mathbf{Q}_{11}\boldsymbol{\Sigma}\right)\theta_0$$

$$- 2\mathbf{v}_{21}^\top\left((n+3)\cdot\boldsymbol{\Sigma}\mathbf{Q}_{11}\boldsymbol{\Sigma}+\text{tr}(\mathbf{Q}_{11}\boldsymbol{\Sigma})\cdot\boldsymbol{\Sigma}\right)\theta_0+(n+1)\cdot\theta_0^\top\boldsymbol{\Sigma}\mathbf{Q}_{11}\boldsymbol{\Sigma}\mathbf{Q}_{11}\boldsymbol{\Sigma}\theta_0-2(n+1)\cdot\theta_0^\top\boldsymbol{\Sigma}\mathbf{Q}_{11}\boldsymbol{\Sigma}\theta_0$$

$$+ (\text{tr}(\mathbf{Q}_{11}\boldsymbol{\Sigma}\mathbf{Q}_{11}\boldsymbol{\Sigma})+n+2)(\theta_0^\top\boldsymbol{\Sigma}\theta_0+\sigma^2)$$

$$+ q^2\cdot\left((\mathbf{v}_{21}^\top+\theta_0^\top)\boldsymbol{\Sigma}(\mathbf{v}_{21}+\theta_0)+\sigma^2\right)+2q\cdot\left(\mathbf{v}_{21}^\top\boldsymbol{\Sigma}\mathbf{Q}_{11}+\theta_0^\top\boldsymbol{\Sigma}\mathbf{Q}_{11}-\theta_0\right)\boldsymbol{\Sigma}\mathbf{v}_{21}. \tag{22}$$

As a result, we have

$$\mathbb{E}_{\mathbf{Z}_{[n]},\mathbf{x},y|\theta_0}\left[\left(\hat{y}_{\text{FS}}(\mathbf{Z}_{[n]},\mathbf{x};\mathbf{V},\mathbf{Q})-y\right)^2\right]$$

$$= \frac{n^2}{(n+1)^2}\cdot\mathbb{E}_{\mathbf{Z}_{[n]},\mathbf{x},y|\theta_0}\left[\left(\hat{y}_{\text{Net-FS}}(\mathbf{Z}_{[n]},\mathbf{x};\mathbf{V},\mathbf{Q})-y\right)^2\right]+\frac{1}{(n+1)^2}\cdot\mathbb{E}_{\mathbf{x},y|\theta_0}\left[\left(\hat{y}_{\text{ZS}}(\mathbf{x};\mathbf{V},\mathbf{Q})-y\right)^2\right]$$

$$+ \frac{2n}{(n+1)^2}\cdot\mathbb{E}_{\mathbf{Z}_{[n]},\mathbf{x},y|\theta_0}\left[\left(\hat{y}_{\text{Net-FS}}(\mathbf{Z}_{[n]},\mathbf{x};\mathbf{V},\mathbf{Q})-y\right)(\hat{y}_{\text{ZS}}(\mathbf{x};\mathbf{V},\mathbf{Q})-y)\right]$$

$$= \frac{n}{(n+1)^2}\cdot\left(n\cdot\mathbb{E}_{\mathbf{Z}_{[n]},\mathbf{x},y|\theta_0}\left[\left(\hat{y}_{\text{Net-FS}}(\mathbf{Z}_{[n]},\mathbf{x};\mathbf{V},\mathbf{Q})-y\right)^2\right]\right.$$

$$\left.+ 2\cdot\mathbb{E}_{\mathbf{Z}_{[n]},\mathbf{x},y|\theta_0}\left[\left(\hat{y}_{\text{Net-FS}}(\mathbf{Z}_{[n]},\mathbf{x};\mathbf{V},\mathbf{Q})-y\right)(\hat{y}_{\text{ZS}}(\mathbf{x};\mathbf{V},\mathbf{Q})-y)\right]\right)$$

$$+ \frac{1}{(n+1)^2}\cdot\mathbb{E}_{\mathbf{x},y|\theta_0}\left[\left(\hat{y}_{\text{ZS}}(\mathbf{x};\mathbf{V},\mathbf{Q})-y\right)^2\right],$$

where the expression inside the parentheses of the first term is given in (22), and the last term is

$$\mathbb{E}_{\mathbf{x},y|\theta_0}\left[\left(\hat{y}_{\text{ZS}}(\mathbf{x};\mathbf{V},\mathbf{Q})-y\right)^2\right]$$

$$= \left(\text{tr}(\mathbf{Q}_{11}\boldsymbol{\Sigma})^2+2\,\text{tr}(\mathbf{Q}_{11}\boldsymbol{\Sigma}\mathbf{Q}_{11}\boldsymbol{\Sigma})\right)\cdot\mathbf{v}_{21}^\top\boldsymbol{\Sigma}\mathbf{v}_{21}+4\,\text{tr}(\mathbf{Q}_{11}\boldsymbol{\Sigma})\cdot\mathbf{v}_{21}^\top\boldsymbol{\Sigma}\mathbf{Q}_{11}\boldsymbol{\Sigma}\mathbf{v}_{21}+8\mathbf{v}_{21}^\top\boldsymbol{\Sigma}\mathbf{Q}_{11}\boldsymbol{\Sigma}\mathbf{Q}_{11}\boldsymbol{\Sigma}\mathbf{v}_{21}$$

$$- 2\mathbf{v}_{21}^\top(2\boldsymbol{\Sigma}\mathbf{Q}_{11}\boldsymbol{\Sigma}+\text{tr}(\mathbf{Q}_{11}\boldsymbol{\Sigma})\cdot\boldsymbol{\Sigma})\theta_0-2q\cdot\mathbf{v}_{21}^\top\boldsymbol{\Sigma}\theta_0+\theta_0^\top\boldsymbol{\Sigma}\theta_0+\sigma^2$$

$$+ q^2\cdot\mathbf{v}_{21}^\top\boldsymbol{\Sigma}\mathbf{v}_{21}+2q\cdot\text{tr}(\mathbf{Q}_{11}\boldsymbol{\Sigma})\cdot\mathbf{v}_{21}^\top\boldsymbol{\Sigma}\mathbf{v}_{21}+4q\cdot\mathbf{v}_{21}^\top\boldsymbol{\Sigma}\mathbf{Q}_{11}\boldsymbol{\Sigma}\mathbf{v}_{21}$$

$$= \mathbf{v}_{21}^\top\left(\text{tr}(\mathbf{Q}_{11}\boldsymbol{\Sigma})^2\cdot\boldsymbol{\Sigma}+2\,\text{tr}(\mathbf{Q}_{11}\boldsymbol{\Sigma}\mathbf{Q}_{11}\boldsymbol{\Sigma})\cdot\boldsymbol{\Sigma}+4\,\text{tr}(\mathbf{Q}_{11}\boldsymbol{\Sigma})\cdot\boldsymbol{\Sigma}\mathbf{Q}_{11}\boldsymbol{\Sigma}+8\boldsymbol{\Sigma}\mathbf{Q}_{11}\boldsymbol{\Sigma}\mathbf{Q}_{11}\boldsymbol{\Sigma}\right)\mathbf{v}_{21}$$

$$- 2\mathbf{v}_{21}^\top\left(2\boldsymbol{\Sigma}\mathbf{Q}_{11}\boldsymbol{\Sigma}+\text{tr}(\mathbf{Q}_{11}\boldsymbol{\Sigma})\cdot\boldsymbol{\Sigma}\right)\theta_0+\theta_0^\top\boldsymbol{\Sigma}\theta_0+\sigma^2$$

$$+ q^2\cdot\mathbf{v}_{21}^\top\boldsymbol{\Sigma}\mathbf{v}_{21}+2q\cdot\text{tr}(\mathbf{Q}_{11}\boldsymbol{\Sigma})\cdot\mathbf{v}_{21}^\top\boldsymbol{\Sigma}\mathbf{v}_{21}+4q\cdot\mathbf{v}_{21}^\top\boldsymbol{\Sigma}\mathbf{Q}_{11}\boldsymbol{\Sigma}\mathbf{v}_{21}-2q\cdot\mathbf{v}_{21}^\top\boldsymbol{\Sigma}\theta_0.$$

$$\square$$

**Lemma A.6.** *Given $\theta_0 \in \mathbb{R}^d$, assume $v_{22} \neq 0$, consider the parameters*

$$\mathbf{V} = \begin{bmatrix} 0 & \mathbf{0}_d^\top & 0 \\ \mathbf{0}_d & \mathbf{0}_d\mathbf{0}_d^\top & \mathbf{0}_d \\ 0 & \mathbf{v}_{21}^\top & v_{22} \end{bmatrix}, \qquad \mathbf{Q} = \begin{bmatrix} 0 & \mathbf{0}_d^\top & 0 \\ \mathbf{0}_d & \mathbf{\Sigma}^{-1} & \mathbf{0}_d \\ 0 & \mathbf{0}_d^\top & 0 \end{bmatrix}. \tag{23}$$

*Then, the following holds:*

$$\mathbb{E}_{\mathbf{x},y|\theta_0}\left[\left(\hat{y}_{\mathrm{ZS}}(\mathbf{x};\mathbf{V},\mathbf{Q}) - y\right)^2\right] = (d+2)(d+4)\mathbf{v}_{21}^\top\mathbf{\Sigma}\mathbf{v}_{21} - 2(d+2)\mathbf{v}_{21}^\top\mathbf{\Sigma}\theta_0 + \theta_0^\top\mathbf{\Sigma}\theta_0 + \sigma^2,$$

$$\mathbb{E}_{\mathbf{Z}_{[n]},\mathbf{x},y|\theta_0}\left[\left(\hat{y}_{\mathrm{FS}}(\mathbf{Z}_{[n]},\mathbf{x};\mathbf{V},\mathbf{Q}) - y\right)^2\right]$$

$$= \frac{1}{(n+1)^2}\Big(\big((d+2)(d+4) + n(n+5+3d)\big)\mathbf{v}_{21}^\top\mathbf{\Sigma}\mathbf{v}_{21} + 2\big(-d+2+n(n+3+2d)v_{22}-n(n+3+d)\big)\mathbf{v}_{21}^\top\mathbf{\Sigma}\theta_0$$

$$+ \big(n(n+1+d)v_{22}^2 - 2n(n+1)v_{22} + (n+1)^2\big)\theta_0^\top\mathbf{\Sigma}\theta_0 + \big(nd\cdot v_{22}^2 + (n+1)^2\big)\sigma^2\Big).$$

$$\lim_{n\to\infty}\mathbb{E}_{\mathbf{Z}_{[n]},\mathbf{x},y|\theta_0}\left[\left(\hat{y}_{\mathrm{FS}}(\mathbf{Z}_{[n]},\mathbf{x};\mathbf{V},\mathbf{Q}) - y\right)^2\right] = \big(\mathbf{v}_{21} + (v_{22}-1)\theta_0\big)^\top\mathbf{\Sigma}\big(\mathbf{v}_{21} + (v_{22}-1)\theta_0\big) + \sigma^2.$$

*Proof.* By the definition of the linear attention model in (1), for any input $\mathbf{Z}$, the prediction $f(\mathbf{Z};\mathbf{V},\mathbf{Q})$ produced by the parameters in (23) is identical to that produced by the rescaled parameter pair

$$\mathbf{V} = \begin{bmatrix} 0 & \mathbf{0}_d^\top & 0 \\ \mathbf{0}_d & \mathbf{0}_d\mathbf{0}_d^\top & \mathbf{0}_d \\ 0 & \frac{1}{v_{22}}\cdot\mathbf{v}_{21}^\top & 1 \end{bmatrix}, \qquad \mathbf{Q} = \begin{bmatrix} 0 & \mathbf{0}_d^\top & 0 \\ \mathbf{0}_d & v_{22}\cdot\mathbf{\Sigma}^{-1} & \mathbf{0}_d \\ 0 & \mathbf{0}_d^\top & 0 \end{bmatrix}.$$

By Lemma A.5, we have

$$\mathbb{E}_{\mathbf{x},y|\theta_0}\left[\left(\hat{y}_{\mathrm{ZS}}(\mathbf{x};\mathbf{V},\mathbf{Q}) - y\right)^2\right] = (d+2)(d+4)\mathbf{v}_{21}^\top\mathbf{\Sigma}\mathbf{v}_{21} - 2(d+2)\mathbf{v}_{21}^\top\mathbf{\Sigma}\theta_0 + \theta_0^\top\mathbf{\Sigma}\theta_0 + \sigma^2,$$

$$h(\mathbf{V},\mathbf{Q};\theta_0) = (n+5+3d)\mathbf{v}_{21}^\top\mathbf{\Sigma}\mathbf{v}_{21} + 2\big((n+3+2d)v_{22} - (n+3+d)\big)\mathbf{v}_{21}^\top\mathbf{\Sigma}\theta_0$$

$$+ (n+1)v_{22}^2\theta_0^\top\mathbf{\Sigma}\theta_0 - 2(n+1)v_{22}\theta_0^\top\mathbf{\Sigma}\theta_0 + (d\cdot v_{22}^2+n+2)(\theta_0^\top\mathbf{\Sigma}\theta_0 + \sigma^2),$$

which implies

$$\mathbb{E}_{\mathbf{Z}_{[n]},\mathbf{x},y|\theta_0}\left[\left(\hat{y}_{\mathrm{FS}}(\mathbf{Z}_{[n]},\mathbf{x};\mathbf{V},\mathbf{Q}) - y\right)^2\right]$$

$$= \frac{1}{(n+1)^2}\cdot\mathbb{E}_{\mathbf{x},y|\theta_0}\left[\left(\hat{y}_{\mathrm{ZS}}(\mathbf{x};\mathbf{V},\mathbf{Q}) - y\right)^2\right] + \frac{n}{(n+1)^2}\cdot h(\mathbf{V},\mathbf{Q};\theta_0)$$

$$= \frac{1}{(n+1)^2}\Big(\big((d+2)(d+4) + n(n+5+3d)\big)\mathbf{v}_{21}^\top\mathbf{\Sigma}\mathbf{v}_{21} + 2\big(-d+2+n(n+3+2d)v_{22}-n(n+3+d)\big)\mathbf{v}_{21}^\top\mathbf{\Sigma}\theta_0$$

$$+ \big(n(n+1+d)v_{22}^2 - 2n(n+1)v_{22} + (n+1)^2\big)\theta_0^\top\mathbf{\Sigma}\theta_0 + \big(nd\cdot v_{22}^2 + (n+1)^2\big)\sigma^2\Big).$$

Moreover, we have

$$\lim_{n\to\infty}\mathbb{E}_{\mathbf{Z}_{[n]},\mathbf{x},y|\theta_0}\left[\left(\hat{y}_{\mathrm{FS}}(\mathbf{Z}_{[n]},\mathbf{x};\mathbf{V},\mathbf{Q}) - y\right)^2\right] = \lim_{n\to\infty}\frac{1}{n}h(\mathbf{V},\mathbf{Q};\theta_0)$$

$$= \mathbf{v}_{21}^\top\mathbf{\Sigma}\mathbf{v}_{21} + 2\big(v_{22}-1\big)\mathbf{v}_{21}^\top\mathbf{\Sigma}\theta_0 + (v_{22}-1)^2\theta_0^\top\mathbf{\Sigma}\theta_0 + \sigma^2$$

$$= \big(\mathbf{v}_{21} + (v_{22}-1)\theta_0\big)^\top\mathbf{\Sigma}\big(\mathbf{v}_{21} + (v_{22}-1)\theta_0\big) + \sigma^2.$$

$\square$

### A.3. Proofs for Optimal Parameters

**Corollary A.7** (Optimal Parameters of Pretrained Models; Corollary of Theorem 1 in Ahn et al. (2023))**.** *Suppose that* $\mathbf{V}$ *and* $\mathbf{Q}$ *in* (1) *satisfy* $q \geq 0$ *and that* $\mathbf{Q}_{11}$ *is positive definite. Consider the loss* $\mathcal{L}(\mathbf{V}, \mathbf{Q})$ *in* (4) *with context length* $m \in \mathbb{N}$. *Then, for sufficiently large* $m$, *the following parameters globally minimize the loss:*

$$\mathbf{V} = \begin{bmatrix} 0 & \mathbf{0}_d^\top & 0 \\ \mathbf{0}_d & \mathbf{0}_d\mathbf{0}_d^\top & \mathbf{0}_d \\ 0 & \mathbf{0}_d^\top & \frac{m}{m+1+d} \end{bmatrix}, \qquad \mathbf{Q} = \begin{bmatrix} 0 & \mathbf{0}_d^\top & 0 \\ \mathbf{0}_d & \mathbf{U} \cdot \mathrm{diag}\left(\frac{m+1+d}{(m+1)\lambda_i + \mathrm{tr}(\boldsymbol{\Sigma})}\right)_{i \in [d]} \cdot \mathbf{U}^\top & \mathbf{0}_d \\ 0 & \mathbf{0}_d^\top & 0 \end{bmatrix}. \tag{24}$$

*Moreover,* $\mathbf{U} \, \mathrm{diag}\left(\frac{m+1+d}{(m+1)\lambda_i + \mathrm{tr}(\boldsymbol{\Sigma})}\right)_{i \in [d]} \mathbf{U}^\top \to \boldsymbol{\Sigma}^{-1}$ *as* $m \to \infty$.

*Proof.* By the definition of the linear attention model in (1), for any input $\mathbf{Z}$, the prediction $f(\mathbf{Z}; \mathbf{V}, \mathbf{Q})$ produced by the parameters in (24) is identical to that produced by the rescaled parameter pair

$$\mathbf{V} = \begin{bmatrix} 0 & \mathbf{0}_d^\top & 0 \\ \mathbf{0}_d & \mathbf{0}_d\mathbf{0}_d^\top & \mathbf{0}_d \\ 0 & \mathbf{0}_d^\top & 1 \end{bmatrix}, \qquad \mathbf{Q} = \begin{bmatrix} 0 & \mathbf{0}_d^\top & 0 \\ \mathbf{0}_d & \mathbf{U} \cdot \mathrm{diag}\left(\frac{m}{(m+1)\lambda_i + \mathrm{tr}(\boldsymbol{\Sigma})}\right)_{i \in [d]} \cdot \mathbf{U}^\top & \mathbf{0}_d \\ 0 & \mathbf{0}_d^\top & 0 \end{bmatrix}.$$

It therefore suffices to show that this rescaled pair is a global minimizer of the loss.

From Lemma A.5, we derived errors for a fixed $\theta_0$. In the pretrained setting, $\theta$ is drawn from $\mathcal{N}(\mathbf{0}_d, \mathbf{I}_d)$. We compute the squared error for random $\theta$ by applying the law of total expectation. From (19) in Lemma A.5,

$$\mathcal{L}(\mathbf{V}, \mathbf{Q}) = \mathbb{E}_\theta\left[\mathbb{E}_{\mathbf{Z}_{[m]}, \mathbf{x}, y | \theta}\left[\left(\hat{y}_{\mathrm{FS}}(\mathbf{Z}_{[m]}, \mathbf{x}; \mathbf{V}, \mathbf{Q}) - y\right)^2\right]\right]$$

$$= \frac{1}{(m+1)^2}\mathbb{E}_\theta\Big[c(\theta; \mathbf{V}, \mathbf{Q}, \boldsymbol{\Sigma}, \sigma^2) + q^2 \cdot \mathbf{v}_{21}^\top\boldsymbol{\Sigma}\mathbf{v}_{21} + 2q \cdot \mathrm{tr}(\mathbf{Q}_{11}\boldsymbol{\Sigma}) \cdot \mathbf{v}_{21}^\top\boldsymbol{\Sigma}\mathbf{v}_{21} + 4q \cdot \mathbf{v}_{21}^\top\boldsymbol{\Sigma}\mathbf{Q}_{11}\boldsymbol{\Sigma}\mathbf{v}_{21}$$

$$+ mq^2 \cdot \left((\mathbf{v}_{21}^\top + \theta^\top)\boldsymbol{\Sigma}(\mathbf{v}_{21} + \theta) + \sigma^2\right) + 2mq \cdot \left(\mathbf{v}_{21}^\top\boldsymbol{\Sigma}\mathbf{Q}_{11} + \theta^\top\boldsymbol{\Sigma}\mathbf{Q}_{11} - \theta^\top\right)\boldsymbol{\Sigma}\mathbf{v}_{21}\Big],$$

where $c(\theta; \mathbf{V}, \mathbf{Q}, \boldsymbol{\Sigma}, \sigma^2)$ collects all terms independent of the parameter $q$. Then,

$$\mathcal{L}(\mathbf{V}, \mathbf{Q}) = \frac{1}{(m+1)^2}\mathbb{E}_\theta\Big[c(\theta; \mathbf{V}, \mathbf{Q}, \boldsymbol{\Sigma}, \sigma^2)\Big] + \frac{q}{(m+1)^2}\Big((m+1)q \cdot \mathbf{v}_{21}^\top\boldsymbol{\Sigma}\mathbf{v}_{21} + 2 \cdot \mathrm{tr}(\mathbf{Q}_{11}\boldsymbol{\Sigma}) \cdot \mathbf{v}_{21}^\top\boldsymbol{\Sigma}\mathbf{v}_{21}$$

$$+ mq \cdot \mathrm{tr}(\boldsymbol{\Sigma}) + mq \cdot \sigma^2 + 2(m+2) \cdot \mathbf{v}_{21}^\top\boldsymbol{\Sigma}\mathbf{Q}_{11}\boldsymbol{\Sigma}\mathbf{v}_{21}\Big).$$

Since $\boldsymbol{\Sigma}$ and $\mathbf{Q}_{11}$ are positive definite, the expression in the parentheses of the last term is nonnegative. Therefore, the loss is minimized by setting $q = 0$.

When $q = 0$ and $m$ is sufficiently large, the prediction reduces to that of Theorem 1 in Ahn et al. (2023), because

$$\hat{y}_{\mathrm{FS}}(\mathbf{Z}_m, \mathbf{x}; \mathbf{V}, \mathbf{Q}) \approx \hat{y}_{\mathrm{Net\text{-}FS}}(\mathbf{Z}_m, \mathbf{x}; \mathbf{V}, \mathbf{Q}) = \begin{bmatrix} \mathbf{v}_{21} \\ v_{22} \end{bmatrix}^\top \left(\frac{1}{m}\sum_{i \in [m]}\begin{bmatrix} \mathbf{x}_i \\ y_i \end{bmatrix}\begin{bmatrix} \mathbf{x}_i \\ y_i \end{bmatrix}^\top\right)\begin{bmatrix} \mathbf{Q}_{11} \\ \mathbf{q}_{21}^\top \end{bmatrix}\mathbf{x}_{n+1}.$$

By Theorem 1 in Ahn et al. (2023), the following parameters globally minimize the loss:

$$\begin{bmatrix} \mathbf{V}_{11} & \mathbf{v}_{12} \\ \mathbf{v}_{21}^\top & v_{22} \end{bmatrix} = \begin{bmatrix} \mathbf{0}_{d \times d} & \mathbf{0}_d \\ \mathbf{0}_d^\top & 1 \end{bmatrix}, \qquad \begin{bmatrix} \mathbf{Q}_{11} & \mathbf{q}_{12} \\ \mathbf{q}_{21}^\top & q_{22} \end{bmatrix} = \begin{bmatrix} \mathbf{U} \cdot \mathrm{diag}\left(\frac{m}{(m+1)\lambda_i + \mathrm{tr}(\boldsymbol{\Sigma})}\right)_{i \in [d]} \cdot \mathbf{U}^\top & \mathbf{0}_d \\ \mathbf{0}_d^\top & 0 \end{bmatrix}.$$

$\square$

**Theorem A.8** (Optimal Parameters of Fully Fine-Tuned Models)**.** *Given a target task* $\theta_0 \in \mathbb{R}^d$, *consider the zero-shot prompt loss* $\mathcal{L}_{\mathrm{ZS}}(\mathbf{V}, \mathbf{Q}; \theta_0)$ *in* (5). *Then, for any* $w > 0$, *the following parameters globally minimize the loss:*

$$\mathbf{V} = \begin{bmatrix} 0 & \mathbf{0}_d^\top & 0 \\ \mathbf{0}_d & \mathbf{0}_d\mathbf{0}_d^\top & \mathbf{0}_d \\ 0 & w\theta_0^\top & 1 \end{bmatrix}, \qquad \mathbf{Q} = \begin{bmatrix} 1/w & \mathbf{0}_d^\top & 0 \\ \mathbf{0}_d & \mathbf{0}_d\mathbf{0}_d^\top & \mathbf{0}_d \\ 0 & \mathbf{0}_d^\top & 0 \end{bmatrix}. \tag{25}$$

*Proof.* From Lemma A.5, we have

$$\mathbb{E}_{\mathbf{x},y|\theta_0}\left[\left(\hat{y}_{\mathrm{ZS}}(\mathbf{x};\mathbf{V},\mathbf{Q})-y\right)^2\right]=\theta_0^\top\mathbf{\Sigma}\theta_0+\sigma^2+\frac{1}{w^2}\cdot w\theta_0^\top\mathbf{\Sigma}w\theta_0-\frac{2}{w}\cdot w\theta_0^\top\mathbf{\Sigma}\theta_0=\sigma^2.$$

This implies that the above parameters attain the global minimum, since $\sigma^2=\mathbb{E}[\mathrm{Var}(y\mid\mathbf{x})]$ is the minimum achievable expected squared loss. $\square$

**Theorem A.9** (Optimal Parameters and Test Error of Value-Matrix Fine-Tuned Models)**.** *Given a target task $\theta_0\in\mathbb{R}^d$, consider the zero-shot prompt loss $\mathcal{L}_{\mathrm{ZS}}(\mathbf{V},\mathbf{Q};\theta_0)$ in* (5)*. For any $w>0$, define*

$$\hat{\mathbf{V}}(w)=\begin{bmatrix}0&\mathbf{0}_d^\top&0\\\mathbf{0}_d&\mathbf{0}_d\mathbf{0}_d^\top&\mathbf{0}_d\\0&\frac{1}{d+4}\theta_0^\top&w\end{bmatrix},\qquad\hat{\mathbf{Q}}=\begin{bmatrix}0&\mathbf{0}_d^\top&0\\\mathbf{0}_d&\mathbf{\Sigma}^{-1}&\mathbf{0}_d\\0&\mathbf{0}_d^\top&0\end{bmatrix}.\tag{26}$$

*Then,*

$$\left\{\begin{bmatrix}0&\mathbf{0}_d^\top&0\\\mathbf{0}_d&\mathbf{V}_{11}&\mathbf{v}_{12}\\0&\frac{1}{d+4}\theta_0^\top&w\end{bmatrix}\middle|\mathbf{V}_{11}\in\mathbb{R}^{d\times d},\mathbf{v}_{12}\in\mathbb{R}^d,w\neq 0\right\}=\arg\min_\mathbf{V}\mathcal{L}_{\mathrm{ZS}}(\mathbf{V},\hat{\mathbf{Q}};\theta_0).$$

*In particular, $\hat{\mathbf{V}}(w)$ minimizes the zero-shot loss for fixed $\hat{\mathbf{Q}}$, i.e.,*

$$\hat{\mathbf{V}}(w)\in\arg\min_\mathbf{V}\mathcal{L}_{\mathrm{ZS}}(\mathbf{V},\hat{\mathbf{Q}};\theta_0),$$

*and the minimum value is $\sigma^2+\frac{2}{d+4}\theta_0^\top\mathbf{\Sigma}\theta_0$. Under this choice of parameters, for any $\theta\in\mathbb{R}^d$,*

$$\mathcal{E}\left(\hat{\mathbf{V}}(w),\hat{\mathbf{Q}},n;\theta\right)=\frac{(d+2)(d+4)+n(n+5+3d)}{(d+4)^2(n+1)^2}\theta_0^\top\mathbf{\Sigma}\theta_0+\frac{2n(n+3+2d)w-2n(n+3+d)-2d+4}{(d+4)(n+1)^2}\theta_0^\top\mathbf{\Sigma}\theta$$

$$+\frac{n(n+1+d)w^2-2n(n+1)w}{(n+1)^2}\theta^\top\mathbf{\Sigma}\theta+\frac{nd\,w^2}{(n+1)^2}\sigma^2+\theta^\top\mathbf{\Sigma}\theta+\sigma^2,\tag{27}$$

$$\lim_{n\to\infty}\mathcal{E}\left(\hat{\mathbf{V}}(w),\hat{\mathbf{Q}},n;\theta\right)=\left(\frac{1}{d+4}\theta_0+(w-1)\theta\right)^\top\mathbf{\Sigma}\left(\frac{1}{d+4}\theta_0+(w-1)\theta\right)+\sigma^2.\tag{28}$$

*Proof.* From Lemma A.6, we have

$$\mathbb{E}_{\mathbf{x},y|\theta_0}\left[\left(\hat{y}_{\mathrm{ZS}}(\mathbf{x};\mathbf{V},\hat{\mathbf{Q}})-y\right)^2\right]=(d+2)(d+4)\mathbf{v}_{21}^\top\mathbf{\Sigma}\mathbf{v}_{21}-2(d+2)\mathbf{v}_{21}^\top\mathbf{\Sigma}\theta_0+\theta_0^\top\mathbf{\Sigma}\theta_0+\sigma^2,$$

Differentiating with respect to $\mathbf{v}_{21}$ and setting the gradient to zero gives

$$2(d+2)(d+4)\mathbf{\Sigma}\mathbf{v}_{21}-2(d+2)\mathbf{\Sigma}\theta_0=\mathbf{0},$$

which is unique since $\mathbf{\Sigma}$ is positive definite. Substituting back gives the minimum value $\frac{2}{d+4}\theta_0^\top\mathbf{\Sigma}\theta_0+\sigma^2$. This proves the minimizer set and, in particular, that $\hat{\mathbf{V}}(w)$ is a minimizer.

Moreover, from Lemma A.6, for any $\theta\in\mathbb{R}^d$, we have

$$\mathbb{E}_{\mathbf{Z}_{[n]},\mathbf{x},y|\theta}\left[\left(\hat{y}_{\mathrm{FS}}(\mathbf{Z}_{[n]},\mathbf{x};\hat{\mathbf{V}}(w),\hat{\mathbf{Q}})-y\right)^2\right]$$

$$=\frac{1}{(n+1)^2}\left(\frac{(d+2)(d+4)+n(n+5+3d)}{(d+4)^2}\theta_0^\top\mathbf{\Sigma}\theta_0+\frac{2\Big(n(n+3+2d)\,w-\big(n(n+3+d)+d-2\big)\Big)}{d+4}\theta_0^\top\mathbf{\Sigma}\theta\right.$$

$$\left.+\left(n(n+1+d)w^2-2n(n+1)w+(n+1)^2\right)\theta^\top\mathbf{\Sigma}\theta+\left(nd\,w^2+(n+1)^2\right)\sigma^2\right)$$

$$=\frac{(d+2)(d+4)+n(n+5+3d)}{(d+4)^2(n+1)^2}\theta_0^\top\mathbf{\Sigma}\theta_0+\frac{2n(n+3+2d)w-2n(n+3+d)-2d+4}{(d+4)(n+1)^2}\theta_0^\top\mathbf{\Sigma}\theta$$

$$+\frac{n(n+1+d)w^2-2n(n+1)w}{(n+1)^2}\theta^\top\mathbf{\Sigma}\theta+\frac{nd\,w^2}{(n+1)^2}\sigma^2+\theta^\top\mathbf{\Sigma}\theta+\sigma^2.$$

Therefore,

$$\lim_{n\to\infty} \mathbb{E}_{\mathbf{Z}_{[n]},\mathbf{x},y|\theta}\left[\left(\hat{y}_{\text{FS}}(\mathbf{Z}_{[n]},\mathbf{x};\hat{\mathbf{V}}(w),\hat{\mathbf{Q}})-y\right)^2\right] = \left(\frac{1}{d+4}\theta_0 + (w-1)\theta\right)^\top \boldsymbol{\Sigma}\left(\frac{1}{d+4}\theta_0 + (w-1)\theta\right) + \sigma^2.$$

$\square$

**Theorem A.10** (Task-Wise Optimal $w$). *Given a target task $\theta_0 \in \mathbb{R}^d$, consider the family of parameters $\left(\hat{\mathbf{V}}(w),\hat{\mathbf{Q}}\right)$ in (26). For any $n \in \mathbb{N}$ and any $\theta \in \mathbb{R}^d$, the few-shot test error on the task $\theta$ is minimized at*

$$w^\star(n;\theta) := \arg\min_w \mathcal{E}\left(\hat{\mathbf{V}}(w),\hat{\mathbf{Q}},n;\theta\right) = \frac{(n+1)\theta^\top\boldsymbol{\Sigma}\theta - \frac{n+3+2d}{d+4}\theta_0^\top\boldsymbol{\Sigma}\theta}{(n+1+d)\theta^\top\boldsymbol{\Sigma}\theta + d\sigma^2}.$$

*Then $w^\star(n;\theta) \to \frac{d+3}{d+4}$ as $n \to \infty$. Hence, for any $\theta \in \mathbb{R}^d$,*

$$\lim_{n\to\infty} \mathcal{E}\left(\hat{\mathbf{V}}(\tfrac{d+3}{d+4}),\hat{\mathbf{Q}},n;\theta\right) = \sigma^2 + \frac{1}{(d+4)^2}(\theta-\theta_0)^\top\boldsymbol{\Sigma}(\theta-\theta_0).$$

**Corollary A.11** (Task-Averaged Optimal $w$). *Given a target task $\theta_0 \in \mathbb{R}^d$, consider the family of parameters $\left(\hat{\mathbf{V}}(w),\hat{\mathbf{Q}}\right)$ in (11). For any $n \in \mathbb{N}$, the few-shot test error averaged over tasks $\theta \sim \mathcal{N}(\mathbf{0}_d,\mathbf{I}_d)$ is minimized at*

$$w^\star(n) := \arg\min_w \mathbb{E}_\theta\left[\mathcal{E}\left(\hat{\mathbf{V}}(w),\hat{\mathbf{Q}},n;\theta\right)\right] = \frac{(n+1)\operatorname{tr}(\boldsymbol{\Sigma})}{(n+1+d)\operatorname{tr}(\boldsymbol{\Sigma}) + d\sigma^2}.$$

*Then $w^\star(n) \to 1$ as $n \to \infty$. Hence, for any $\theta \in \mathbb{R}^d$,*

$$\lim_{n\to\infty} \mathcal{E}\left(\hat{\mathbf{V}}(1),\hat{\mathbf{Q}},n;\theta\right) = \sigma^2 + \frac{1}{(d+4)^2}\theta_0^\top\boldsymbol{\Sigma}\theta_0.$$

*Proof of Theorem A.10 and Corollary A.11.* Fix $\theta \in \mathbb{R}^d$. Collecting the terms in (27) that depend on $w$, we obtain

$$\mathbb{E}_{\mathbf{Z}_{[n]},\mathbf{x},y|\theta}\left[\left(\hat{y}_{\text{FS}}(\mathbf{Z}_{[n]},\mathbf{x};\hat{\mathbf{V}}(w),\hat{\mathbf{Q}})-y\right)^2\right]$$

$$= \frac{2n(n+3+2d)w}{(d+4)(n+1)^2}\theta_0^\top\boldsymbol{\Sigma}\theta + \frac{n(n+1+d)w^2 - 2n(n+1)w}{(n+1)^2}\theta^\top\boldsymbol{\Sigma}\theta + \frac{nd\,w^2}{(n+1)^2}\sigma^2 + \text{const}(\theta)$$

$$= \frac{n}{(n+1)^2}\left(\left((n+1+d)\theta^\top\boldsymbol{\Sigma}\theta + d\sigma^2\right)w^2 + 2\left(\frac{n+3+2d}{d+4}\theta_0^\top\boldsymbol{\Sigma}\theta - (n+1)\theta^\top\boldsymbol{\Sigma}\theta\right)w\right) + \text{const}(\theta).$$

Since $n \in \mathbb{N}$ and $\sigma^2 \geq 0$, we have $(n+1+d)\theta^\top\boldsymbol{\Sigma}\theta + d\sigma^2 > 0$, and therefore the above is strictly convex in $w$ and has a unique minimizer given by the first-order condition, which is

$$\left((n+1+d)\theta^\top\boldsymbol{\Sigma}\theta + d\sigma^2\right)w + \frac{n+3+2d}{d+4}\theta_0^\top\boldsymbol{\Sigma}\theta - (n+1)\theta^\top\boldsymbol{\Sigma}\theta = 0$$

Therefore,

$$w^\star(n;\theta) := \arg\min_w \mathcal{L}_{\text{FS}}\left(\hat{\mathbf{V}}(w),\hat{\mathbf{Q}},n;\theta\right) = \frac{(n+1)\,\theta^\top\boldsymbol{\Sigma}\theta - \frac{n+3+2d}{d+4}\theta_0^\top\boldsymbol{\Sigma}\theta}{(n+1+d)\,\theta^\top\boldsymbol{\Sigma}\theta + d\sigma^2}.$$

Next, taking expectation over $\theta$, we obtain

$$\mathbb{E}_\theta\left[\mathbb{E}_{\mathbf{Z}_{[n]},\mathbf{x},y|\theta}\left[\left(\hat{y}_{\text{FS}}(\mathbf{Z}_{[n]},\mathbf{x};\hat{\mathbf{V}}(w),\hat{\mathbf{Q}})-y\right)^2\right]\right]$$

$$= \frac{n}{(n+1)^2}\left(\left((n+1+d)\operatorname{tr}(\boldsymbol{\Sigma}) + d\sigma^2\right)w^2 - 2(n+1)\operatorname{tr}(\boldsymbol{\Sigma})w\right) + \text{const}(\theta),$$

and hence

$$w^\star(n) := \arg\min_w \mathbb{E}_\theta\left[\mathcal{L}_{\text{FS}}\left(\hat{\mathbf{V}}(w),\hat{\mathbf{Q}},n;\theta\right)\right] = \frac{(n+1)\operatorname{tr}(\boldsymbol{\Sigma})}{(n+1+d)\operatorname{tr}(\boldsymbol{\Sigma}) + d\sigma^2}.$$

The stated limits follow from (28) in Theorem A.9 by letting $n \to \infty$. $\square$

**Proposition A.12.** *Assume* $\Sigma = \mathbf{I}_d$, *and let* $\hat{\mathbf{V}}$ *and* $\hat{\mathbf{Q}}$ *be the parameters in* (6) *which minimize the loss* $\mathcal{L}(\mathbf{V}, \mathbf{Q})$ *in* (4). *Given a target task* $\theta_0 \in \mathbb{R}^d$, *consider* $\hat{\mathbf{V}}(w)$ *in* (11). *Then*

$$\hat{\mathbf{V}}\left(\tfrac{m}{m+1+d}\right) = \arg\min\left\{ \|\mathbf{V} - \hat{\mathbf{V}}\|_{\mathrm{F}}^2 \,\middle|\, \mathbf{V} \in \arg\min_{\mathbf{V}} \mathcal{L}_{\mathrm{ZS}}(\mathbf{V}, \hat{\mathbf{Q}}; \theta_0) \right\},$$

*where* $\|\cdot\|_{\mathrm{F}}$ *denotes the Frobenius norm.*

*Moreover, if* $m = n$, *then for sufficiently large* $n + d$, *the choice* $w = \frac{m}{m+1+d}$ *closely approximates the task-averaged optimum* $w^\star(n)$, *defined in Corollary A.10.*

*Proof.* Since $\Sigma = \mathbf{I}_d$, the parameters in (24) reduce to

$$\hat{\mathbf{V}} = \begin{bmatrix} 0 & \mathbf{0}_d^\top & 0 \\ \mathbf{0}_d & \mathbf{0}_d\mathbf{0}_d^\top & \mathbf{0}_d \\ 0 & \mathbf{0}_d^\top & \frac{m}{m+1+d} \end{bmatrix}, \qquad \hat{\mathbf{Q}} = \begin{bmatrix} 0 & \mathbf{0}_d^\top & 0 \\ \mathbf{0}_d & \mathbf{I}_d & \mathbf{0}_d \\ 0 & \mathbf{0}_d^\top & 0 \end{bmatrix}.$$

By Theorem A.9, we have

$$\mathbb{S} := \left\{ \begin{bmatrix} 0 & \mathbf{0}_d^\top & 0 \\ \mathbf{0}_d & \mathbf{V}_{11} & \mathbf{v}_{12} \\ 0 & \frac{1}{d+4}\theta_0^\top & w \end{bmatrix} \,\middle|\, \mathbf{V}_{11} \in \mathbb{R}^{d\times d}, \mathbf{v}_{12} \in \mathbb{R}^d, w \neq 0 \right\} = \arg\min_{\mathbf{V}} \mathbb{E}_{\mathbf{x},y|\theta_0}\left[ \left(\hat{y}_{\mathrm{ZS}}(\mathbf{x}; \mathbf{V}, \hat{\mathbf{Q}}) - y\right)^2 \right].$$

For any $\mathbf{V} \in \mathcal{S}$,

$$\left\|\mathbf{V} - \hat{\mathbf{V}}\right\|_{\mathrm{F}}^2 = \|\mathbf{V}_{11}\|_{\mathrm{F}} + \|\mathbf{v}_{12}\|_{\mathrm{F}} + \frac{1}{d+4}\|\theta_0\|_{\mathrm{F}} + \left(w - \frac{m}{m+1+d}\right)^2.$$

Therefore, the expression is minimized by $\mathbf{V}_{11} = \mathbf{0}_d\mathbf{0}_d^\top$, $\mathbf{v}_{12} = \mathbf{0}_d$, and $w = \frac{m}{m+1+d}$, yielding the minimizer $\hat{\mathbf{V}}\left(\frac{m}{m+1+d}\right)$.

Note that under $\Sigma = \mathbf{I}_d$ we have $w^\star(n) = \frac{n+1}{n+1+d+\sigma^2}$. Thus, if $m = n$, then

$$w^\star(n) - \frac{m}{m+1+d} = \frac{n+1}{n+1+d+\sigma^2} - \frac{n}{n+1+d} = \frac{n - n\sigma^2 + 1 + d}{(n+1+d)(n+1+d+\sigma^2)}.$$

This difference becomes small when $n + d$ is sufficiently large. $\qquad\square$

## A.4. Proofs for Test Errors

**Theorem A.13.** *Suppose $\mathbf{V}$ and $\mathbf{Q}$ in (1) satisfy $v_{22} = 1$, $\mathbf{q}_{21} = \mathbf{0}_d$, and $\mathbf{Q}_{11}$ is positive definite. For any $\theta \in \mathbb{R}^d$, the following inequality holds:*

$$\mathbb{E}_{\mathbf{x},y|\theta_0} \left[ \left( \hat{y}_{\mathrm{ZS}}(\mathbf{x}; \mathbf{V}, \mathbf{Q}) - y \right)^2 \right] \leq \mathbb{E}_{\mathbf{Z}_{[n]},\mathbf{x},y|\theta_0} \left[ \left( \hat{y}_{\mathrm{FS}}(\mathbf{Z}_{[n]}, \mathbf{x}; \mathbf{V}, \mathbf{Q}) - y \right)^2 \right], \tag{29}$$

*if and only if*

$$\left( \mathbf{v}_{21} - \mathbf{A}^{-1}\mathbf{B}\theta_0 \right)^\top \mathbf{A} \left( \mathbf{v}_{21} - \mathbf{A}^{-1}\mathbf{B}\theta_0 \right) \leq c, \tag{30}$$

*where*

$$\begin{aligned}
\mathbf{A} := {}& (n+2) \cdot \mathrm{tr}(\mathbf{Q}_{11}\mathbf{\Sigma})^2 \cdot \mathbf{\Sigma} + (2n+3) \cdot \mathrm{tr}(\mathbf{Q}_{11}\mathbf{\Sigma}\mathbf{Q}_{11}\mathbf{\Sigma}) \cdot \mathbf{\Sigma} + (4n+6) \cdot \mathrm{tr}(\mathbf{Q}_{11}\mathbf{\Sigma}) \cdot \mathbf{\Sigma}\mathbf{Q}_{11}\mathbf{\Sigma} \\
& + (7n+11) \cdot \mathbf{\Sigma}\mathbf{Q}_{11}\mathbf{\Sigma}\mathbf{Q}_{11}\mathbf{\Sigma} + q \cdot ((n+1)q \cdot \mathbf{\Sigma} + 2(n+2) \cdot \mathrm{tr}(\mathbf{Q}_{11}\mathbf{\Sigma}) \cdot \mathbf{\Sigma} + (4n+6) \cdot \mathbf{\Sigma}\mathbf{Q}_{11}\mathbf{\Sigma}), \\
\mathbf{B} := {}& \mathrm{tr}(\mathbf{Q}_{11}\mathbf{\Sigma}\mathbf{Q}_{11}\mathbf{\Sigma}) \cdot \mathbf{\Sigma} + (n+3) \cdot \mathbf{\Sigma}\mathbf{Q}_{11}\mathbf{\Sigma}\mathbf{Q}_{11}\mathbf{\Sigma} + \mathrm{tr}(\mathbf{Q}_{11}\mathbf{\Sigma}) \cdot \mathbf{\Sigma}\mathbf{Q}_{11}\mathbf{\Sigma} + (n+1) \cdot \mathbf{\Sigma}\mathbf{Q}_{11}\mathbf{\Sigma} \\
& + (n+1) \cdot \mathrm{tr}(\mathbf{Q}_{11}\mathbf{\Sigma}) \cdot \mathbf{\Sigma} + q \left( \mathbf{\Sigma}\mathbf{Q}_{11}\mathbf{\Sigma} + (q-1) \cdot \mathbf{\Sigma} \right), \\
\mathbf{C} := {}& (n+1) \cdot \mathbf{\Sigma}\mathbf{Q}_{11}\mathbf{\Sigma}\mathbf{Q}_{11}\mathbf{\Sigma} - 2(n+1) \cdot \mathbf{\Sigma}\mathbf{Q}_{11}\mathbf{\Sigma} + \mathrm{tr}(\mathbf{Q}_{11}\mathbf{\Sigma}\mathbf{Q}_{11}\mathbf{\Sigma}) \cdot \mathbf{\Sigma} + q^2 \cdot \mathbf{\Sigma} \\
c := {}& \theta_0^\top \left( \mathbf{B}\mathbf{A}^{-1}\mathbf{B} + \mathbf{C} \right) \theta_0 + \left( \mathrm{tr}(\mathbf{Q}_{11}\mathbf{\Sigma}\mathbf{Q}_{11}\mathbf{\Sigma}) + q^2 \right) \cdot \sigma^2.
\end{aligned}$$

*Proof.* From Lemma A.5, we have

$$\begin{aligned}
& \mathbb{E}_{\mathbf{Z}_{[n]},\mathbf{x},y|\theta_0} \left[ \left( \hat{y}_{\mathrm{FS}}(\mathbf{Z}_{[n]}, \mathbf{x}; \mathbf{V}, \mathbf{Q}) - y \right)^2 \right] - \mathbb{E}_{\mathbf{x},y|\theta_0} \left[ \left( \hat{y}_{\mathrm{ZS}}(\mathbf{x}; \mathbf{V}, \mathbf{Q}) - y \right)^2 \right] \\
& = \frac{n}{(n+1)^2} \cdot \left( h(\mathbf{V}, \mathbf{Q}; \theta_0) - (n+2) \cdot \mathbb{E}_{\mathbf{x},y|\theta_0} \left[ \left( \hat{y}_{\mathrm{ZS}}(\mathbf{x}; \mathbf{V}, \mathbf{Q}) - y \right)^2 \right] \right),
\end{aligned} \tag{31}$$

which implies that it suffices to check the sign of the expression inside the parentheses in (31). Then,

$$\begin{aligned}
& h(\mathbf{V}, \mathbf{Q}; \theta_0) - (n+2) \cdot \mathbb{E}_{\mathbf{x},y|\theta_0} \left[ \left( \hat{y}_{\mathrm{ZS}}(\mathbf{x}; \mathbf{V}, \mathbf{Q}) - y \right)^2 \right] \\
& = \mathbf{v}_{21}^\top \left( \mathrm{tr}(\mathbf{Q}_{11}\mathbf{\Sigma}\mathbf{Q}_{11}\mathbf{\Sigma}) \cdot \mathbf{\Sigma} + (n+5) \cdot \mathbf{\Sigma}\mathbf{Q}_{11}\mathbf{\Sigma}\mathbf{Q}_{11}\mathbf{\Sigma} + 2\,\mathrm{tr}(\mathbf{Q}_{11}\mathbf{\Sigma}) \cdot \mathbf{\Sigma}\mathbf{Q}_{11}\mathbf{\Sigma} \right) \mathbf{v}_{21} \\
& \quad + 2\mathbf{v}_{21}^\top \left( \mathrm{tr}(\mathbf{Q}_{11}\mathbf{\Sigma}\mathbf{Q}_{11}\mathbf{\Sigma}) \cdot \mathbf{\Sigma} + (n+3) \cdot \mathbf{\Sigma}\mathbf{Q}_{11}\mathbf{\Sigma}\mathbf{Q}_{11}\mathbf{\Sigma} + \mathrm{tr}(\mathbf{Q}_{11}\mathbf{\Sigma}) \cdot \mathbf{\Sigma}\mathbf{Q}_{11}\mathbf{\Sigma} \right) \theta_0 \\
& \quad - 2\mathbf{v}_{21}^\top \left( (n+3) \cdot \mathbf{\Sigma}\mathbf{Q}_{11}\mathbf{\Sigma} + \mathrm{tr}(\mathbf{Q}_{11}\mathbf{\Sigma}) \cdot \mathbf{\Sigma} \right) \theta_0 \\
& \quad + (n+1) \cdot \theta_0^\top \mathbf{\Sigma}\mathbf{Q}_{11}\mathbf{\Sigma}\mathbf{Q}_{11}\mathbf{\Sigma}\theta_0 - 2(n+1) \cdot \theta_0^\top \mathbf{\Sigma}\mathbf{Q}_{11}\mathbf{\Sigma}\theta_0 + (\mathrm{tr}(\mathbf{Q}_{11}\mathbf{\Sigma}\mathbf{Q}_{11}\mathbf{\Sigma}) + n + 2)(\theta_0^\top \mathbf{\Sigma}\theta_0 + \sigma^2) \\
& \quad + q^2 \cdot \left( (\mathbf{v}_{21}^\top + \theta_0^\top)\mathbf{\Sigma}(\mathbf{v}_{21} + \theta_0) + \sigma^2 \right) + 2q \cdot \left( \mathbf{v}_{21}^\top \mathbf{\Sigma}\mathbf{Q}_{11} + \theta_0^\top \mathbf{\Sigma}\mathbf{Q}_{11} - \theta_0 \right) \mathbf{\Sigma}\mathbf{v}_{21} \\
& \quad - \mathbf{v}_{21}^\top \left( (n+2) \cdot \mathrm{tr}(\mathbf{Q}_{11}\mathbf{\Sigma})^2 \cdot \mathbf{\Sigma} + 2(n+2) \cdot \mathrm{tr}(\mathbf{Q}_{11}\mathbf{\Sigma}\mathbf{Q}_{11}\mathbf{\Sigma}) \cdot \mathbf{\Sigma} \right) \mathbf{v}_{21} \\
& \quad - \mathbf{v}_{21}^\top \left( 4(n+2) \cdot \mathrm{tr}(\mathbf{Q}_{11}\mathbf{\Sigma}) \cdot \mathbf{\Sigma}\mathbf{Q}_{11}\mathbf{\Sigma} + 8(n+2) \cdot \mathbf{\Sigma}\mathbf{Q}_{11}\mathbf{\Sigma}\mathbf{Q}_{11}\mathbf{\Sigma} \right) \mathbf{v}_{21} \\
& \quad + 2\mathbf{v}_{21}^\top \left( 2(n+2) \cdot \mathbf{\Sigma}\mathbf{Q}_{11}\mathbf{\Sigma} + (n+2) \cdot \mathrm{tr}(\mathbf{Q}_{11}\mathbf{\Sigma}) \cdot \mathbf{\Sigma} \right) \theta_0 - (n+2) \cdot (\theta_0^\top \mathbf{\Sigma}\theta_0 + \sigma^2) \\
& \quad - (n+2) \left( q^2 \cdot \mathbf{v}_{21}^\top \mathbf{\Sigma}\mathbf{v}_{21} + 2q \cdot \mathrm{tr}(\mathbf{Q}_{11}\mathbf{\Sigma}) \cdot \mathbf{v}_{21}^\top \mathbf{\Sigma}\mathbf{v}_{21} + 4q \cdot \mathbf{v}_{21}^\top \mathbf{\Sigma}\mathbf{Q}_{11}\mathbf{\Sigma}\mathbf{v}_{21} \right) \\
& = -\mathbf{v}_{21}^\top \mathbf{A}\mathbf{v}_{21} + 2\mathbf{v}_{21}^\top \mathbf{B}\theta_0 + \theta_0^\top \mathbf{C}\theta_0 + \left( \mathrm{tr}(\mathbf{Q}_{11}\mathbf{\Sigma}\mathbf{Q}_{11}\mathbf{\Sigma}) + q^2 \right) \cdot \sigma^2,
\end{aligned}$$

where $\mathbf{A}$, $\mathbf{B}$, and $\mathbf{C}$ are defined as in Eq. A.13.

Since $\mathbf{Q}_{11}$ is positive definite, it follows that $\mathbf{A}$ is also positive definite and hence invertible. Thus, we can rewrite the above expression as

$$-\mathbf{v}_{21}^\top \mathbf{A}\mathbf{v}_{21} + 2\mathbf{v}_{21}^\top \mathbf{B}\theta_0 + \theta_0^\top \mathbf{C}\theta_0 + \left( \mathrm{tr}(\mathbf{Q}_{11}\mathbf{\Sigma}\mathbf{Q}_{11}\mathbf{\Sigma}) + q^2 \right) \cdot \sigma^2 = - \left( \mathbf{v}_{21} - \mathbf{A}^{-1}\mathbf{B}\theta_0 \right)^\top \mathbf{A} \left( \mathbf{v}_{21} - \mathbf{A}^{-1}\mathbf{B}\theta_0 \right) + c,$$

where $c := \theta_0^\top \left( \mathbf{B}\mathbf{A}^{-1}\mathbf{B} + \mathbf{C} \right) \theta_0 + \left( \mathrm{tr}(\mathbf{Q}_{11}\boldsymbol{\Sigma}\mathbf{Q}_{11}\boldsymbol{\Sigma}) + q^2 \right) \cdot \sigma^2$. Therefore, the condition in (29) is equivalent to

$$\left( \mathbf{v}_{21} - \mathbf{A}^{-1}\mathbf{B}\theta_0 \right)^\top \mathbf{A} \left( \mathbf{v}_{21} - \mathbf{A}^{-1}\mathbf{B}\theta_0 \right) \leq c.$$

$\square$

**Corollary A.14** (Test Error of Pretrained Models). *Let $\hat{\mathbf{V}}$ and $\hat{\mathbf{Q}}$ be the parameters in (6) which minimize the loss $\mathcal{L}(\mathbf{V}, \mathbf{Q})$ in (4). Then, for any $\theta \in \mathbb{R}^d$, the following holds:*

$$\sigma^2 + \theta^\top \boldsymbol{\Sigma}\theta = \mathcal{E}(\hat{\mathbf{V}}, \hat{\mathbf{Q}}; 0, \theta) \geq \mathcal{E}(\hat{\mathbf{V}}, \hat{\mathbf{Q}}; n, \theta), \tag{32}$$

*if $n \in \mathbb{N}$ is sufficiently large such that*

$$n \geq \sum_{i \in [d]} a_i^2 \cdot \frac{\lambda_1 \|\theta\|_2^2 + \sigma^2}{a_d(2 - a_d) \cdot \lambda_1 \|\theta\|_2^2} - 1, \qquad \text{where } a_i := \frac{m\lambda_i}{(m+1)\lambda_i + \mathrm{tr}(\boldsymbol{\Sigma})}.$$

*Moreover, the $n$-shot error is monotone decreasing in $n$, and thus the minimum error is*

$$\lim_{n \to \infty} \mathcal{E}(\hat{\mathbf{V}}, \hat{\mathbf{Q}}; n, \theta) = \sigma^2 + \theta^\top \mathbf{U} \, \mathrm{diag}((a_i - 1)^2 \lambda_i)_{i \in [d]} \mathbf{U}^\top \theta,$$

*which converges to $\sigma^2$ as $m \to \infty$, since $a_i \to 1$ for all $i \in [d]$.*

*Proof.* From (24), which gives $\mathbf{v}_{21} = \mathbf{0}_d$ and $q = 0$, the condition (30) in Theorem A.13 simplifies to

$$\theta^\top \mathbf{B}\mathbf{A}^{-1}\mathbf{B}\theta \geq \theta^\top \left( \mathbf{B}\mathbf{A}^{-1}\mathbf{B} + \mathbf{C} \right) \theta + \mathrm{tr}(\mathbf{Q}_{11}\boldsymbol{\Sigma}\mathbf{Q}_{11}\boldsymbol{\Sigma}) \cdot \sigma^2.$$

Therefore, by Theorem A.13, (32) holds if and only if $\theta^\top \mathbf{C}\theta + \mathrm{tr}(\mathbf{Q}_{11}\boldsymbol{\Sigma}\mathbf{Q}_{11}\boldsymbol{\Sigma}) \cdot \sigma^2 \leq 0$.

Let $\mathbf{D} := \boldsymbol{\Sigma}^{1/2}\mathbf{Q}_{11}\boldsymbol{\Sigma}^{1/2}$, then

$$\begin{aligned}
\theta^\top \mathbf{C}\theta + \mathrm{tr}(\mathbf{Q}_{11}\boldsymbol{\Sigma}\mathbf{Q}_{11}\boldsymbol{\Sigma}) \cdot \sigma^2 &= (n+1) \cdot \left( \theta^\top \boldsymbol{\Sigma}\mathbf{Q}_{11}\boldsymbol{\Sigma}\mathbf{Q}_{11}\boldsymbol{\Sigma}\theta - 2 \cdot \theta^\top \boldsymbol{\Sigma}\mathbf{Q}_{11}\boldsymbol{\Sigma}\theta + \theta^\top \boldsymbol{\Sigma}\theta \right) \\
&\quad - (n+1) \cdot \theta^\top \boldsymbol{\Sigma}\theta + \mathrm{tr}(\mathbf{Q}_{11}\boldsymbol{\Sigma}\mathbf{Q}_{11}\boldsymbol{\Sigma}) \cdot \theta^\top \boldsymbol{\Sigma}\theta + \mathrm{tr}(\mathbf{Q}_{11}\boldsymbol{\Sigma}\mathbf{Q}_{11}\boldsymbol{\Sigma}) \cdot \sigma^2 \\
&= (n+1) \left\| (\mathbf{D} - \mathbf{I})\boldsymbol{\Sigma}^{1/2}\theta \right\|_2^2 + \left( \mathrm{tr}(\mathbf{D}^2) - (n+1) \right) \left\| \boldsymbol{\Sigma}^{1/2}\theta \right\|_2^2 + \mathrm{tr}(\mathbf{D}^2) \cdot \sigma^2 \\
&\leq \left( \mathrm{tr}(\mathbf{D}^2) + (n+1)(\delta - 1) \right) \left\| \boldsymbol{\Sigma}^{1/2}\theta \right\|_2^2 + \mathrm{tr}(\mathbf{D}^2) \cdot \sigma^2, \tag{33}
\end{aligned}$$

where $\delta$ denotes the maximum eigenvalue of $(\mathbf{D} - \mathbf{I})^2$, and the inequality follows from the Rayleigh-Ritz theorem. Furthermore as $\mathbf{Q}_{11}$ is given in (24), we have

$$\mathbf{D} = \mathbf{U}\,\mathrm{diag}\,(a_i)_{i \in [d]}\,\mathbf{U}^\top, \qquad \text{where } a_i := \frac{m\lambda_i}{(m+1)\lambda_i + \mathrm{tr}(\boldsymbol{\Sigma})}.$$

Then, we have $\mathrm{tr}(\mathbf{D}^2) = \sum_{i \in [d]} a_i^2$, and $\delta = \max_{i \in [d]}(1 - a_i)^2 = (1 - a_d)^2$, since $a_i$ is increasing with respect to $\lambda_i$ and $a_i < 1$ for all $i \in [d]$.

Let $n$ be sufficiently large such that $\left( \mathrm{tr}(\mathbf{D}^2) + (n+1)(\delta - 1) \right) < 0$, then

$$\left( \mathrm{tr}(\mathbf{D}^2) + (n+1)(\delta - 1) \right) \left\| \boldsymbol{\Sigma}^{1/2}\theta \right\|_2^2 \geq \left( \mathrm{tr}(\mathbf{D}^2) + (n+1)(\delta - 1) \right) \cdot \lambda_1 \|\theta\|_2^2 \tag{34}$$

holds from the Rayleigh-Ritz theorem. Therefore, if $n$ is sufficiently large such that

$$0 \geq \left( \mathrm{tr}(\mathbf{D}^2) + (n+1)(\delta - 1) \right)\lambda_1 \|\theta\|_2^2 + \mathrm{tr}(\mathbf{D}^2)\sigma^2 = -(n+1)a_d(2 - a_d)\lambda_1\|\theta\|_2^2 + \sum_{i \in [d]} a_i^2 \left( \lambda_1\|\theta\|_2^2 + \sigma^2 \right).$$

which is equal to

$$n \geq \sum_{i \in [d]} a_i^2 \cdot \frac{\lambda_1 \|\theta\|_2^2 + \sigma^2}{a_d(2 - a_d) \cdot \lambda_1 \|\theta\|_2^2} - 1,$$

then it follows $\theta^\top \mathbf{C}\theta + \mathrm{tr}(\mathbf{Q}_{11}\mathbf{\Sigma}\mathbf{Q}_{11}\mathbf{\Sigma}) \cdot \sigma^2 \leq 0$ from (33) and (34). Therefore, condition (32) also holds.

Moreover, because $\mathbf{V}$ and $\mathbf{Q}$ satisfy (24), it follows from Lemma A.5 that

$$\mathbb{E}_{\mathbf{Z}_{[n]}, \mathbf{x}, y|\theta} \left[ (\hat{y}_{\mathrm{FS}}(\mathbf{Z}_{[n]}, \mathbf{x}; \mathbf{V}, \mathbf{Q}) - y)^2 \right] = \frac{1}{(n+1)^2}(\theta_0^\top \mathbf{\Sigma}\theta_0 + \sigma^2) + \frac{n}{n+1}\theta^\top \mathbf{\Sigma}\mathbf{Q}_{11}\mathbf{\Sigma}\mathbf{Q}_{11}\mathbf{\Sigma}\theta$$

$$- \frac{n}{n+1}2\theta^\top \mathbf{\Sigma}\mathbf{Q}_{11}\mathbf{\Sigma}\theta + \frac{n}{(n+1)^2}(\mathrm{tr}(\mathbf{Q}_{11}\mathbf{\Sigma}\mathbf{Q}_{11}\mathbf{\Sigma}) + n + 2)(\theta^\top \mathbf{\Sigma}\theta + \sigma^2)$$

$$= \theta_0^\top \mathbf{\Sigma}\theta_0 + \sigma^2 + \frac{n}{(n+1)^2} \mathrm{tr}(\mathbf{Q}_{11}\mathbf{\Sigma}\mathbf{Q}_{11}\mathbf{\Sigma})(\theta^\top \mathbf{\Sigma}\theta + \sigma^2)$$

$$+ \frac{n}{n+1}\theta^\top \mathbf{U} \mathrm{diag}((a_i - 2)a_i\lambda_i)_{i \in [d]}\mathbf{U}^\top \theta,$$

which is monotone decreasing in $n$, since $(a_i - 2)a_i\lambda_i < 0$ for all $i \in [d]$. Therefore, the minimum error is

$$\lim_{n \to \infty} \mathbb{E}_{\mathbf{Z}_{[n]}, \mathbf{x}, y|\theta} \left[ (\hat{y}_{\mathrm{FS}}(\mathbf{Z}_{[n]}, \mathbf{x}; \mathbf{V}, \mathbf{Q}) - y)^2 \right] = \theta_0^\top \mathbf{\Sigma}\theta_0 + \sigma^2 + \theta^\top \mathbf{U} \mathrm{diag}((a_i - 2)a_i\lambda_i)_{i \in [d]}\mathbf{U}^\top \theta$$

$$= \sigma^2 + \theta^\top \mathbf{U} \mathrm{diag}((a_i - 1)^2\lambda_i)_{i \in [d]}\mathbf{U}^\top \theta.$$

$\square$

**Corollary A.15.** *Suppose that* $\mathbf{\Sigma} = \lambda \mathbf{I}_d$ *for some* $\lambda > 0$*, and that* $\mathbf{V}$ *and* $\mathbf{Q}$ *are given by the parameters in* (24)*. Let* $\theta_0 \in \mathbb{R}^d$*. Then*

$$\mathbb{E}_{\mathbf{Z}_{[n]}, \mathbf{x}, y|\theta_0} \left[ (\hat{y}_{\mathrm{FS}}(\mathbf{Z}_{[n]}, \mathbf{x}; \mathbf{V}, \mathbf{Q}) - y)^2 \right] \leq \mathbb{E}_{\mathbf{x}, y|\theta_0} \left[ (\hat{y}_{\mathrm{ZS}}(\mathbf{x}; \mathbf{V}, \mathbf{Q}) - y)^2 \right]$$

*holds* if and only if

$$n \geq \frac{da}{2 - a}\left(1 + \frac{\sigma^2}{\lambda\|\theta_0\|_2^2}\right) - 1, \qquad a := \frac{m}{m + 1 + d}. \tag{35}$$

*Moreover, as* $m \to \infty$*, the condition in* (35) *reduces to*

$$n \geq d\left(1 + \frac{\sigma^2}{\lambda\|\theta_0\|_2^2}\right) - 1.$$

*Proof.* Since $\mathbf{\Sigma} = \lambda \mathbf{I}_d$, all eigenvalues of $\mathbf{\Sigma}$ are equal and (24) gives

$$\mathbf{D} := \mathbf{\Sigma}^{1/2}\mathbf{Q}_{11}\mathbf{\Sigma}^{1/2} = a\mathbf{I}_d, \qquad a = \frac{m}{m + 1 + d}.$$

Hence $\mathrm{tr}(\mathbf{D}^2) = da^2$ and $(\mathbf{D} - \mathbf{I})^2 = (a - 1)^2\mathbf{I}_d$. From Theorem A.13, the condition to satisfy is

$$0 \geq \theta_0^\top \mathbf{C}\theta_0 + \mathrm{tr}(\mathbf{D}^2)\sigma^2$$

$$= (n+1)\|(\mathbf{D} - \mathbf{I})\mathbf{\Sigma}^{1/2}\theta_0\|_2^2 + \left(\mathrm{tr}(\mathbf{D}^2) - (n+1)\right)\|\mathbf{\Sigma}^{1/2}\theta_0\|_2^2 + \mathrm{tr}(\mathbf{D}^2)\sigma^2$$

$$= \left(-(n+1)a(2 - a) + da^2\right)\lambda\|\theta_0\|_2^2 + da^2\sigma^2.$$

Therefore,

$$(n+1)a(2 - a)\lambda\|\theta_0\|_2^2 \geq da^2\left(\lambda\|\theta_0\|_2^2 + \sigma^2\right),$$

which is equivalent to the necessary and sufficient condition in (35). Taking $m \to \infty$ completes the proof. $\square$

**Corollary A.16** (Test Error of Fully Fine-Tuned Models). *Given a target task $\theta_0 \in \mathbb{R}^d$, consider the family of parameters $(\hat{\mathbf{V}}(w), \hat{\mathbf{Q}}(w))$ in (9). Then, for any $n \in \mathbb{N}$ and any $w > 0$, the following inequality holds:*

$$\sigma^2 = \mathcal{E}(\hat{\mathbf{V}}(w), \hat{\mathbf{Q}}(w); 0, \theta_0) < \mathcal{E}(\hat{\mathbf{V}}(w), \hat{\mathbf{Q}}(w); n, \theta_0).$$

*Moreover, for any $\theta \in \mathbb{R}^d$, the test errors are given by*

$$\mathcal{E}(\hat{\mathbf{V}}(w), \hat{\mathbf{Q}}(w); 0, \theta) = \sigma^2 + (\theta - \theta_0)^\top \mathbf{\Sigma}(\theta - \theta_0),$$
$$\lim_{n \to \infty} \mathcal{E}(\hat{\mathbf{V}}(w), \hat{\mathbf{Q}}(w); n, \theta) = \sigma^2 + \theta^\top \mathbf{\Sigma}\theta.$$

*Proof.* From Lemma A.5, for any $\theta \in \mathbb{R}^d$, we have

$$\mathbb{E}_{\mathbf{Z}_{[n]}, \mathbf{x}, y|\theta}\left[\left(\hat{y}_{\mathrm{FS}}(\mathbf{Z}_{[n]}, \mathbf{x}; \mathbf{V}, \mathbf{Q}) - y\right)^2\right] = \frac{1}{(n+1)^2}\mathbb{E}_{\mathbf{x}, y|\theta}\left[\left(\hat{y}_{\mathrm{ZS}}(\mathbf{x}; \mathbf{V}, \mathbf{Q}) - y\right)^2\right] + \frac{n}{(n+1)^2}h(\mathbf{V}, \mathbf{Q}; \theta).$$

Plugging in the optimal zero-shot parameters in (25), i.e., $\mathbf{Q}_{11} = \mathbf{0}_d\mathbf{0}_d^\top$, $\mathbf{v}_{21} = c\theta_0$, and $q = 1/c$, gives

$$\mathbb{E}_{\mathbf{x}, y|\theta}\left[\left(\hat{y}_{\mathrm{ZS}}(\mathbf{x}; \mathbf{V}, \mathbf{Q}) - y\right)^2\right] = \theta^\top \mathbf{\Sigma}\theta + \sigma^2 + \theta_0^\top \mathbf{\Sigma}\theta_0 - 2\theta_0^\top \mathbf{\Sigma}\theta = (\theta - \theta_0)^\top \mathbf{\Sigma}(\theta - \theta_0) + \sigma^2.$$

In particular, $\mathbb{E}_{\mathbf{x}, y|\theta_0}\left[\left(\hat{y}_{\mathrm{ZS}}(\mathbf{x}; \mathbf{V}, \mathbf{Q}) - y\right)^2\right] = \sigma^2$.

Moreover, under the same parameter choice, we have

$$h(\mathbf{V}, \mathbf{Q}; \theta) = (n+2)(\theta^\top \mathbf{\Sigma}\theta + \sigma^2) + \frac{1}{c^2}\left((c\theta_0 + \theta)^\top \mathbf{\Sigma}(c\theta_0 + \theta) + \sigma^2\right) - 2\theta^\top \mathbf{\Sigma}\theta_0,$$

which implies

$$\mathbb{E}_{\mathbf{Z}_{[n]}, \mathbf{x}, y|\theta}\left[\left(\hat{y}_{\mathrm{FS}}(\mathbf{Z}_{[n]}, \mathbf{x}; \mathbf{V}, \mathbf{Q}) - y\right)^2\right] = \frac{1}{(n+1)^2}\left((\theta - \theta_0)^\top \mathbf{\Sigma}(\theta - \theta_0) + \sigma^2\right) + \frac{n(n+2)}{(n+1)^2}(\theta^\top \mathbf{\Sigma}\theta + \sigma^2)$$
$$+ \frac{n}{(n+1)^2}\left(\frac{1}{c^2}\left((c\theta_0 + \theta)^\top \mathbf{\Sigma}(c\theta_0 + \theta) + \sigma^2\right) - 2\theta^\top \mathbf{\Sigma}\theta_0\right). \quad (36)$$

If $\theta = \theta_0$, since $\mathbf{\Sigma}$ is positive definite, for any $n \in \mathbb{N}$

$$\mathbb{E}_{\mathbf{Z}_{[n]}, \mathbf{x}, y|\theta_0}\left[\left(\hat{y}_{\mathrm{FS}}(\mathbf{Z}_{[n]}, \mathbf{x}; \mathbf{V}, \mathbf{Q}) - y\right)^2\right] = \sigma^2 + \frac{n}{(n+1)^2}\left(n\theta_0^\top \mathbf{\Sigma}\theta_0 + \frac{1}{c^2}\left((c+1)^2\theta_0^\top \mathbf{\Sigma}\theta_0 + \sigma^2\right)\right)$$
$$= \sigma^2 + \frac{n}{(n+1)^2}\left(\left(n + \frac{(c+1)^2}{c^2}\right)\theta_0^\top \mathbf{\Sigma}\theta_0 + \frac{1}{c^2}\sigma^2\right)$$
$$> \mathbb{E}_{\mathbf{x}, y|\theta_0}\|\hat{y}_{\mathrm{ZS}}(\mathbf{x}; \mathbf{V}, \mathbf{Q}) - y\|_2^2.$$

Finally, taking the limit $n \to \infty$ in (36) gives

$$\lim_{n \to \infty}\mathbb{E}_{\mathbf{Z}_{[n]}, \mathbf{x}, y|\theta}\left[\left(\hat{y}_{\mathrm{FS}}(\mathbf{Z}_{[n]}, \mathbf{x}; \mathbf{V}, \mathbf{Q}) - y\right)^2\right] = \theta^\top \mathbf{\Sigma}\theta + \sigma^2.$$

$\square$

**Proposition A.17.** *Given a target task $\theta_0 \in \mathbb{R}^d$, consider the family of parameters $(\hat{\mathbf{V}}(w), \hat{\mathbf{Q}})$ in (11). Fix $n \in \mathbb{N}$. For simplicity, define $\mathcal{E}(w; \theta) := \mathcal{E}(\hat{\mathbf{V}}(w), \hat{\mathbf{Q}}; n, \theta)$ and $w^\star(\theta) := w^\star(n; \theta)$. For any $\theta \in \mathbb{R}^d$, the excess error is*

$$\mathcal{E}(w^\star(\theta_0); \theta) - \mathcal{E}(w^\star(\theta); \theta) = \frac{n((n+1+d)\theta^\top \mathbf{\Sigma}\theta + d\sigma^2)}{(n+1)^2} \cdot (w^\star(\theta_0) - w^\star(\theta))^2 \geq 0, \quad (37)$$

*with equality if and only if $w^\star(\theta) = w^\star(\theta_0)$. Moreover, if $\theta$ satisfies $\theta^\top \Sigma \theta = \theta_0^\top \Sigma \theta_0$, then* (37) *reduces to*

$$\mathcal{E}(w^\star(\theta_0); \theta) - \mathcal{E}(w^\star(\theta); \theta) = \frac{n(n+3+2d)^2}{(n+1)^2(d+4)^2} \cdot \frac{\left(\theta_0^\top \Sigma \theta_0\right)^2}{(n+1+d)\,\theta_0^\top \Sigma \theta_0 + d\sigma^2} \cdot \left(1 - \rho(\theta, \theta_0)\right)^2,$$

*where*

$$\rho(\theta, \theta_0) := \frac{\theta^\top \Sigma \theta_0}{\sqrt{(\theta^\top \Sigma \theta)(\theta_0^\top \Sigma \theta_0)}} \in [-1, 1]$$

*is the cosine similarity induced by the $\Sigma$-inner product.*

*Proof.* Fix $n \in \mathbb{N}$ and $\theta \in \mathbb{R}^d$. From the proof of Theorem A.10, the few-shot error is a quadratic function of $w$ of the form

$$\mathcal{E}(w; \theta) = \frac{1}{(n+1)^2}\left(a(\theta)\,w^2 + b(\theta)\,w\right) + \mathrm{const}(\theta) = \frac{a(\theta)}{(n+1)^2}\left(w^2 - \frac{b(\theta)}{a(\theta)}\,w\right) + \mathrm{const}(\theta),$$

where

$$a(\theta) := n\left((n+1+d)\,\theta^\top \Sigma \theta + d\sigma^2\right), \quad b(\theta) := 2n\left(\frac{n+3+2d}{d+4}\,\theta_0^\top \Sigma \theta - (n+1)\,\theta^\top \Sigma \theta\right).$$

Since $a(\theta) > 0$ and $w^\star(\theta) = -b(\theta)/(2a(\theta))$, we have

$$\mathcal{E}(w; \theta) = \frac{a(\theta)}{(n+1)^2}\left(w - w^\star(\theta)\right)^2 + \mathcal{E}(w^\star(\theta); \theta),$$

which implies

$$\mathcal{E}(w^\star(\theta_0); \theta) - \mathcal{E}(w^\star(\theta); \theta) = \frac{a(\theta)}{(n+1)^2}\left(w^\star(\theta_0) - w^\star(\theta)\right)^2 = \frac{n\left((n+1+d)\,\theta^\top \Sigma \theta + d\sigma^2\right)}{(n+1)^2}\left(w^\star(\theta_0) - w^\star(\theta)\right)^2 \geq 0,$$

where the equality condition is $w^\star(\theta) = w^\star(\theta_0)$.

Now assume $\theta^\top \Sigma \theta = \theta_0^\top \Sigma \theta_0$. From Theorem A.10, we have

$$w^\star(\theta_0) - w^\star(\theta) = \frac{\frac{n+3+2d}{d+4}\left(\theta_0^\top \Sigma \theta - \theta_0^\top \Sigma \theta_0\right)}{(n+1+d)\,\theta_0^\top \Sigma \theta_0 + d\sigma^2} = \frac{\frac{n+3+2d}{d+4}\,\theta_0^\top \Sigma \theta_0\left(\rho(\theta, \theta_0) - 1\right)}{(n+1+d)\,\theta_0^\top \Sigma \theta_0 + d\sigma^2},$$

and hence

$$\begin{aligned}
\mathcal{E}(w^\star(\theta_0); \theta) - \mathcal{E}(w^\star(\theta); \theta) &= \frac{n\left((n+1+d)\,\theta_0^\top \Sigma \theta_0 + d\sigma^2\right)}{(n+1)^2}\left(\frac{\frac{n+3+2d}{d+4}\,\theta_0^\top \Sigma \theta_0\left(\rho(\theta, \theta_0) - 1\right)}{(n+1+d)\,\theta_0^\top \Sigma \theta_0 + d\sigma^2}\right)^2 \\
&= \frac{n(n+3+2d)^2}{(n+1)^2(d+4)^2} \cdot \frac{\left(\theta_0^\top \Sigma \theta_0\right)^2}{(n+1+d)\,\theta_0^\top \Sigma \theta_0 + d\sigma^2} \cdot \left(1 - \rho(\theta, \theta_0)\right)^2.
\end{aligned}$$

$\square$

## A.5. Proofs for Test Errors under the Auxiliary Few-Shot Prompt Loss

To understand the role of the auxiliary few-shot objective during fine-tuning, we analyze the test error under a structured parameterization of the linear attention model in (1).

**Assumption A.18.** We assume that $m \to \infty$, $\mathbf{\Sigma} = \mathbf{I}_d$, and the parameters $\mathbf{V}$ and $\mathbf{Q}$ in (1) take the following forms:

$$\mathbf{V} = \begin{bmatrix} 0 & \mathbf{0}_d^\top & 0 \\ \mathbf{0}_d & \mathbf{0}_d\mathbf{0}_d^\top & \mathbf{0}_d \\ 0 & \alpha \cdot \theta_0^\top & 1 \end{bmatrix}, \qquad \mathbf{Q} = \begin{bmatrix} \beta & \mathbf{0}_d^\top & 0 \\ \mathbf{0}_d & \gamma \cdot \mathbf{I}_d & \mathbf{0}_d \\ 0 & \mathbf{0}_d^\top & 0 \end{bmatrix},$$

where $\alpha, \beta, \gamma \in \mathbb{R}$.

**Lemma A.19.** *Under Assumption A.18, we have*

$$\mathbb{E}_{\mathbf{Z}_{[n]}, \mathbf{x}, y | \theta_0} \left[ \left( \hat{y}_{\mathrm{FS}}(\mathbf{Z}_{[n]}, \mathbf{x}; \mathbf{V}, \mathbf{Q}) - y \right)^2 \right] = \tfrac{1}{(n+1)^2} \cdot \mathbb{E}_{\mathbf{x}, y | \theta_0} \left[ \left( \hat{y}_{\mathrm{ZS}}(\mathbf{x}; \mathbf{V}, \mathbf{Q}) - y \right)^2 \right] + \tfrac{n}{(n+1)^2} \cdot h(\mathbf{V}, \mathbf{Q}; \theta_0),$$

*where*

$$\mathbb{E}_{\mathbf{x}, y | \theta_0} \left[ \left( \hat{y}_{\mathrm{ZS}}(\mathbf{x}; \mathbf{V}, \mathbf{Q}) - y \right)^2 \right] = \|\theta_0\|_2^2 \cdot ((\beta + (d+2)\gamma)\alpha - 1)^2 + \|\theta_0\|_2^2 \cdot 2(d+2)\gamma^2\alpha^2 + \sigma^2, \tag{38}$$

$$h(\mathbf{V}, \mathbf{Q}; \theta_0) = \|\theta_0\|_2^2 \cdot \alpha^2 \Big( (3d+n+5)\gamma^2 + 2\beta\gamma + \beta^2 \Big) \tag{39}$$

$$+ \|\theta_0\|_2^2 \cdot 2\alpha \Big( (2d+n+3)\gamma^2 - (d+n+3)\gamma + \beta^2 + \beta\gamma - \beta \Big)$$

$$+ \|\theta_0\|_2^2 \Big( (d+n+1)\gamma^2 - 2(n+1)\gamma + (n+2) + \beta^2 \Big) + \sigma^2(d\gamma^2 + n + 2 + \beta^2).$$

*Moreover,* $\lim_{n \to \infty} \mathbb{E}_{\mathbf{Z}_{[n]}, \mathbf{x}, y | \theta_0} \left[ \left( \hat{y}_{\mathrm{FS}}(\mathbf{Z}_{[n]}, \mathbf{x}; \mathbf{V}, \mathbf{Q}) - y \right)^2 \right] = \|\theta_0\|_2^2 \cdot (\gamma\alpha + \gamma - 1)^2 + \sigma^2.$

*Proof.* Note that $\mathrm{tr}(\mathbf{Q}_{11}\mathbf{\Sigma}) = d\gamma$ and $\mathrm{tr}(\mathbf{Q}_{11}\mathbf{\Sigma}\mathbf{Q}_{11}\mathbf{\Sigma}) = d\gamma^2$. From Lemma A.5, we have (38) and (39). Therefore,

$$\lim_{n \to \infty} \mathbb{E}_{\mathbf{Z}_{[n]}, \mathbf{x}, y | \theta_0} \left[ \left( \hat{y}_{\mathrm{FS}}(\mathbf{Z}_{[n]}, \mathbf{x}; \mathbf{V}, \mathbf{Q}) - y \right)^2 \right] = 0 + \lim_{n \to \infty} \tfrac{1}{n} \cdot h(\mathbf{V}, \mathbf{Q}; \theta_0) = \|\theta_0\|_2^2 \cdot (\gamma\alpha + \gamma - 1)^2 + \sigma^2.$$

$\square$

**Theorem A.20.** *Under Assumption A.18, suppose the parameters are close to the optimal zero-shot solution in Theorem A.8. If one additional gradient descent step is performed on the asymptotic few-shot prompt loss in (14) with a sufficiently small learning rate $\eta$ and $n \to \infty$, then the zero-shot error increases.*

*Proof.* From (38) in Lemma A.19, define the zero-shot error as

$$\mathcal{E}(\alpha, \beta, \gamma) := \left( (\beta + (d+2)\gamma)\alpha - 1 \right)^2 \|\theta_0\|_2^2 + 2(d+2)\alpha^2\gamma^2 \|\theta_0\|_2^2 + \sigma^2.$$

From Lemma A.19, the asymptotic few-shot loss under Assumption A.18 is

$$\mathcal{L}(\alpha, \beta, \gamma) := \lim_{n \to \infty} \mathbb{E}_{\mathbf{Z}_{[n]}, \mathbf{x}, y | \theta_0} \left[ \left( \hat{y}_{\mathrm{FS}}(\mathbf{Z}_{[n]}, \mathbf{x}; \mathbf{V}, \mathbf{Q}) - y \right)^2 \right] = \|\theta_0\|_2^2 \cdot (\gamma\alpha + \gamma - 1)^2 + \sigma^2.$$

One gradient descent step with step size $\eta$ gives

$$\Delta\alpha = -\eta \frac{\partial \mathcal{L}}{\partial \alpha} = -2\eta\gamma(\gamma\alpha + \gamma - 1)\|\theta_0\|_2^2, \qquad \Delta\beta = 0, \qquad \Delta\gamma = -\eta \frac{\partial \mathcal{L}}{\partial \gamma} = -2\eta(\alpha + 1)(\gamma\alpha + \gamma - 1)\|\theta_0\|_2^2.$$

At the zero-shot optimum $(\alpha, \beta, \gamma) = (1, 1, 0)$, one gradient descent step gives $\Delta\alpha = 0$ and $\Delta\gamma = 4\eta\|\theta_0\|_2^2$, so $(\alpha, \beta, \gamma)$ is updated to $(1, 1, 4\eta\|\theta_0\|_2^2)$. Therefore,

$$\mathcal{E}(1, 1, 4\eta\|\theta_0\|_2^2) - \mathcal{E}(1, 1, 0) = (d+2)(d+4)\left(4\eta\|\theta_0\|_2^2\right)^2\|\theta_0\|_2^2 = 16(d+2)(d+4)\eta^2\|\theta_0\|_2^6 > 0,$$

and by continuity the same holds in a neighborhood of $(1, 1, 0)$. $\square$

# B. Experimental Details

We provide additional details on the experimental setup and supplementary results for the linear regression tasks in Section B.1 and the MMLU benchmark in Section B.2.

Code is available at `https://github.com/leechungpa/finetuning-without-forgetting-icl`.

## B.1. Experimental Setup and Additional Results for Linear Regression Tasks

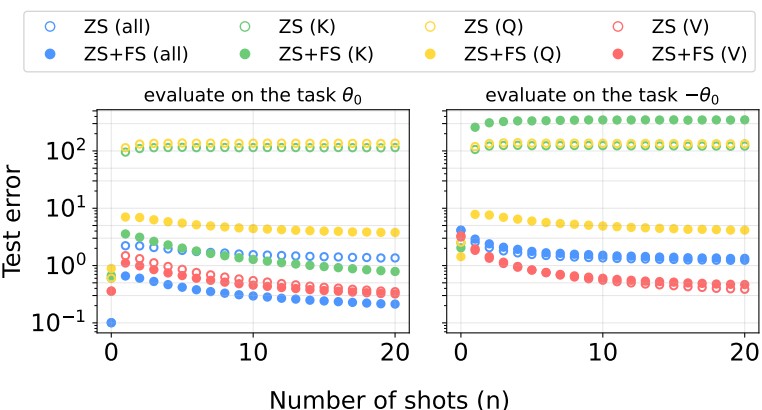

*Figure 7.* Test errors on *(Left)* the in-distribution task $\theta_0$ and *(Right)* the out-of-distribution task $-\theta_0$, using a 1-head attention model.

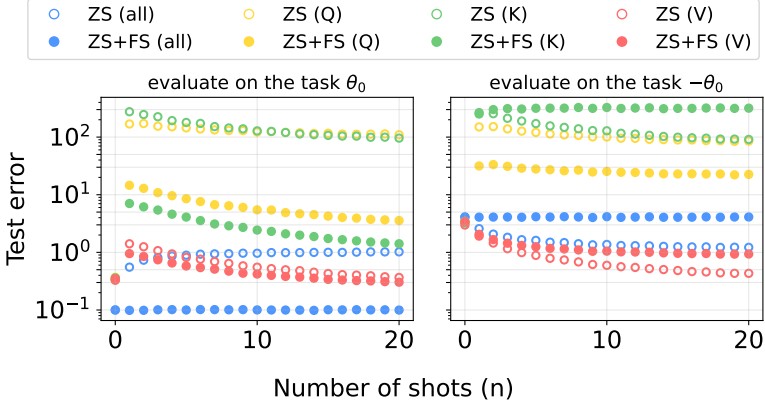

*Figure 8.* Test errors on *(Left)* the in-distribution task $\theta_0$ and *(Right)* the out-of-distribution task $-\theta_0$, using a 2-head attention model.

**Experimental Setup.** Following prior work (Ahn et al., 2023), we use the setup in Section 3 and set $d = 5$, $\sigma^2 = 0.1$, $\Sigma = \mathbf{I}_5$, $\|\theta_0\|_2^2 = 1$, and $m = n = 20$. In the pretraining phase, we generate data by sampling $\theta \sim \mathcal{N}(\mathbf{0}_5, \mathbf{I}_5)$ and $\mathbf{x} \sim \mathcal{N}(\mathbf{0}_5, \mathbf{I}_5)$, and setting $y = \theta^\top \mathbf{x} + e$, where $e \sim \mathcal{N}(0, 0.1)$. We fix the number of shots to $m = 20$ during pretraining. For fine-tuning, we first randomly generate a target task vector $\theta_0 \in \mathbb{R}^5$ such that $\|\theta_0\|_2 = 1$. We then construct the fine-tuning dataset by sampling $\mathbf{x} \sim \mathcal{N}(\mathbf{0}_5, \mathbf{I}_5)$ and generating $y = \theta_0^\top \mathbf{x} + e$, where $e \sim \mathcal{N}(0, 0.1)$. When using the auxiliary few-shot loss during fine-tuning, we fix the number of shots to $n = 20$.

**Architectural Variants.** Figure 5 reports results for the linear self-attention model in (1), which uses a merged query-key matrix. In additional experiments, we also consider a variant with separate query and key matrices, following the standard attention formulation, as well as a 2-head linear attention model.

In our framework, the multi-head extension is straightforward when query-key parameters are shared across heads, since the output reduces to a linear combination of the value matrices and is equivalent to a single-head model with an aggregated value matrix. We note, however, that our 2-head experiment does not impose this shared query-key structure. Even in this

more flexible setting, the results exhibit trends consistent with those of the single-head model, suggesting that the qualitative behavior is not specific to the merged query-key formulation.

**Additional Results.** We evaluate $n$-shot test errors on the in-distribution task $\theta_0$ and the out-of-distribution task $-\theta_0$ across fine-tuning strategies. Empty circles denote models fine-tuned with the zero-shot (ZS) loss only, whereas filled circles denote models fine-tuned with both the zero-shot and few-shot (FS) losses. Parentheses indicate which parameters are updated: `all` updates all parameters, while `K`, `Q`, and `V` update the key, query, and value matrices, respectively. Value-matrix fine-tuning without the auxiliary few-shot loss achieves the lowest 20-shot error on the out-of-distribution task $-\theta_0$.

### B.2. Experimental Setup and Additional Results for the MMLU Benchmark

*Table 2.* Zero-shot and 7-shot accuracies evaluated on four MMLU categories (*Humanities*, *Social Sciences*, *STEM*, and *Other*). We fine-tune the base model (`Qwen2.5-3B-Instruct`) on the *Humanities* subset. ZS and FS denote fine-tuning with the zero-shot objective and the auxiliary few-shot objective, respectively, and Q, K, and V indicate that LoRA adapters are applied to the query, key, and value matrices, respectively. The second row for each model reports the difference from the base model in percentage points.

| Model | 0-shot accuracies | | | | 7-shot accuracies | | | |
|---|---|---|---|---|---|---|---|---|
| | *Humanities* | *Social Sciences* | *STEM* | *Other* | *Humanities* | *Social Sciences* | *STEM* | *Other* |
| Base model | 63.62 (0.11) | 70.12 (0.14) | 70.69 (0.17) | 63.52 (0.12) | 70.13 (0.15) | 75.33 (0.12) | 71.77 (0.17) | 64.06 (0.07) |
| ZS fine-tuning on Q/K/V | 69.30 (0.16) | 75.15 (0.14) | 68.27 (0.20) | 63.65 (0.09) | 66.06 (0.06) | 73.26 (0.19) | 66.72 (0.11) | 62.91 (0.11) |
| | +5.69 | +5.03 | -2.42 | +0.13 | -4.07 | -2.06 | -5.05 | -1.15 |
| ZS fine-tuning on Q | 66.05 (0.11) | 71.77 (0.36) | 67.42 (0.38) | 60.83 (0.13) | 68.57 (0.14) | 73.13 (0.16) | 71.17 (0.27) | 64.01 (0.23) |
| | +2.44 | +1.65 | -3.28 | -2.69 | -1.55 | -2.19 | -0.60 | -0.05 |
| ZS fine-tuning on K | 66.42 (0.22) | 73.99 (0.18) | 70.76 (0.27) | 67.14 (0.18) | 69.06 (0.12) | 75.01 (0.19) | 74.07 (0.45) | 67.02 (0.14) |
| | +2.81 | +3.87 | +0.06 | +3.62 | -1.06 | -0.32 | +2.30 | +2.95 |
| ZS fine-tuning on V | 70.21 (0.18) | 76.40 (0.21) | 65.22 (0.37) | 66.13 (0.02) | 67.75 (0.07) | 72.86 (0.31) | 71.35 (0.52) | 64.74 (0.18) |
| | +6.59 | +6.28 | -5.47 | +2.61 | -2.38 | -2.47 | -0.42 | +0.68 |
| ZS+FS fine-tuning on V | 69.62 (0.07) | 77.96 (0.17) | 52.25 (0.09) | 67.48 (0.40) | 68.36 (0.31) | 75.56 (0.06) | 65.29 (0.30) | 64.70 (0.10) |
| | +6.00 | +7.84 | -18.45 | +3.96 | -1.77 | +0.23 | -6.48 | +0.64 |

**Experimental Setup.** We use the LLaMAFactory framework (Zheng et al., 2024) for all experiments. All runs are conducted on a single NVIDIA RTX 4090 GPU or NVIDIA RTX A5000 GPU.

We fine-tune `Qwen2.5-3B-Instruct` (Yang et al., 2024) on the *Humanities* subset of MMLU (Hendrycks et al., 2021), which contains 1,000 training examples. For each fine-tuning instance, we use `gpt-4.1-mini` (Achiam et al., 2023) to generate a chain-of-thought rationale, and train the model on prompts that include both the final answer and the reasoning trace. Fine-tuning is performed for 5 epochs using low-rank adaptation (LoRA) (Hu et al., 2022) with rank $r = 128$. We apply adapters either to all attention projection matrices or only to the query (Q), key (K), and value (V) projections. We adopt LoRA as a parameter-efficient approach for optimizing the zero-shot objective (ZS), which is also theoretically motivated by our analysis where the optimal parameter change admits a rank-one update under value-matrix fine-tuning.

We sweep the learning rate over $\{1 \times 10^{-3}, 3 \times 10^{-4}, 1 \times 10^{-4}, 3 \times 10^{-5}, 1 \times 10^{-5}\}$ and select the model that achieves the best zero-shot performance on the *Humanities* category. When using the zero-shot objective with the auxiliary few-shot objective (ZS+FS), we sweep the initial weight of the few-shot loss term over $\{0.25, 0.50, 0.75, 1.00\}$ and linearly anneal it to 0 within each epoch. We evaluate the resulting models on the MMLU test set across four categories (*Humanities*, *Social Sciences*, *STEM*, and *Other*), consisting of 1,000 examples in total. Inference is performed with a temperature of 0.05, evaluation is repeated three times, and the mean accuracy is reported along with the standard error.

**Additional Results.** Our fine-tuning objective is primarily to improve performance on the *Humanities* category. Therefore, the models trained with *ZS fine-tuning on Q/K/V*, *ZS fine-tuning on V*, and *ZS+FS fine-tuning on V* achieve substantial zero-shot improvements on *Humanities*, with gains of +5.69, +6.59, and +6.00 percentage points (pp), respectively, whereas *ZS fine-tuning on Q* and *ZS fine-tuning on K* yield considerably smaller gains (+2.44 and +2.81 pp). In the remainder, we therefore focus on the three competitive variants.

These trends change in the 7-shot setting. In particular, *ZS fine-tuning on Q/K/V* substantially harms few-shot performance, reducing 7-shot *Humanities* accuracy by −4.07 pp and 7-shot *STEM* accuracy by −5.05 pp. In contrast, *ZS fine-tuning on V*

better preserves few-shot generalization, with only a $-2.38$ pp change on 7-shot *Humanities* and a $-0.42$ pp change on 7-shot *STEM*. Moreover, incorporating the auxiliary few-shot objective (*ZS+FS fine-tuning on V*) further improves few-shot performance on categories closer to the fine-tuning domain: it increases 7-shot *Social Sciences* by $+0.23$ pp and reduces the 7-shot *Humanities* drop to $-1.77$ pp (compared to $-2.38$ pp without the auxiliary loss). However, this auxiliary objective comes at a clear cost on *STEM*, where it decreases 7-shot *STEM* accuracy by $-6.48$ pp (vs. $-0.42$ pp without the auxiliary loss). Overall, these results indicate that value-matrix fine-tuning best preserves few-shot capability, while the auxiliary few-shot objective primarily improves few-shot performance on semantically similar categories but can significantly harm out-of-domain generalization. Detailed results are reported in Table 2 and are summarized in Figure 6.

**Auxiliary MMLU Word-Count Task.**    To reduce the possibility that the evaluated task has already been learned during prior training, we additionally construct an auxiliary task based on MMLU. For each MMLU question, the model is asked to predict the word count of the question by selecting one of four bins, 0 to 9, 10 to 19, 20 to 29, or 30 or more words, instead of predicting the original answer. This modification changes the target label while keeping the input distribution unchanged. In this setting, the base model achieves 26.67% zero-shot accuracy, which is close to chance and suggests that the task is unlikely to have been explicitly learned during prior training.

We evaluate two models in the 7-shot setting: full fine-tuning with the zero-shot loss and value matrix fine-tuning with the zero-shot loss. On this auxiliary task, value matrix fine-tuning achieves 47.76% accuracy, while full fine-tuning achieves 46.57% accuracy. This result is consistent with our theoretical analysis, but should be interpreted as supportive rather than conclusive evidence because the performance gap is small and we have not explored a broad range of settings.

