# OpenReview forum: "Fine-Tuning Without Forgetting In-Context Learning: A Theoretical Analysis of Linear Attention Models"
_ICML.cc/2026/Conference — ICML 2026 regular_

### Official Review · Reviewer_Q3jd · 2026-02-22

**Soundness:** 3
**Presentation:** 4
**Significance:** 3
**Originality:** 3
**Overall Recommendation:** 4
**Confidence:** 3

**Summary:**

This paper investigates the trade-off between zero-shot and few-shot performance when fine-tuning large language models. Using linear attention models as a tractable theoretical framework, the authors analyze how fine-tuning modifies attention parameters and identify conditions leading to degraded in-context learning. The central finding is that full fine-tuning optimizes zero-shot performance on a target task but significantly harms few-shot learning, even on that same task. In contrast, restricting updates solely to the value matrix (while freezing the query-key matrix) preserves strong few-shot capabilities while still improving zero-shot performance. The authors also show that incorporating an auxiliary few-shot loss during fine-tuning can further enhance performance on the target task at the expense of generalization to out-of-distribution tasks. The theoretical results are empirically validated on both synthetic linear regression tasks and a language model (Qwen2.5-3B-Instruct) on the MMLU benchmark.

**Compliance With Llm Reviewing Policy:**

Affirmed.

**Final Justification:**

I believe this paper offers valuable technical and theoretical contributions. The authors' rebuttal has addressed my concerns; therefore, I maintain my original score.

**Key Questions For Authors:**

1. How might the theoretical framework and conclusions change if we consider a multi-head linear attention architecture? Does the recommendation to fine-tune only the value matrix apply to all heads equally, or would a selective strategy per head be theoretically justified?

2. Based on your findings, what specific, practical fine-tuning protocol (e.g., initial learning rate for value parameters, scheduling for the auxiliary loss weight, choice of w) would you recommend to practitioners aiming to balance zero-shot gains and few-shot retention for a new downstream task?

**Limitations:**

yes

**Strengths And Weaknesses:**

Strengths:
1. Rigorous Theoretical Analysis: Provides clear, closed-form solutions for optimal parameters and test errors under distinct training regimes (pretraining, full fine-tuning, value-matrix fine-tuning), offering deep insight into the mechanics of the trade-off.

2. Well-Structured Comparative Framework: Systematically compares four key training regimes (summarized effectively in Table 1), clearly demonstrating the performance consequences of different fine-tuning strategies on both in-distribution and out-of-distribution tasks.

3. Clear and Actionable Insight: The identified strategy of value-matrix fine-tuning is a concrete, parameter-efficient method to mitigate the forgetting of in-context learning, with direct practical implications for model adaptation.

4. Strong Empirical Validation: The experiments on linear regression tasks show a precise match with theoretical predictions, while the supplementary experiments on Qwen2.5 provide encouraging evidence that the core phenomenon extends to large-scale, non-linear transformers.

Weaknesses:
1. Limitation of the Linear Model: The theoretical analysis relies on simplified linear attention models. While the authors note this and provide supporting experiments, the extent to which the precise mathematical conclusions (e.g., the exact form of the optimal V matrix) translate to standard softmax Transformers with multiple layers and heads remains an open question.

2. Strong Assumptions in Theory: The theoretical derivations depend on specific assumptions, such as the input covariance matrix Σ being known and positive definite, and task vectors being drawn from a normal distribution. The real-world complexity of data and tasks may deviate from these idealized conditions.

3. Relatively Limited Empirical Scope: The primary validation on language models is confined to one model (Qwen2.5-3B-Instruct) and one benchmark (MMLU), focusing only on the Humanities-STEM split. Broader validation across more models, tasks, and fine-tuning scales would strengthen the claim's generality.

---

> ### Author Rebuttal · Authors · 2026-03-31
>
> We appreciate the reviewer for finding our theoretical analysis **rigorous**, our comparative framework **well-structured**, and our value-matrix fine-tuning insight **clear and actionable** with **strong empirical validation**. Below, we address each of the reviewer’s comments in detail.
>
> ---
>
> # Q1. What if we consider a multi-head linear attention?
>
> Under our framework, extending to multi-head linear attention is straightforward when query-key parameters are shared across heads, since the output reduces to a linear combination of value matrices and is equivalent to a single-head model with an aggregated value matrix. Accordingly, the recommendation to fine-tune only the value matrix applies uniformly across heads, and our analysis does not suggest a need for head-specific selective tuning.
>
> This is empirically supported in Appendix C, where the 2-head case (Figure 8) shows trends consistent with the 1-head case (Figure 7), as discussed in the last paragraph of Appendix C.1.
>
> ---
>
> # Q2. Practical fine-tuning protocol.
>
> We thank the reviewer for this practical question. Our theoretical analysis suggests the following protocol for balancing zero-shot gains and few-shot (in-context learning) retention.
>
> We recommend defaulting to value-only fine-tuning, as it achieves zero-shot gains on the target task while preserving in-context learning ability. Whether to additionally introduce an auxiliary few-shot objective depends on the similarity between the tasks used in fine-tuning and the tasks whose in-context learning ability we wish to preserve. Proposition 4.10 quantifies this, showing that the degradation in in-context learning ability for a specific downstream task scales as $(1-\rho)^2$, where $\rho$ is the cosine similarity between the task vectors of the fine-tuning and downstream tasks, and task vectors parametrize the underlying data generating process of each task as defined in L103. When tasks are closely related ($\rho \approx 1$), auxiliary few-shot tuning is relatively safe and can further improve preservation. When downstream tasks diverge from the fine-tuning tasks, however, the degradation grows proportionally to $(1-\rho)^2$, which can become substantial.
>
> We note that this protocol is formally derived only within our linear attention setting, and its applicability to broader architectures remains to be validated empirically. Furthermore, extending this notion of task similarity to real-world datasets, where task vectors are not explicitly defined, is an important direction for future work. We therefore recommend treating it as a principled heuristic for broader practice.
>
> ---
>
> # W1. Limitation of the linear model.
>
> We acknowledge this as a limitation and discuss it explicitly in the last paragraph on page 8 of our manuscript.
>
> Expanding the number of heads is a natural extension within our framework (Q1), but extending to deeper architectures and softmax-based attention requires additional theoretical development, which we leave for future work.
>
> We observe similar trends in both our MMLU experiments on Qwen2.5, which is based on a standard softmax Transformer, and our experiments with multi-head linear attention models reported in Appendix C.1.
>
> ---
>
> # W2. Strong assumptions in theory.
>
> We acknowledge that these assumptions are simplifications for theoretical tractability. At the same time, as noted in L104, setting q=0 in our model recovers the formulation of prior work, and some existing theoretical studies impose even stricter constraints on the parameters. In this sense, our setup is more general within the landscape of theoretical analyses of linear Transformers.
>
> Regarding the specific assumptions: (1) The Gaussian task vector assumption applies only to the pretraining objective. All of our fine-tuning results hold for an arbitrary fixed target task $\theta_0$ without any distributional assumption. (2) The model does not have direct access to the covariance matrix. Rather, Corollary 3.1 shows that through pretraining, the model implicitly learns parameters that approximate the inverse of the covariance matrix in the query–key matrix. The covariance matrix thus appears in our analysis as the population quantity underlying the data, not as an input to the model.
>
> ---
>
> # W3. Relatively limited empirical scope.
>
> We respectfully note that our empirical study goes beyond the Humanities–STEM split. In Appendix C.2, we provide detailed zero-shot and few-shot results across all four MMLU categories (Humanities, Social Sciences, STEM, and Other), along with the corresponding interpretation. In addition, as discussed in our response to Reviewer PgXU’s `Q2. & W1.`, we include further experiments on the fine-tuned model’s ability to count the number of words in a sentence, providing complementary empirical evidence. We nonetheless agree that these experiments are illustrative rather than exhaustive, and we will revise the text to clarify this scope.

---

> > ### Author Rebuttal · Reviewer_Q3jd · 2026-04-01
> >
> > The author's reply addressed my concerns.

---

### Official Review · Reviewer_AVxH · 2026-03-02

**Soundness:** 3
**Presentation:** 4
**Significance:** 2
**Originality:** 3
**Overall Recommendation:** 4
**Confidence:** 4

**Summary:**

This paper provides a theoretical and empirical analysis of the trade-off between zero-shot adaptation and in-context learning (ICL) in Transformer models, specifically focusing on how fine-tuning objectives modify attention parameters. Using linear attention models as an analytically tractable proxy, the authors demonstrate that while fine-tuning all attention parameters optimizes a model for a specific target task in a zero-shot setting, it severely degrades its ability to learn from examples in-context, even on the target task itself. To mitigate this "forgetting" of ICL capabilities, they propose value-matrix fine-tuning—restricting parameter updates to the value matrix ($V$) while freezing the query-key matrices ($Q$)—which preserves the underlying mechanism for few-shot learning while still achieving competitive zero-shot performance. Their findings, validated on both linear regression tasks and the MMLU benchmark with large-scale models like Qwen2.5, further reveal that adding an auxiliary few-shot loss can boost ICL on the target domain but at the expense of generalization to out-of-distribution tasks.

**Compliance With Llm Reviewing Policy:**

Affirmed.

**Final Justification:**

This paper provide comprehensive theoretical results to support the experiment intuition and hence I recommeend accept.

**Key Questions For Authors:**

- Could the theoretical framework be extended to a "mixed-training" regime where the model is optimized on a weighted sum of zero-shot and few-shot losses simultaneously, and would the functional separation of the value matrix ($V$) and query-key ($QK$) circuits emerge naturally as the global optimum in that setting?
- For the empirical results, how does updating all parameters using both zero-shot and auxiliary few-shot losses compare to the restricted value-matrix approach—specifically, does the added flexibility allow it to outperform on the target task, or does it further exacerbate the degradation of out-of-distribution performance?
- Beyond the linear regression case, can the success of value-matrix-only fine-tuning be linked to the preservation of specific circuits in real-world LLMs, such as "induction heads," where the $QK$ circuit handles the "where to look" logic of in-context learning while the $V$ matrix (or $OV$ circuit) handles the task-specific "what to copy"?

**Limitations:**

yes

**Strengths And Weaknesses:**

Strength:
-  The paper provides a clear, closed-form characterization of four different training regimes, explicitly demonstrating the trade-offs between zero-shot and few-shot performance.  It identifies that restricting updates to the value matrix ($V$) is a superior method for preserving in-context learning (ICL) while still improving zero-shot performance.
- The theoretical predictions are directly backed by experiments on linear regression tasks, where the empirical test error points perfectly align with the derived theoretical curves. The authors extend their toy model findings to a large language model (Qwen2.5-3B-Instruct) on the MMLU benchmark, showing that the observed trade-offs persist in real-world, large-scale architectures.

Weakness:
- The model architecture forces several blocks of the $V$ and $Q$ matrices to zero, which may not fully reflect the complexities of standard Transformer layers.
- The theoretical results is not surpring given the designed finetuning method, .e.g, pretraining on few-shot loss but test on zero-shot cases which can definitely fails & fine-parameter finetuning on few-shot task breaks the original few-shot pretrained model

---

> ### Author Rebuttal · Authors · 2026-03-31
>
> We appreciate the reviewer for finding our closed-form characterization of the four training regimes **clear** and our theoretical predictions **precisely aligned** with the empirical results. Below, we address each of the reviewer’s comments in detail.
>
> ---
>
> # Q1. Could the theoretical framework be extended to a mixed-training regime where the model is optimized on a weighted sum of zero-shot and few-shot losses simultaneously?
>
> We thank the reviewer for this insightful question about viewing our analysis as functional separation. We partially address this setting in Section 3.4, which analyzes the mixed-training regime under the assumption that the query-key matrix is fixed at its pretrained value. In this setting, the functional separation of the value matrix still emerges. In particular, Theorem 3.9 shows that $\hat{V}(w^\star)$, where $w^\star$ is given in Equation (13), is a global optimum of the value matrix, and $\hat{V}(w^\star)$ contains the target task vector $\theta_0$ scaled by $\frac{1}{d+4}$ as one of its blocks, as shown in Equation (11).
>
> ---
>
> # Q2. Does the added flexibility allow it to outperform on the target task, or does it further exacerbate the degradation of out-of-distribution performance?
>
> The added flexibility of updating all parameters improves target-task performance, achieving lower test error, but further exacerbates degradation in out-of-distribution performance. In contrast, the restricted value-matrix approach better preserves few-shot performance across tasks. These empirical results are shown by the blue dots in Figures 5, 7, and 8, as discussed in the last paragraph of Section 4.1 and Appendix C.1 of our manuscript.
>
> ---
>
> # Q3. Can the success of value-matrix-only fine-tuning be linked to the preservation of specific circuits in real-world LLMs, such as the $QK$ circuit handles the 'where to look' logic of in-context learning while the $V$ matrix handles the task-specific 'what to copy'?
>
> This is an interesting perspective. Our theoretical results admit an interpretation that shares some high-level similarities with circuit-level views you mentioned, while differing in the underlying mechanism. In particular, the query–key interaction does not perform token-level selection ('where to look'), but instead aggregates information across the context to form a task representation (L128).
>
> In our framework, the value matrix can be viewed as storing a parametric task representation encoded in the model weights (L244). When no context is provided, predictions are driven by this parametric representation. When context is available, the query–key mechanism extracts a task vector from the context, which is then used for prediction. From this perspective, value-only fine-tuning preserves the query–key mechanism for context aggregation, while updating the value matrix to encode the target task in a parametric form (L220).
>
> We emphasize that, despite superficial similarities, our interpretation is closer to context aggregation than token-level retrieval. Moreover, our theoretical results are derived in the setting of linear regression tasks, and whether this perspective extends to real-world tasks remains an open question.
>
> ---
>
> # W1. The model architecture forces several blocks of the value and key matrices to zero.
>
> We acknowledge that these assumptions are simplifications for theoretical tractability. At the same time, as noted in L104, setting q=0 in our model recovers the formulation of prior work, and some existing theoretical studies impose even stricter constraints on the parameters. In this sense, our setup is more general within the landscape of theoretical analyses of linear Transformers.
>
> ---
>
> # W2. The theoretical results are not surprising given the designed finetuning method.
>
> Our key contribution is to show that this behavior is not merely a consequence of a particular training setup, but arises from the structure of the model itself. In particular, Theorem 3.5 demonstrates that updating only the value matrix can lead to zero-shot performance exceeding few-shot performance, thereby motivating value-only fine-tuning.
>
> While this phenomenon may appear intuitive in hindsight, it is not obvious a priori that restricting updates to the value matrix, as opposed to updating the query and key matrices, would mitigate the trade-off between zero-shot and few-shot performance, nor that the effect cannot simply be attributed to a general mismatch between fine-tuning objectives and in-context learning behavior.
>
> Moreover, as discussed in the paragraph starting from L192, we show that even on the same task, optimizing for zero-shot performance can degrade few-shot performance, further indicating that this effect cannot be explained solely by differences between training and testing setups.

---

> > ### Author Rebuttal · Reviewer_AVxH · 2026-04-02
> >
> > The author's response has solved my concerns and hence I increase the score.

---

### Official Review · Reviewer_PgXU · 2026-03-11

**Soundness:** 3
**Presentation:** 3
**Significance:** 4
**Originality:** 4
**Overall Recommendation:** 5
**Confidence:** 3

**Summary:**

Transformer models display trade-offs on zero-shot and few-shot task performance when it comes to in-context learning capability before and after task finetuning. Through a theoretical analysis using linear attention models as proxies for standard transformers, the paper derives closed form terms modeling the error of zero-shot and few-shot (in-context) task performance. Using these insights, they determine that finetuning the value matrix only retains few-shot (in-context) performance while improving zero-shot performance on the target task. They empirically validate on both actual linear model and standard pretrained model performance.

**Compliance With Llm Reviewing Policy:**

Affirmed.

**Key Questions For Authors:**

1. Is the assumption of average m-shot test error minimization over many tasks actually representative of pretraining, i.e. is it supported by any evidence?
2. As far as I am aware, the Qwen model used is not just a pretrained model but also undergoes task instruction finetuning, so is it not better to have used a purely pretrained model to reduce the risk of the model being already finetuned on the task and hence not being representative of the theoretical assumptions?

**Limitations:**

yes

**Strengths And Weaknesses:**

**Strengths:**
- The theoretical contribution is interesting and a welcome addition to the literature on in-context learning hopefully inspiring further rigorous work.
- The extension of previous work on linear attention models (namely Ahn et al 2024) to understand why and how task finetuning affects in-context learning ability is novel to the best of my knowledge.
- The closed-form bounds derived are intuitive and explain some of the empirical observations of in-context learning quite nicely.
- The proofs are outlined in detail (in the appendix), and while I have not rigorously verified their correctness they at least seem thorough enough from a quick glance.

**Weaknesses:**
- The empirical verification is in my opinion quite limited in terms of models and tasks, also due to the use of a closed-data model we are not able to fully know whether the MMLU task is in-distribution or out-of-distribution and how to interpret that within the context of the theoretical analysis. Hence I am unsure whether the claim of the results being “empirical validated” in the abstract holds fully in this iteration of the work. Perhaps softer language like that used in the introduction (“empirical results consistent with the theoretical findings….”, L44) would avoid overclaiming the empirical contribution of the paper.
- I understand that the paper’s material is dense and the space may not suffice, but the lack of extended discussion of related work or the positioning of the paper’s contribution in the literature in the main paper should be addressed (I am aware of the discussion in the appendix).

---

> ### Author Rebuttal · Authors · 2026-03-31
>
> We thank the reviewer for viewing our theoretical contribution as an **interesting and welcome addition** to the in-context learning literature, and for finding our closed-form solutions **intuitive** in explaining empirical observations. We address the reviewer's constructive and detailed feedback below.
>
> ---
>
> # Q1. Is the assumption of average m-shot test error minimization over many tasks actually representative of pretraining?
>
> We thank the reviewer for this important question. We agree that minimizing the average few-shot test error over many tasks should not be interpreted as a literal objective optimized during real-world pretraining, and there is no direct empirical evidence that pretraining explicitly optimizes this quantity.
>
> Instead, this objective is widely used as an analytical surrogate to study in-context learning in a tractable setting. In particular, prior theoretical work [1,2] shows that training under such objectives can lead transformers to exhibit in-context learning behavior. We adopt this formulation to enable a tractable analysis of the mechanisms of in-context learning and their interaction with fine-tuning.
>
> We acknowledge that this assumption abstracts away several important aspects of real pretraining. We will clarify this point in the updated manuscript.
>
>
> [1] Ahn, Kwangjun, et al. "Transformers learn to implement preconditioned gradient descent for in-context learning." NeurIPS 2023.
>
> [2] Zhang, Ruiqi, et al. "Trained transformers learn linear models in-context." JMLR 2024.
>
> ---
>
> # Q2. & W1. Is it not better to have used a purely pretrained model to reduce the risk of the model being already finetuned on the task and hence not being representative of the theoretical assumptions?
>
> We agree that using a purely pretrained model would better align with our theoretical assumptions. However, in practice, most state-of-the-art language models do not disclose their exact training data and have undergone some form of instruction tuning, making it difficult to obtain comparable large-scale pretrained models. As a result, this concern is challenging to fully eliminate in current empirical settings.
>
> To mitigate this issue, we design an auxiliary task based on MMLU where the model predicts the word count of each question. For each question, the model selects one of four bins (0–9, 10–19, 20–29, 30+ words) in a multiple-choice format, instead of predicting the original answer. This changes the objective in a way that is unlikely to have been explicitly seen during training while keeping the input distribution unchanged. In this setting, the base model shows random zero-shot performance (26.67%), which suggests that the task is unlikely to have been learned during pretraining.
>
> Under a 7-shot setting, we evaluate two fine-tuned models reported in Section 4.2: (b) fine-tuning with the zero-shot loss and (c) value matrix fine-tuning with the zero-shot loss. On this auxiliary task, value-only fine-tuning achieves slightly higher performance (47.76%) compared to standard fine-tuning (46.57%), providing preliminary evidence that value-only fine-tuning better preserves in-context learning capability during fine-tuning. However, as the performance gap is small and we have not explored a wide range of settings, we consider this observation suggestive rather than conclusive.
>
> We will clarify these limitations and adopt more cautious language (e.g., revising sentences such as “empirical results consistent with the theoretical findings”), and position this experiment as supportive evidence complementing our theoretical analysis.
>
> ---
>
> # W2. The lack of extended related work discussion and clear positioning of the paper’s contributions in the main text.
>
> In the updated manuscript, we will move the related work section from Appendix into the main paper, and revise it to better highlight our contributions in the context of existing literature.

---

> > ### Author Rebuttal · Reviewer_PgXU · 2026-04-02
> >
> > I maintain my positive score.

---

### Official Review · Reviewer_geo1 · 2026-03-13

**Soundness:** 3
**Presentation:** 3
**Significance:** 3
**Originality:** 3
**Overall Recommendation:** 5
**Confidence:** 2

**Summary:**

The paper investigates the underlying mechanisms by which task-specific fine-tuning degrades the in-context learning capabilities of pre-trained language models. By analyzing a linear self-attention model trained on continuous linear regression tasks, the authors derive closed-form mathematical expressions for test errors under various fine-tuning regimes. The analysis reveals that full-parameter fine-tuning to optimize zero-shot performance intrinsically destroys few-shot capabilities by overriding the model's similarity-matching engine. The authors theoretically prove and empirically validate that restricting gradient updates exclusively to the value projection matrix allows the model to optimize zero-shot accuracy while preserving the query-key similarity calculations necessary for robust in-context learning.

[Disclaimer]: I want to highlight that I am not an expert on this topic - I have only very basic understanding of model analysis. While I tried my best in understanding the paper and linking it to existing works, I might have missed / misunderstood key points

**Compliance With Llm Reviewing Policy:**

Affirmed.

**Key Questions For Authors:**

-

**Limitations:**

yes

**Strengths And Weaknesses:**

Strengths:
- The paper aims to  translate an empirical problem (catastrophic forgetting of capabilities) into a tractable mathematical framework.
- The papers conclusion that value-matrix tuning is mathematically optimal for preserving in-context learning is validated through parameter-efficient fine-tuning methods like LoRA.

Weaknesses:
- The paper relies exclusively on single-layer linear attention without MLP or softmax functions  which drastically reduces the direct generalizability of the proofs to modern transformer architectures.

---

> ### Author Rebuttal · Authors · 2026-03-31
>
> We thank the reviewer for recognizing the key strength of our work in formulating catastrophic forgetting within a **tractable mathematical framework** and for acknowledging the **mathematical and empirical validity** of value-matrix tuning.
>
> ---
>
> # W1. The paper relies exclusively on single-layer linear attention.
>
> We acknowledge the limitation of adopting a single-layer linear attention model for analytical tractability. Extending our analysis to modern Transformer architectures is an important direction, and we discuss this point in L427 of the manuscript. We also refer the reviewer to our responses to Reviewer Q3jd's `Q1.` and `W1.`, where we provide additional clarification. These points will be incorporated into the revised version.

---

> > ### Author Rebuttal · Reviewer_geo1 · 2026-03-31
> >
> > authors will address my concerns in the revised paper

---

### Decision · Program_Chairs · 2026-04-30

**Decision:**

Accept (regular)

**Comment:**

This work studies the attention mechanism underlying transformers’ in-context learning. Its main strength is a clear theoretical analysis of how fine-tuning affects the generalization ability of transformers. The main weaknesses are that the theory relies heavily on simplified linear-attention assumptions and that the experimental validation on realistic transformer-based applications is still limited. Overall, the reviewers were positive about the paper’s theoretical contribution, though there also appears to be a noticeable discrepancy in the self ranking.